# The asymptotic structure of cosmological integrals

**Paolo Benincasa[1,2]⋆ and Francisco Vazão[1]†**

**1** Max-Planck-Institut für Physik, Werner-Heisenberg-Institut,
D-80805 München, Germany
**2** Instituto Galego de Física de Altas Enerxías IGFAE,
Universidade de Santiago de Compostela,
E-15782 Galicia-Spain

⋆ pablowellinhouse@anche.no , † fvvazao@mpp.mpg.de

## Abstract

We provide a general analysis of the asymptotic behaviour of perturbative contributions to observables in arbitrary power-law FRW cosmologies, indistinctly the Bunch-Davies wavefunction of the universe and cosmological correlators. We consider a large class of scalar toy models, including conformally-coupled and massless scalars in arbitrary dimensions, that admits a first principle definition in terms of (generalised/weighted) cosmological polytopes. The perturbative contributions to an observable can be expressed as an integral of the canonical function associated to such polytopes and to site- and edge-weighted graphs. We show how the asymptotic behaviour of these integrals is governed by a special class of *nestohedra* living in the graph-weight space, both at tree and loop level. As the singularities of a cosmological process described by a graph can be associated to its subgraphs, we provide a realisation of the nestohedra as a sequential truncation of a top-dimensional simplex based on the underlying graph. This allows us to determine all the possible directions – both in the infra-red and in the ultra-violet –, where the integral can diverge as well as their degree of divergence. Both of them are associated to the facets of the nestohedra, which are identified by overlapping tubings of the graph: the specific tubing determines the divergent directions while the number of overlapping tubings its degree of divergence. This combinatorial formulation makes straightforward the application of sector decomposition for extracting the – both leading and subleading – divergences from the integral, as the sectors in which the integration domain can be tiled are identified by the collection of compatible facets of the nestohedra, with the latter that can be determined via the graph tubings. Finally, the leading divergence has a beautiful interpretation as a restriction of the canonical function of the relevant polytope onto a special hyperplane.

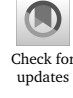

# 1   Introduction

Recent years have seen a big deal of progress in understanding the perturbative structure of cosmological observables and, more generally, observables in a fixed and expanding background [1–4]. As cosmological observables provide the initial conditions for the classical evolution leading to the patterns that we can observe in the cosmic microwave background (CMB) and in the large scale structures (LSS) of our universe, having control over their perturbative structure is of crucial importance for understanding how the physics of the pre-hot big bang era affects the actual late-time observables. Furthermore, precisely as they encode the physics of the pre-hot big bang phase – an elegant description of which is provided by inflation – a deeper understanding of their analytic structure would provide a handle on how to extract fundamental physics out of them.

Particular focus has been put on the Bunch-Davies wavefunction of the universe: despite not being an observable in the purest sense of the word, it enjoys physical properties such as gauge invariance, and its squared modulus provides the probability distribution of the field configurations at the space-like boundary. Because of the Bunch-Davies condition, no particle decay is allowed in the physical region. This implies that the perturbative wavefunction involving a set $\mathcal{N} := \{1, \ldots, n\}$ of $n$ external states cannot have singularities of the type $\sum_{j \in \mathcal{L}} \sigma_j E_j + \sum_{e \in \mathcal{E}_{\mathcal{L}}^{\text{ext}}} \sigma_e y_e = 0$, where $E_j = |\vec{p}_j|$ – which, with an abuse of language, we will refer to as *energy* of the $j$-the external state –, $\mathcal{L} \subset \mathcal{N}$ is a subset of the external states, $\mathcal{E}_{\mathcal{L}}^{\text{ext}}$ is the set of states which are internal to $\mathcal{N}$ but external to $\mathcal{L}$, $\{y_e, \ e \in \mathcal{E}_{\mathcal{L}}^{\text{ext}}\}$ is the set of their energies, and $\sigma_{j/e} = \pm$ in such a way that not all of them are equal. As a consequence, the perturbative wavefunction can show singularities just outside the physical region at the loci $E_{\mathcal{N}} := \sum_{j=1}^{n} E_j = 0$, *i.e.* the total energy for the full process $\mathcal{N}$, and $E_{\mathcal{L}} := \sum_{j \in \mathcal{L}} E_j + \sum_{e \in \mathcal{E}_{\mathcal{L}}^{\text{ext}}} y_e = 0$, $E_{\mathcal{L}}$ being the total energy of a subprocess involving the states in $\mathcal{L}$. As the total energy conservation locus $E_{\mathcal{N}} = 0$ is approached, the wavefunction reduces to the (high energy limit of) the flat-space scattering amplitude [5, 6], while at $E_{\mathcal{L}} = 0$ the wavefunction factorises into a flat-space scattering amplitude associated to the subprocess $\mathcal{L}$ and a certain linear combination of wavefunctions associated to $\mathcal{N} \setminus \mathcal{L}$ – see [2] and references therein. These properties have been used to compute the wavefunction of certain tree-level processes [7–10].

The cosmological correlation functions are related to the wavefunction via the squared modulus of the latter and turn out to inherit a number of properties from it, such as its very same singularity structure, with the addition of the region of kinematic space where the real part of the two-point wavefunction vanishes; the factorisation properties as the singularities shared with the wavefunction are approached; and a contour integral representation at a given order in perturbation theory [4].

Most of the explicit results have been obtained at tree-level, while the loop level computation still remains an open challenge despite some recent advances [11–13]. Even if loop corrections can be quite smaller than the tree-level ones and, consequently, difficult to detect observationally, the understanding of their structure is of crucial importance to test the robustness of the tree-level predictions and the more general consistency of the theory itself: they can possibly be responsible for large infra-red effects when light states are involved [14–52], which could lead to the instability of the expanding space-time itself [14–19, 21, 34].

There are strong indications that these infra-red effects can be re-summed [27, 36, 42, 47]. In particular, it has been shown that for states at equal points, the de Sitter wavefunction factorises into a term containing the leading logarithms and another term describing higher energy modes, with the re-summed probability distribution associated to it satisfying the Fokker-Planck equation [47] and agreeing with Starobinski's stochastic inflation proposal [53, 54]. A similar conclusion has been reached via different methods, *e.g.* using a dynamical renormalisation group approach [55–60]; via open effective field theory [61]; analysing more general correlation functions [42]; combining the wavefunction point of view with the exact renormalisation group [51]. It has been also argued that such a Markovian behaviour still persists at subleading order [48]. If the resummation of the leading divergences into a solution of the Fokker-Planck equation reached some consensus, much less known – and understood – is what happens with the subleading ones, despite, at least in line of principle, the approaches used in [61] and in [42] are suited for obtaining more insights. Also, most of these analyses rely on considering strictly massless modes in a fixed de Sitter space-time. However, the following questions, at least for our understanding, goes still unanswered: given a system of sufficiently light states in arbitrary FRW cosmologies suffering from infra-red divergences,[1] which conditions the divergent terms need to satisfy in order to come from a resummed, well-behaved, quantity? Said differently, how we can know from the very beginning that, despite a correlator (or the associated wavefunction[2]) shows such divergences, they are bounded to re-sum and hence the theory is well-behaved? Even more importantly, one can ask how perfectly infra-red finite observables can be defined?

In this paper, we begin a program that aims to address these issues. We take a step back and analyse the asymptotic behaviour of cosmological integrals for power-law FRW cosmologies. These integrals are associated to weighted graphs which ca appear in the perturbative expansion of the Bunch-Davies wavefunction and of the cosmological correlators. These graphs $\mathcal{G} := \{\mathcal{V}, \mathcal{E}\}$ are characterised by weights $\{x_s, y_e, s \in \mathcal{V}, e \in \mathcal{E}\}$ associated to both their sites $\mathcal{V}$[3] and their edges $\mathcal{E}$. The site weights parametrise the space of the moduli of the momenta of the external states; while the edges weights instead parametrise the space of the angles among the external momenta if the related edge is in a tree substructure of $\mathcal{G}$, or the loop

---

[1] We would like to stress that with *infra-red divergences* we refer to both those divergences associated to loop corrections and the so-called secular effects.

[2] Despite the wavefunction coefficients do not usually suffer from infra-red divergences coming from loop corrections, they can have divergences due to the space-time expansion – these infinite volume divergences have been argued that they can be subtracted with a procedure similar to holographic renormalisation [52]. An alternative subtraction procedure, based on the analysis of the divergences presented in this paper, is possible [62]. It extends the analysis [63] for flat-space Feynman integrals to cosmological integrals.

[3] We will refer to the vertices of a graph as *sites* and reserve the word *vertices* to the geometrical picture in terms of polytopes, in order to avoid a language clash.



space if they are associated to loop edges. To these weighted graphs, it is possible to associate combinatorial structures, named *cosmological polytopes* [64], *generalised cosmological polytopes* [65, 66], and weighted cosmological polytopes [4], that describe the Bunch-Davies wavefunction and the correlation function for conformally coupled as well as light scalars with polynomial interactions. In particular, each of them is univocally associated to a rational function $\Omega_{\mathcal{G}}(x, y)$ associated to a given graph $\mathcal{G}$, named *canonical function*, which constitutes the integrand whose integration over the weight space returns the wavefunction or correlator associated to $\mathcal{G}$. These integrals then have the following general features:

❑ The measure of the site weight space, whose dimension is set by the number of sites $n_s$, is related to the specific cosmology – in the chosen case of a power-law cosmology, this integration becomes a multi-dimensional Mellin transform of the integrand, with the Mellin transform parameter being not only related to cosmology itself but also to the states involved [2, 65];

❑ The measure of the loop edge-weight space can be written in terms of the volume of a simplex, and the integration contour is given by the requirement that it is non-negative;[4]

❑ The integrand $\Omega_{\mathcal{G}}(x, y)$ shows poles, each of which is $1-1$ correspondence with a subgraph $\mathfrak{g} \subseteq \mathcal{G}$, and is identified by the vanishing of the linear polynomial $q_{\mathfrak{g}}(x, y)$.

The asymptotic structure of these integrals is controlled by the combinatorics of a special class of *nestohedra*,[5] whose facets encode both the directions were the integral becomes divergent and its degree of divergence along that direction. We describe a specific realisation of these nestohedra, that allows to predict possible divergent directions and the degree of divergence directly from the graph and its subgraphs. Furthermore, the compatibility conditions on its facets[6] allow identifying the sectors of the integrals that contribute to a certain divergence and allow us to apply sector decomposition to study the divergent structure along the related direction [68–74]. This provides a general framework to consistently analyse the asymptotic (both leading and subleading) divergences.

The paper is structure as follows. In Section 2, we set up the necessary background introducing the observables of interest – *i.e.* the Bunch-Davies wavefunction and the cosmological correlators – as well as the basics of their combinatorial formulation in terms of cosmological polytopes. Section 3 discusses the general properties of the class of cosmological integrals of interest in terms of site- and edge-weighted graphs, with an extensive discussion of integration measure in the edge-weight space parametrising the loop space. Section 4 contains the core of our paper, showing that the asymptotic behaviour of our class of cosmological integrals is encoded by nestohedra which enjoy a realisation in terms of tubings of the underlying graph. We also discuss the extraction of the divergent behaviour using sector decomposition. Section 5 is devoted to conclusion and outlook. Finally, Appendix A extends the discussion on the measure for the edge-weight integration and its integration contour, providing also explicit examples, while Appendix B provides a further example of the analysis of the divergences using the nestohedra structure and sector decomposition.

## 2 Observables in an expanding universe

Let us begin with discussing the generalities of the observables in an expanding universe, in particular the wavefunction of the universe and the correlation functions, as well as their re-

---

[4]This is equivalent to the statement that loop space is Euclidean.

[5]The nestohedra were introduced in [67] and are polytopes that can be written as a Minkowksi sum of simplices.

[6]Two facets of a polytope are said to be compatible if their intersection in codimension-2 is on the polytope and hence it constitutes a codimension-2 face of the polytope itself.

lation. The context is the one of a large class of scalar toy models which include conformally coupled and light states with (non-conformal) polynomial interactions in arbitrary FRW cosmologies.

**The model**  Let us consider a class of flat-space toy models for scalars with a time-dependent mass $\mu(\eta)$ and time-dependent polynomial couplings $\lambda_k(\eta)$:

$$S[\phi] = -\int_{-\infty}^{0} d\eta \int d^d x \left[ \frac{1}{2} (\partial \phi)^2 - \frac{1}{2} \mu^2(\eta) \phi^2 - \sum_{k \geq 3} \frac{\lambda_k(\eta)}{k!} \phi^k \right]. \tag{1}$$

Such a model describes a general scalar in FRW cosmologies with metric

$$ds^2 = a^2(\eta) \left[ -d\eta^2 + d\vec{x}^2 \right], \tag{2}$$

provided that the functions $\mu^2(\eta)$ and $\lambda_k(\eta)$ are taken to have the following expression [64,65]

$$\mu^2(\eta) = m^2 a^2(\eta) + 2d \left( \xi - \frac{d-1}{4d} \right) \left[ \partial_\eta \left( \frac{\dot{a}}{a} \right) + \frac{d-1}{2} \left( \frac{\dot{a}}{a} \right)^2 \right],$$

$$\lambda_k(\eta) = \lambda_k [a(\eta)]^{2 + \frac{(d-1)(2-k)}{2}}, \tag{3}$$

where "˙" indicates the derivative with respect to the conformal time $\eta$, $m$ and $\lambda_k$ on the right-hand-sides are constants, $\xi$ is a parameter which can acquire two values, $\xi = 0, (d-1)/4d$, respectively the minimal and conformal coupling. We will consider the case for which $\mu^2(\eta) = \mu_\gamma^2/\eta^2$, which corresponds to having massless states ($m = 0$) in a FRW cosmology with warp factor $a(\eta) = (-\ell_\gamma/\eta)^\gamma$, or states with an arbitrary mass in dS ($\gamma = 1$). In the first case, $\mu_\gamma^2 = 2d\gamma[\xi - (d-1)/4d][1 + (d-1)\gamma/2]$, while in the second one $\mu_1^2 = m^2 \ell_1^2 + d(1+d)(\xi + (d-1)/4d)$. The mode function $\phi_\circ^{(\nu)}$ for such states satisfying the Bunch-Davies condition in the infinite past, is given in terms of the Hankel functions

$$\phi_\circ^{(\nu)}(-E\eta) = \sqrt{-E\eta} \, H_\nu^{(2)}(-E\eta), \qquad \nu = \sqrt{\frac{1}{4} - \mu_\gamma^2}, \tag{4}$$

where the order parameter $\nu$ of the Hankel function is related to the parameter $\mu_\gamma^2$. For conformally coupled scalars, $\xi = (d-1)/4d$ and the order parameter $\nu$ can take the values $\nu = 1/2$ for $m^2 = 0$ and arbitrary $\gamma$. For minimally coupled scalars, $\xi = 0$ and $\nu$ becomes $\nu = 1/2 + (d-1)\gamma/2$ for massless states and arbitrary $\gamma$, while $\nu = \sqrt{d^2/4 - m^2\ell_1^2}$.

In our discussion we will be mainly concerned with states identified by $\nu = l + 1/2$ with $l \in \mathbb{Z}_+ \cup \{0\}$, or, which is the same, by $\mu_\gamma^2 = -l(l+1)$. The mode function of such states with arbitrary $l$ is related to the plane wave, corresponding to $l = 0$ (and, hence, $\mu_\gamma = 0$) via a differential operator [65]:

$$\phi_\circ^{(\nu)}(-E\eta) = \frac{1}{(-E\eta)^l} \hat{\mathcal{O}}_l(E) e^{iE\eta}, \qquad \hat{\mathcal{O}}_l(E) := \prod_{r=1}^{l} \left[ (2r-1) - E \frac{\partial}{\partial E} \right]. \tag{5}$$

This allows to derive processes involving these states from massless scalars with time-dependent couplings in flat-space [65].

**The Bunch-Davies wavefunction**  The Bunch-Davies wavefunction of the universe for this class of toy models is then given by

$$\Psi[\Phi] = \mathcal{N} \int_{\phi(-\infty)=0}^{\phi(0)=\Phi} \mathcal{D}\phi \, e^{iS^{(\varepsilon)}[\phi]}, \tag{6}$$

where $S^{(\epsilon)}[\phi]$ is the action (1) suitably regularised, $\Phi$ is the boundary state at future infinity. The regularisation of the action is needed as it contains phases which become infinitely oscillating at early times, as well as it can diverge for sufficiently light states as future infinity is approached.[7] Splitting the field $\phi$ into its classical part and a quantum fluctuation

$$\phi(\vec{p}, \eta) := \Phi(\vec{p})\phi_\circ^{(\nu)} + \varphi, \tag{7}$$

with the classical part satisfying the Bunch-Davies condition and the mode function $\phi_\circ^{(\nu)}$ being normalised to 1, while the quantum fluctuation having to vanish at both early ($\eta = -\infty$) and late ($\eta = 0$) times, the wavefunction of the universe acquires the form

$$\begin{aligned}
\Psi[\Phi] &= e^{iS_2^{(\varepsilon)}[\Phi]} \mathcal{N} \int_{\varphi(-\infty)=0}^{\varphi(0)=0} \mathcal{D}\varphi \, e^{iS_2^{(\varepsilon)}[\varphi] + iS_{\text{int}}^{(\varepsilon)}[\Phi,\varphi]} \\
&= e^{iS_2^{(\varepsilon)}[\Phi]} \left\{ 1 + \sum_{n\geq 1} \int \prod_{j=1}^{n} \left[ \frac{d^d p_j}{(2\pi)^d} \Phi[\vec{p}_j] \right] \sum_{L\geq 0} \tilde{\psi}_n^{(L)}(\vec{p}_1, \ldots, \vec{p}_n; \varepsilon) \right\},
\end{aligned} \tag{8}$$

where $S_2^{(\varepsilon)}$ is the (regularised) free action, $S_{\text{int}}^{(\varepsilon)}$ is the (regularised) interaction action, while the second line has been obtained by expanding the first one in perturbation theory, and $\tilde{\psi}_n^{(L)}(\vec{p}_1, \ldots, \vec{p}_n; \varepsilon)$ is the contribution from Feynman graphs with $n$-external states and at $L$-loops, whose connected part is commonly called *wavefunction coefficients*. The connected Feynman graphs for such wavefunction coefficients are such that the sites[8] are associated to the time-dependent couplings $\lambda_k(\eta)$, the external lines to bulk-to-boundary propagators $\phi_\circ$, *i.e.* the mode functions (4) and the internal lines to bulk-to-bulk propagators:

$$\tilde{\psi}_n^{(L)}\Big|_{\text{connected}} = \sum_{\mathcal{G}\subset \mathcal{G}_n^{(L)}} \tilde{\psi}_{\mathcal{G}}, \qquad \tilde{\psi}_{\mathcal{G}} = \int_{-\infty}^{0} \prod_{s\in\mathcal{V}} \left[ d\eta_s \, V_s \, \phi_\circ^{(s)} \right] \prod_{e\in\mathcal{E}} G_e\Big(y_e; \eta_{s_e}, \eta_{s'_e}\Big), \tag{9}$$

where $\mathcal{G}_n^{(L)}$ is the set of graphs with $n$ external states and $L$ loops, $\mathcal{G}$ is an element of $\mathcal{G}_n^{(L)}$, $\mathcal{V}$ and $\mathcal{E}$ be respectively the set of sites and internal edges of $\tilde{\psi}_{\mathcal{G}}$, $V_s$ is the interaction associated with the site $s$ which, in the present context, is given by $V_s = i\lambda_{k_s}(\eta_s)$, $\phi_\circ^{(s)}$ is the product of the bulk-to-boundary propagators at the site $s$, and finally $G_e\Big(y_e; \eta_{s_e}, \eta_{s'_e}\Big)$ is the bulk-to-bulk propagator associated to the internal edge $e$ with energy $y_e$:

$$\begin{aligned}
G_e\Big(y_e; \eta_{s_e}, \eta_{s'_e}\Big) = \frac{1}{2\text{Re}\left\{\psi_2^{(\varepsilon)}(y_e)\right\}} \Big[ &\overline{\phi}_\circ(\eta_{s_e})\phi_\circ(\eta_{s'_e})\vartheta(\eta_{s_e} - \eta_{s'_e}) \\
&+ \phi_\circ(\eta_{s_e})\overline{\phi}_\circ(\eta_{s'_e})\vartheta(\eta_{s'_e} - \eta_{s_e}) - \phi_\circ(\eta_{s_e})\phi_\circ(\eta_{s'_e}) \Big],
\end{aligned} \tag{10}$$

where $\psi_2^{(\varepsilon)}$ is the free two-point wavefunction. suitably regularised. The propagator $G_e$ shows three terms, the first two being the retarded and advanced solutions, while the last one is required by the condition that the fluctuations vanish at the boundary.

---

[7]In the literature, the early time divergence is usually taken care of by deforming the contour of the time integration, $-\infty \longrightarrow -\infty(1 - i\epsilon)$. The one at future infinity is instead taken care via a hard cut off $\eta_\circ$ which is then sent to zero at the end of the computation. In the present paper, we take a different route, as the former prescription violates unitarity and the latter can introduce spurious logarithmic divergences. In the first case, we use a deformation in energy space, $E \longrightarrow E - i\epsilon_E$, for each energy involved in the process [75]. In the latter instead, the divergences are regularised via analytic regularisation.

[8]In order to avoid a language clash with the upcoming discussion involving polytopes, we use *site* for indicating the vertex of a graph, and reserve the word *vertex* for the polytopes.

**Cosmological correlators**  The Bunch-Davies wavefunction described above provides the probability distribution of the field configuration $\Phi$ at the boundary via its modulus squared $|\Psi[\Phi]|^2$. Such distribution allows computing the correlation function of any operator $\widehat{\mathcal{O}}[\Phi]$ of such fields. In particular cosmological correlators are defined by taking $\widehat{\mathcal{O}}[\Phi] = \Phi(\vec{p}_1)\cdots\Phi(\vec{p}_n)$:

$$\langle \prod_{j=1}^{n}\Phi(\vec{p}_j)\rangle = \mathcal{N}_c \int \mathcal{D}\Phi \, |\Psi[\Phi]|^2 \prod_{j=1}^{n}\Phi(\vec{p}_j), \qquad \mathcal{N}_c^{-1} = \int \mathcal{D}\Phi \, |\Psi[\Phi]|^2. \tag{11}$$

The perturbative expansion (8) of the Bunch-Davies wavefunction allows and its diagrammatic rules (9), together with (11), allow extracting diagrammatic rules for the correlation functions in terms of wavefunction graphs [2,4]. First, a $n$-point cosmological correlator at $L$-loops can be written as

$$\langle \prod_{j=1}^{n}\Phi(\vec{p}_j)\rangle = \prod_{j=1}^{n}\frac{1}{2\mathrm{Re}\left\{\psi_2^{(\varepsilon)}(E_j)\right\}} \sum_{\mathcal{G}\subset\mathcal{G}_n^{(L)}} \widetilde{\mathcal{C}}_{\mathcal{G}}, \tag{12}$$

where $\mathcal{G}_n^{(L)}$ is the set of graphs at $L$-loops with $n$ external states, and $\mathcal{C}_{\mathcal{G}}$ is computed by summing (twice the real part of) the wavefunction coefficient $\tilde{\psi}_{\mathcal{G}}$ associated to the graph $\mathcal{G}$ with the contribution coming from all the possible ways of the edges and replacing them with the inverse of the real part of two-point wavefunction $\psi_2^{(\varepsilon)}$. The edge deletion operation and replacement by $\psi_2^{(\varepsilon)}$, can be graphically represented as a dash on the relevant edge $\,\rule[0.5ex]{1.5em}{0.4pt}\!\!\!\!\mid\!\!\!\rule[0.5ex]{1.5em}{0.4pt}\,$ : then a cosmological correlator can be obtained by summing over all the possible graph topologies with a fixed number of external states $n$ and at a given loop order $L$ as well as over all the possible ways of dashing the edges of each of these graphs. The function $\widetilde{\mathcal{C}}_{\mathcal{G}}$, which we will refer to simply as *correlator*, can then be written in terms of wavefunction graphs as

$$\widetilde{\mathcal{C}}_{\mathcal{G}} = \sum_{j=0}^{n_e}\sum_{\{\mathcal{G}_j\}}\psi_{\mathcal{G}_j}, \tag{13}$$

where $\mathcal{G}_j$ is the graph $\mathcal{G}$ with $j$ edges deleted,[9] $\{\mathcal{G}_j\}$ is the set whose elements are given by all the possible ways of deleting $j$ edges from $\mathcal{G}$, and $\psi_{\mathcal{G}_j}$ are the wavefunction coefficients associated to $\mathcal{G}_j$ – as $\mathcal{G}_j$ can be either connected or disconnected, $\psi_{\mathcal{G}_j}$ can represent either contributions coming from connected and disconnected graphs.

Note that $\mathcal{C}_{\mathcal{G}}$ has the same singularity structure as $\psi_{\mathcal{G}}$ with the additional singularities when the energies of the internal state vanish.

**The universal integrands**  Let us now consider the general expression (9) for the contribution to the wavefunction associated to a graph $\mathcal{G}$

$$\tilde{\psi}_n^{(L)} = \sum_{\mathcal{G}\subset\mathcal{G}_n^{(L)}}\tilde{\psi}_{\mathcal{G}}, \qquad \tilde{\psi}_{\mathcal{G}} = \int_{-\infty}^{0}\prod_{s\in\mathcal{V}}\left[d\eta_s V_s \phi_\circ^{(s)}\right]\prod_{e\in\mathcal{E}}G_e\left(y_e;\eta_{s_e},\eta_{s'_e}\right), \tag{14}$$

where now the mode functions are considered as appropriately normalised. Using the relation (5) between the states with order parameter $\nu = l + 1/2$ and the conformally coupled one ($l = 0$) and rescaling the bulk-to-bulk propagator via

$$G\left(y_e;\eta_{s_e},\eta_{s'_e}\right) \longrightarrow \left(-y_e\eta_{s_e}\right)^{-l_e}\left(-y_e\eta_{s'_e}\right)^{-l_e}G\left(y_e;\eta_{s_e},\eta_{s'_e}\right), \tag{15}$$

---

[9]Hence $\mathcal{G}_0 = \mathcal{G}$ and $\mathcal{G}_{n_e}$ is the disconnected graph given by the union of all the sites of $\mathcal{G}$.

the wavefunction $\tilde{\psi}_n^{(L)}$ can be written as [65]

$$\tilde{\psi}_n^{(L)} = \prod_{j=1}^{n}\left[\frac{1}{E_j^l}\hat{\mathcal{O}}_l(E_j)\right]\prod_{e\in\mathcal{E}}\left[\frac{1}{y_e^{2l_e}}\right]\tilde{\psi}_{\mathcal{G}}^{\{l_e\}}, \tag{16}$$

with

$$\tilde{\psi}_{\mathcal{G}}^{\{l_e\}} = \int_{-\infty}^{0}\prod_{s\in\mathcal{V}}\left[d\eta_s\frac{\lambda_k(\eta_s)}{(-\eta_s)^{\rho_s}}e^{iX_s\eta_s}\prod_{e\in\mathcal{E}}G\left(y_e;\eta_{s_e},\eta_{s'_e}\right)\right], \tag{17}$$

where the apex $\{l_e\}$ indicates that each edge $e$ has a state with order parameter $l_e$ associate to it, $X_s$ is the sum of the external energies at the site $s$, $\rho_s$ is a function of the order parameters of both external and internal states at the site $s$

$$X_s := \sum_{j=1}^{k_s}E_j, \qquad \rho_s := \sum_{j=1}^{k_s}l_j + \sum_{e\in\mathcal{E}_s}l_e, \tag{18}$$

where $k_s$ is the number of external states at the site $s$, $\mathcal{E}_s\subset\mathcal{E}$ the subset of edges incidents at the site $s$, while $l_j$ and $l_e$ are the integer order parameters associate to the $j$-th external state and to the edge $e$ respectively.

The time-dependent function $\lambda_{k_s}(\eta_s)/(-\eta_s)^{\rho_s}$ can be conveniently written in the following integral representation

$$\frac{\lambda_k(\eta_s)}{(-\eta_s)^{\rho_s}} = \int_{-\infty}^{+\infty}dz_s\,e^{iz_s\eta_s}f(z_s). \tag{19}$$

The time-dependent coupling constant is given by

$$\lambda_k(\eta_s) = \lambda_k\left[a(\eta_s)\right]^{\gamma\left[2-\frac{(k-2)(d-1)}{2}\right]}, \tag{20}$$

and for cosmologies described via the warp factor $a(\eta) = \left(-\ell_\gamma/\eta\right)^\gamma$, (19) becomes

$$\frac{\lambda_k(\eta_s)}{(-\eta_s)^{\rho_s}} = i^{\beta_{k_s,l}}(i\lambda_k)\ell_\gamma^{\gamma\left[2-\frac{(k-2)(d-1)}{2}\right]}\int_{-\infty}^{+\infty}dz_s\,e^{iz_s\eta_s}z_s^{\beta_{k_s,l}-1}\vartheta(z_s), \tag{21}$$

where $\beta_{k_s,l} := \rho_s + \gamma[2-(k-2)(d-1)/2]$. Hence, the wavefunction coefficient $\tilde{\psi}_{\mathcal{G}}^{\{l_e\}}$ associated to a graph $\mathcal{G}$ can be written as

$$\tilde{\psi}_{\mathcal{G}}^{\{l_e\}} = \int_{-\infty}^{+\infty}\prod_{s\in\mathcal{V}}[dx_s f(x_s - X_s)]\psi_{\mathcal{G}}^{\{l_e\}},$$

$$\psi_{\mathcal{G}}^{\{l_e\}} := (i\lambda_k)^{n_s}\int_{-\infty}^{0}\prod_{s\in\mathcal{V}}\left[d\eta_s e^{ix_s\eta_s}\right]\prod_{e\in\mathcal{E}}G\left(y_e;\eta_{s_e},\eta_{s'_e}\right), \tag{22}$$

where $n_s$ is the number of sites of the graph. This representation allows encoding the details of the specific cosmology into the measure of integration $f(x_s - X_s)$ and single out the information which is common to all cosmologies and encoded into $\psi_{\mathcal{G}}^{\{l_e\}}$, which we refer to as *wavefunction universal integrand*. Such an integrand turns out to satisfy a recursion relation involving wavefunctions with lower deformation parameters for the internal states [65]

$$\psi_{\mathcal{G}}^{\{l_e\}} = \sum_{\substack{e,\bar{e}\in\mathcal{E}\\\bar{e}\neq e}}\hat{\mathcal{O}}_e\psi_{\mathcal{G}}^{l_e-1,\{l_{\bar{e}}\}}, \qquad \hat{\mathcal{O}}_e := \frac{2(l_e-1)}{\sum_{s\in\mathcal{V}}x_s}\left(\frac{\partial}{\partial x_{s_e}}+\frac{\partial}{\partial x_{s'_e}}\right)-\frac{\partial^2}{\partial x_{s_e}\partial x_{s'_e}}. \tag{23}$$

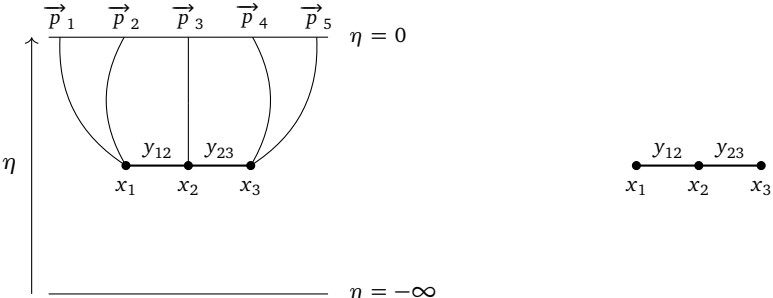

Figure 1: *On the left:* Example of a Feynman graph. The upper horizontal line represents the space-like boundary at $\eta = 0$, while the lines getting to it from the sites of the graph represent the external states. The edges connecting the sites are the propagation of internal states. *On the right*: Example of reduced graph. It is obtained by the previous Feynman graph by suppressing the external states and assigning both site and edge weights to the graphs, which parametrise the kinematic space associated to it [79].

Iterating it, the recursion relation (23) has the wavefunction universal integrands for conformally coupled states as an endpoint, which has a combinatorial description in terms of cosmological polytopes [64] and can be computed via a large class of representations [64, 76]. For $\{l_e = 1, \forall\, e \in \mathcal{E}\}$, the wavefunction universal integrand also enjoys a combinatorial description in terms of a generalisation of the cosmological polytopes [65, 66].

Note that $\psi_{\mathcal{G}}^{\{l_e\}}(x_s, y_e)$ is a function of $\{x_s,\, s \in \mathcal{V}\}$ and $\{y_e\,|\,e \in \mathcal{E}\}$, which parametrise the kinematic space as well as the loop space in case of graphs with loops. It is then possible to associate a weighted reduced graph to $\psi_{\mathcal{G}}^{\{l_e\}}(x_s, y_e)$ by simply suppressing the lines representing the external states and attaching a weight $x_s$ to each site $s \in \mathcal{V}$ and the pair $(y_e, l_e)$ to each edge $e \in \mathcal{E}$. For any value of the set of integer parameters $\{l_e,\, e \in \mathcal{E}\}$, the recursion relation (23) has its seed for $l_e = 0$, which corresponds to the case extensively studied in [64, 75–79]: as the singularity structure of $\psi_{\mathcal{G}}^{\{0\}}$ is determined by the total energies associated to the subgraphs $\{\mathfrak{g} \subseteq \mathcal{G}\}$ [64], the recursion relation (23) makes $\psi_{\mathcal{G}}^{\{l_e\}}$ inherit the very same singularity structure, differing just by the order of the poles. Explicitly, the singularities of the universal integrand $\psi_{\mathcal{G}}^{\{l_e\}}$ are located at the loci $\{E_{\mathfrak{g}} = 0,\, \mathfrak{g} \subseteq \mathcal{G}\}$, where

$$E_{\mathfrak{g}} := \sum_{s \in \mathcal{V}_{\mathfrak{g}}} x_s + \sum_{e \in \mathcal{E}_{\mathfrak{g}}^{\text{ext}}} y_e, \tag{24}$$

with $\mathcal{V}_{\mathfrak{g}}$ and $\mathcal{E}_{\mathfrak{g}}^{\text{ext}}$ being respectively the set of sites in the subgraph $\mathfrak{g}$ and the set of edges departing from it.

As a cosmological correlator can be written in terms of wavefunction coefficients, their integral representation of the latter in terms of a universal wavefunction integrand allows for a similar representation for the former [4].

Finally, let us observe that as the integrand depends just on the variables $\{x_s,\, s \in \mathcal{V}\}$ and $\{y_e,\, e \in \mathcal{E}\}$ associated to sites and edge of the graph, and it is completely agnostic to the number of external states, it is possible to consider a *reduced graph* associated to both integrand and integral, obtained from the original graph by suppressing the lines associated to the external states – see Figure 1. From now on, we will refer to reduced graphs simply as graphs.

**Integrands from combinatorics** All the above field theoretical analysis can be bypassed if we consider a combinatorial first principle formulation for cosmological observables, which is given by (generalised) cosmological polytopes for the Bunch-Davies wavefunction [64, 65],

or by the weighted cosmological polytopes for cosmological correlators [4]. Here we review the salient features of these constructions, with focus on those which are useful in the current work. For a more extensive discussion, see the original literature or the review [2].

In a nutshell, let us consider a graph $\mathcal{G}$ endowed with its sets of site- and edge- weights, $\{x_s, s \in \mathcal{V}\}$ and $\{y_e, e \in \mathcal{E}\}$ respectively. It is possible to associate a linear polynomial $q_{\mathfrak{g}}(\mathcal{Y})$ to each subgraph $\mathfrak{g} \subseteq \mathcal{G}$ defined as the sum over the weights associated to the sites in $\mathfrak{g}$ and the weights of the edges departing from it. Let us consider both the site and edge weights as the local coordinates in projective space $\mathbb{P}^{n_s+n_e-1}$, so that a generic point in it can be written as $\mathcal{Y} := (x, y)$, $x$ and $y$ being a shorthand notation for $x := (x_1, \ldots, x_{n_s})$ and $y := (y_{e_1}, \ldots, y_{e_{n_e}})$ and $n_s$, $n_e$ being the number of sites and edges respectively. The polynomials

$$\left\{ q_{\mathfrak{g}}(\mathcal{Y}) = \sum_{s \in \mathcal{V}_{\mathfrak{g}}} x_s + \sum_{e \in \mathcal{E}_{\mathfrak{g}}^{\text{ext}}} y_e, \, \mathfrak{g} \subseteq \mathcal{G} \right\},$$

can be written as $q_{\mathfrak{g}}(\mathcal{Y}) = \mathcal{Y}^I \mathcal{W}_I^{(\mathfrak{g})}$ where $\mathcal{W}^{(\mathfrak{g})}$ is a vector in the dual space of $\mathbb{P}^{n_s+n_e-1}$, which is still indicated as $\mathbb{P}^{n_s+n_e-1}$ – we will refer to it as a co-vector. Then the inequalities $\{q_{\mathfrak{g}}(\mathcal{Y}) \geq 0\}$ define a cosmological polytope $\mathcal{P}_{\mathcal{G}}$, with the co-vector $\mathcal{W}^{(\mathfrak{g})}$ identifying a hyperplane that intersect the cosmological polytope along its boundary only, containing a facet (*i.e.* a codimension-1 face) $\mathcal{P}_{\mathcal{G}} \cap \mathcal{W}^{(\mathfrak{g})}$. This means that a vertex $\mathcal{Z}^I$ of $\mathcal{P}_{\mathcal{G}}$ on $\mathcal{P}_{\mathcal{G}} \cap \mathcal{W}^{(\mathfrak{g})}$ satisfies the condition $\mathcal{Z}^I \mathcal{W}^{(\mathfrak{g})} = 0$. Given a cosmological polytope $\mathcal{P}_{\mathcal{G}} \subset \mathbb{P}^{n_s+n_e-1}$ there is a unique rational function $\Omega(\mathcal{Y}, \mathcal{P}_{\mathcal{G}})$– up to an overall constant –, named *canonical function* associated to it, such that: *i)* its only singularities are simple poles along the boundaries of $\mathcal{P}_{\mathcal{G}}$; *ii)* its residue of a given pole is a canonical function of a codimension-1 polytope associated to the boundary identified by the pole; *iii*) all its highest codimension singularities have the same normalisation, up to a sign. It turns out that the canonical function $\Omega(\mathcal{Y}, \mathcal{P}_{\mathcal{G}})$ is the wavefunction universal integrand associated to the graph $\mathcal{G}$:

$$\Omega(\mathcal{Y}, \mathcal{P}_{\mathcal{G}}) = \psi_{\mathcal{G}}(x, y). \tag{25}$$

Then the boundaries of $\mathcal{P}_{\mathcal{G}}$ are associated to the residues of $\psi_{\mathcal{G}}(x, y)$. As the boundaries of $\mathcal{P}_{\mathcal{G}}$ are identified by the co-vectors $\{\mathcal{W}^{(\mathfrak{g})}, \mathfrak{g} \subseteq \mathcal{G}\}$, the canonical function $\Omega(\mathcal{Y}, \mathcal{P}_{\mathcal{G}})$ can be generically written as

$$\Omega(\mathcal{Y}, \mathcal{P}_{\mathcal{G}}) = \frac{\mathfrak{n}_{\delta}(\mathcal{Y})}{\prod_{\mathfrak{g} \subseteq \mathcal{G}} q_{\mathfrak{g}}(\mathcal{Y})}, \tag{26}$$

where $\mathfrak{n}_{\delta}(\mathcal{Y})$ is a homogeneous polynomial in $\mathcal{Y}$ of degree $\delta$. From a geometrical perspective, it provides the locus of the intersection of the hyperplanes $\{\mathcal{W}^{(\mathfrak{g})}, \mathfrak{g} \subseteq \mathcal{G}\}$ *outside* of $\mathcal{P}_{\mathcal{G}}$ [76, 79]. Said differently, it is determined by the set of vanishing multiple residues of (26), which determine the compatibility condition from the facets (*i.e.* which intersection among the facets form a higher codimension face), and, from a physics perspective, they determine Steinmann-like relations [76, 79].

There is a generalisation of this construction – the *generalised cosmological polytopes* – which has a rational function associated to it with multiple poles [65, 66]: this rational function has still the form (25) but with multiple poles, and its singularities are still associated to subgraphs.

A further generalisation – the *weighted cosmological polytope* $\mathcal{P}_{\mathcal{G}}^{(w)}$ – has additional boundaries, with the special feature that both the half-planes identifies by the polynomial associated to it, $q_{\mathfrak{g}_e}(\mathcal{Y})$ intersect the geometry [4]. This type of boundary is named *internal boundary* [4, 80]. In the system of local coordinates associated to the weights of the graph, the internal boundaries are given by $\{q_{\mathfrak{g}_e}(\mathcal{Y}) := \mathcal{Y}^I \widetilde{\mathcal{W}}^{(\mathfrak{g}_e)} = y_e, e \in \mathcal{E}\}$. The canonical function

associated to a weighted cosmological polytope $\mathcal{P}_{\mathcal{G}}^{(w)}$ provides the universal integrand for the cosmological correlator associated to the graph $\mathcal{G}$:

$$\Omega\left(\mathcal{Y}, \mathcal{P}_{\mathcal{G}}^{(w)}\right) = \mathcal{C}_{\mathcal{G}}(x, y). \tag{27}$$

We will not go into any further details for any of these constructions. The take-home message is that there are first principle combinatorial formulations to which a rational function is associated to, and such a rational function is related to the field theoretical universal integrand. So we take this combinatorial picture as the definition for the integrands of the integrals we are interested in.

## 3 The general structure of cosmological integrals

A general graph $\mathcal{G}$ contributing to the perturbative observables has associated an integral, whose general form – irrespectively of the topology of $\mathcal{G}$ – is given by

$$\mathcal{I}_{\mathcal{G}}[\alpha, \beta; \mathcal{X}] = \int_{\mathbb{R}_{+}^{\kappa_{\sim}}} \prod_{s \in \mathcal{V}} \left[ \frac{dx_s}{x_s} x_s^{\alpha_s} \right] \int_{\Gamma} \prod_{e \in \mathcal{E}^{(L)}} \left[ \frac{dy_e}{y_e} y_e^{\beta_e} \right] \mu_d(y_e; \mathcal{X}) \frac{\mathfrak{n}_\delta(z, \mathcal{X})}{\prod_{\mathfrak{g} \subseteq \mathcal{G}} \left[ q_{\mathfrak{g}}(z, \mathcal{X}) \right]^{\tau_{\mathfrak{g}}}}, \tag{28}$$

where $z := (x, y)$ is a vector whose entries are given by the integration variables $x := (x_s)_{s \in \mathcal{V}}$ and $y := (y_e)_{e \in \mathcal{E}^{(L)}}$; $\mathcal{X}$ indicates the set of rotational invariants parametrising the kinematic space; $\mathcal{E}^{(L)} \subseteq \mathcal{E}$ is the subset of the edges of the graph $\mathcal{G}$ associated to loops,[10] the integrations over $\{y_e \,|\, e \in \mathcal{E}^{(L)}\}$ parametrise the loop integration in terms of, at most, $n_e^{(L)} := \dim\{\mathcal{E}^{(L)}\}$ edge weights, $\mu_d(y_e)$ is the measure of integration which turns out to be always positive within the domain of integration $\Gamma$ and zero at its boundaries; the set of parameters,

$$\left\{ \sigma := (\alpha, \beta) \in \mathbb{R}^n \,\middle|\, \alpha := (\alpha_s)_{s \in \mathcal{V}} \,; \beta := (\beta_e)_{e \in \mathcal{E}^{(L)}} \,, n := n_s + n_e^{(L)} \right\},$$

depends on the cosmology as well as on the states at the site $s_j$ (for $\alpha$) and on those propagating along the loop edges (for $\beta$); $q_{\mathfrak{g}}(z, \mathcal{X})$ is a linear polynomial associate to the subgraph $\mathcal{G}$

$$q_{\mathfrak{g}}(z, \mathcal{X}) = \sum_{s \in \mathcal{V}_{\mathfrak{g}}} x_s + \sum_{e \in \mathcal{E}_{\mathfrak{g}, L}^{\text{ext}}} y_e + \mathcal{X}_{\mathfrak{g}}, \qquad \mathcal{X}_{\mathfrak{g}} := \sum_{s \in \mathcal{V}_{\mathfrak{g}}} X_s + \sum_{e \in \mathcal{E}_{\mathfrak{g}, 0}^{\text{ext}}} y_e, \tag{29}$$

$\tau_{\mathfrak{g}}$ is generally an integer number associated to the subgraph $\mathfrak{g}$ whose value depends on the states propagating along the edges; $\mathfrak{n}_\delta(z; \mathcal{X})$ is a polynomial of degree $\delta < \tau := \sum_{\mathfrak{g} \subseteq \mathcal{G}} \tau_{\mathfrak{g}}$ – *i.e.* the integrand vanishes as $z^{-(\tau - \delta)}$ as $z \longrightarrow +\infty$ – which is fixed by compatibility conditions among the subgraphs $\mathfrak{g}$ [76].

Importantly, both the site weights $\{x_s \,|\, s \in \mathcal{V}\}$ and the edge weights $\{y_e \,|\, e \in \mathcal{E}^{(L)}\}$, are always positive along the integration contour. Also, the rotational invariants are always positive in the physical region. This implies that each linear polynomial $q_{\mathfrak{g}}$ can vanish just outside the physical region: the integration contour never intersects the codimension-1 hyperplane, where a given $q_{\mathfrak{g}}$ vanishes for arbitrary values of the kinematic data $\{\mathcal{X}_{\mathfrak{g}}, \mathfrak{g} \subseteq \mathcal{G}\}$. Hence, the integral $\mathcal{I}_{\mathcal{G}}$ can diverge as the integration variables become small or large as long as the external kinematics takes values in the physical region. Upon analytic continuation of the external kinematics in such a way that some of the $\mathcal{X}_{\mathfrak{g}}$'s can acquire negative values, the integration contour $\mathbb{R}_{+}^{n_s} \cup \Gamma$ can cross the hyperplanes identified by $\{q_{\mathfrak{g}}(z, \mathcal{X}) = 0, \mathfrak{g} \subseteq \mathcal{G}\}$ and novel singular sheets arise. If one or more of the external data $\sigma$ vanish, then there are singular regions

---

[10]Said differently, there is no integration over the edge weights associate to purely tree subgraph. Such edge weights instead parametrise the angles among external states.

which intersect the contour of integration in its boundary, worsening the infrared behaviour of our integrals. Such regions are the equivalent of the hard collinear regions which characterise scattering processes in flat-space – see [81] and references therein.

As a final comment, we consider the integrand as being the canonical function of a certain cosmological polytope, as described in the previous section. Then, natural regularisation parameters for the integral (28) are given by suitably analytic continuing the parameters $(\alpha, \beta, \tau)$ in the spirit of analytic regularisation [82].[11]

In the next sections, we will be concerned with the study of the infra-red/ultraviolet behaviour of the class of integrals (28), which need to be regularised. Such a regularisation is implemented by analytically continuing the parameter vector $\sigma$ as well as $\{\tau_{\mathfrak{g}}, \mathfrak{g} \subseteq \mathcal{G}\}$ to arbitrary complex values [82] as well as consider the suitable $i\epsilon$-prescription – for an extended discussion, see [75]. The infrared regions for physical kinematics are obtained for large $x_s$'s and/or small $y_e$'s – importantly, as we will see later on, at most $L$ edge variables, associated to different loops, can be taken to be simultaneously small.

Before digging into the infra-red/ultraviolet behaviour of (28), there is one more ingredient that deserves a bit of discussion: the loop integration measure $\mu_d(y, \mathcal{X})$ and the loop integration contour $\Gamma$. They turn out to have a beautiful geometrical interpretation.

## 3.1 The geometry of the loop measure

A cosmological integral of the form (28) is the Mellin transform[12] over the site weights and the integral over the space of loop edge weights of a rational function which is the canonical function of a cosmological polytope [64], generalised cosmological polytope [66] or a weighted cosmological polytope [4], with the numerator $\mathfrak{n}_\delta(z, \mathcal{X})$ being the adjoint surface of the relevant geometry, and the denominators $\{q_{\mathfrak{g}}(z, \mathcal{X}), \mathfrak{g} \subseteq \mathcal{G}\}$ identify their boundaries and are associated to the subgraphs $\{\mathfrak{g}, \mathfrak{g} \subseteq \mathcal{G}\}$. It turns out that the loop measure $\mu_d(y_e, \mathcal{X})$ and the loop contour of integration have an interesting geometrical interpretation.

**The measure at 1-loop** Let us begin with considering $\mathcal{G}$ to be a one-loop graph. For $d \geq n_e^{(1)}$, the standard measure $d^d l$ in loop space can be then written in terms of $\vec{l}^2$ as well as $n_e^{(1)} - 1$ scalar products $(\vec{l} \cdot \vec{q}_j)$, where the vectors $\vec{q}_j$'s are a basis in the space spanned by the $n_e^{(1)} - 1$ linearly independent momenta at each site of the graph. The Jacobian of this change of variable is given in terms of a Gram-determinant in momentum space

$$d^d l = dV_{d-n_e^{(1)}} \prod_{e \in \mathcal{E}^{(1)}} \left[ dy_e^2 \right] \left[ G(\vec{q}_1, \dots, \vec{q}_{n_e^{(1)}-1}) \right]^{-\frac{1}{2}} \left[ \frac{G(\vec{l}, \vec{q}_1, \dots, \vec{q}_{n_e^{(1)}-1})}{G(\vec{q}_1, \dots, \vec{q}_{n_e^{(1)}-1})} \right]^{\frac{d-n_e^{(1)}-1}{2}}, \qquad (30)$$

where $dV_{d-n_e^{(1)}}$ is the volume form for a $(d - n_e^{(1)})$-dimensional sphere, and $G(\vec{v}_1, \dots, \vec{v}_n)$ is the Gram determinant whose $(i, j)$-entry is given by $\vec{v}_i \cdot \vec{v}_j$. The contour integration $\Gamma$ can be expressed as a positivity condition on the Gram determinants

$$\Gamma = \frac{G(\vec{l}, \vec{q}_1, \dots \vec{q}_{n_e^{(1)}-1})}{G(\vec{q}_1, \dots, \vec{q}_{n_e^{(1)}-1})} \geq 0. \qquad (31)$$

---

[11]Indeed this come from the introduction of a regularisation scale. We will be sloppy with it, as we will just focus on the analysis itself of the asymptotic behaviour of the cosmological integrals.

[12]Let us stress that the appearance of an $n_s$-dimensional Mellin transform is just a consequence of the choice of focusing on power-law cosmologies – the measure of integration is related to the choice of cosmologies.

As a basis for the space spanned by the external momenta at the sites of $\mathcal{G}$, let us take

$$\left\{ \vec{q}_j = \vec{P}_{2\ldots j+1} := \sum_{k=2}^{j+1} \vec{p}_k, \ j = 1, \ldots, n_e^{(1)} - 1 \right\}. \tag{32}$$

Interestingly, the integration variables in $\mathcal{I}_{\mathcal{G}}$ parametrise precisely the scalar products among the loop momentum and the external ones, and allow writing each Gram determinant in terms of a Cayley-Menger determinant

$$G\left(\vec{l}, \vec{q}_1, \ldots \vec{q}_{n_e^{(1)}-1}\right) = (-1)^{n_e^{(1)}+1} \mathrm{CM}\left(y_{i,i+1}^2, P_{2\ldots j+1}^2\right), \tag{33}$$

where,

$$\mathrm{CM}\left(y_{i,i+1}^2, P_{2,j+1}^2\right) = \begin{vmatrix} 0 & 1 & 1 & 1 & \cdots & 1 & \cdots & 1 \\ 1 & 0 & y_{12}^2 & y_{23}^2 & \cdots & y_{i,i+1}^2 & \cdots & y_{n_e^{(1)},1}^2 \\ 1 & y_{12}^2 & 0 & P_2^2 & \cdots & P_{2\ldots i}^2 & \cdots & P_{2\ldots n_e^{(1)}}^2 \\ 1 & y_{23}^2 & P_2^2 & 0 & \cdots & P_{3\ldots i}^2 & \cdots & P_{3\ldots n_e^{(1)}}^2 \\ \vdots & \vdots & \vdots & \vdots & \cdots & \vdots & \cdots & \vdots \\ 1 & y_{n_e^{(L)},1}^2 & P_{2\ldots n_e^{(1)}}^2 & P_{3\ldots n_e^{(1)}}^2 & \cdots & P_{i+1\ldots n_e^{(1)}}^2 & \cdots & 0 \end{vmatrix}, \tag{34}$$

with $P_{s_j} = X_{s_j}$ if there is just one external state at the site $s_j$. The Gram determinant is given by $G(\vec{q}_1, \ldots, \vec{q}_{n_e^{(1)}-1}) = (-1)^{n_e^{(1)}} \mathrm{CM}^{(2,2)}$, where the index indicates the $(2,2)$-minor of the Cayley-Menger determinant (34). The condition (31) is just the statement that the space is Euclidean. From the perspective of the Cayley-Menger determinant, the Euclidean space condition is reflected into the matrix (34) being a matrix with non-negative entries which can be associated to Euclidean distances. The determinant (34) is therefore proportional to the squared volume of a $n_e$-dimensional simplex $\Sigma_{n_e^{(1)}}$ whose squared side lengths are given by the non-zero entries, with the condition that the squared volume is positive [83]

$$(-1)^{n_e^{(1)}+1} \mathrm{CM}\left(y_{i,i+1}^2, P_{2\ldots j+1}^2\right) = \left(n_e^{(1)}!\right)^2 \mathrm{Vol}^2\left\{\Sigma_{n_e^{(1)}}\right\} \geq 0. \tag{35}$$

Another way of stating that the space is Euclidean, is that the distance matrix $D$, defined as $D := \left\{ D_{ij}, i, j = 1, j = 1, \ldots, n_e + 1 \,\middle|\, D_{ij} := \sqrt{\mathrm{CM}_{ij}^{(1,1)}} \geq 0 \right\}$, is embeddable in Euclidean space. Let $\mathcal{I}_k$ and $\mathcal{J}_k$ be a set of $n_e^{(1)} - k$ rows and a set of $n_e^{(1)} - k$ columns of CM in (34), then a necessary and sufficient condition for the associated matrix $D$ to be embeddable in Euclidean space is that for all sets $\mathcal{I}_k$ and $\mathcal{J}_k$ and for all $k = 1, \ldots, n_e^{(L)}$ then

$$(-1)^{k+1} \mathrm{CM}^{(\mathcal{I}_k, \mathcal{J}_k)} \geq 0. \tag{36}$$

$\mathrm{CM}^{(\mathcal{I}_k, \mathcal{J}_k)}$ is the minor of CM obtained from the latter erasing the rows in $\mathcal{I}_k$ and the columns in $\mathcal{J}_k$ [83]. Recalling that $G(\vec{q}_1, \ldots \vec{q}_{n_e^{(1)}}) = (-1)^{n_e^{(1)}} \mathrm{CM}^{(2,2)}$, then the embeddability of $D$ in Euclidean space guarantees (31). Said differently, the integration region is given by the non-negativity condition on the volume of a $n_e$-simplex with side lengths given by the non-zero entries of the distance matrix $D$.

The boundary of the integration region is given by the vanishing of the Cayley-Menger determinant, $i.e.$ by the condition that the volume of $\Sigma_{n_e}$ vanishes. This implies that the vertices of $\Sigma_{n_e^{(1)}}$ lie in a proper affine subspace of $\mathbb{R}^{n_e^{(1)}}$ [83]. The Cayley-Menger determinant is in general an irreducible multivariate polynomial, except for $n_e^{(1)} = 2$ [84]. The integral (28)

can then be rewritten as

$$\mathcal{I}_{\mathcal{G}} = \mathfrak{c}_{d,n_e^{(1)}} \int\limits_{\mathbb{R}^{n_s} \cup \Sigma_{n_e^{(1)}}} \left[ \frac{dz}{z} \right] \left[ \frac{\mathrm{Vol}^2\{\Sigma_{n_e^{(1)}}(y, P_{2\dots j})\}}{\mathrm{Vol}^2\{\Sigma_{n_e^{(1)}-1}(P_{2\dots j}))\}} \right]^{\frac{d-n_e^{(1)}-1}{2}} \frac{\mathfrak{n}_\delta(z, \mathcal{X})}{\prod_{\mathfrak{g} \subseteq \mathcal{G}} \left[ q_{\mathfrak{g}}(z, \mathcal{X}) \right]^{\tau_{\mathfrak{g}}}}, \qquad (37)$$

where the overall factor $\mathfrak{c}_{d,n_e^{(1)}}$ is a function of the external data $P_{2\dots j}^2$ ant its explicit expression is given by

$$\mathfrak{c}_{d,n_e^{(1)}} = \mathfrak{c}_{d,n_e^{(1)}}\left( P_{2\dots j}^2 \right) := 2^{n_e^{(1)}} \left( n_e^{(1)} \right)^{d-n_e^{(1)}-2} \left( n_e^{(1)}! \right) \frac{\pi^{\frac{d-n_e^{(1)}}{2}}}{\Gamma\left( \frac{d-n_e^{(1)}+1}{2} \right)} \mathrm{Vol}\left\{ \Sigma_{n_e^{(1)}-1}(P_{2\dots j}) \right\}^{-1}. \qquad (38)$$

A comment is now in order. The loop-space measure $\mu_d(y_e)$ is a function of the edge weights, depends on the graph $\mathcal{G}$. However, irrespectively of the graph, they are encoding information on the same loop space $d^d l$. How are they connected to each other? Indeed, given a one-loop graph $\mathcal{G}$, we can parametrise the loop space introducing additional edge variables which could be associated to a larger polygon. Nevertheless, the integrand would not depend on them. Hence, if $\mu_d(y_e, p; n_s)$ is the integration measure directly associated to $\mathcal{G}$[13] and $\mu_d(y_{\tilde{e}}, \tilde{p}; \tilde{n}_s)$ is the one associate to a polygon $\tilde{\mathcal{G}}$ with $\tilde{n}_s > n_s$ sites, with $y_{\tilde{e}}$ being the weights of the edges $\tilde{\mathcal{E}}^{(1)}$ in $\tilde{\mathcal{G}}$, then the two are related via

$$\mu_d(y_e, p; n_s) = \int_{\tilde{\Gamma}=0} \prod_{\tilde{e} \in \mathcal{E}^{(1)} \setminus \tilde{\mathcal{E}}^{(1)}} \left[ dy_{\tilde{e}}^2 \right] \mu_d(y_{\tilde{e}}, \tilde{p}; \tilde{n}_s), \qquad (39)$$

where the integral is performed along the boundaries $\tilde{\Gamma} = 0$ of the integration contour $\tilde{\Gamma}$ associated to the parametrisation of the loop space via $\{y_{\tilde{e}} \,|\, \tilde{e} \in \tilde{\mathcal{E}}^{(1)}\}$. From a geometrical point of view, the left-hand-side of (39) is related to the volume of an $n_e^{(1)}$-dimensional simplex $\Sigma_{n_e^{(1)}}$, while the integrand in the right-hand-side is related to the volume of an $\tilde{n}_e^{(1)}$-dimensional one $\Sigma_{\tilde{n}_e^{(1)}}$. The integration projects $\tilde{n}_e^{(1)} - n_e^{(1)}$ vertices of $\Sigma_{\tilde{n}_e^{(1)}}$ on an affine subspace spanned by the vertices of $\Sigma_{n_e^{(1)}}$.

**The measure at $L$-loops** Let us now move to the measure for the multi-loop integrals. Proceeding as before, for each loop momentum $\{l_l \,|\, l = 1, \dots L\}$ we can write:

$$d^d l_l = dV_{d-n_{e_l}-L+l} \prod_{e \in \mathcal{E}_l^{(L)}} \left[ dy_e^2 \right] \left[ \mathrm{CM}(Q_{i\dots j}^2) \right]^{-\frac{1}{2}} \frac{\left[ \mathrm{CM}\left( y_e^2, Q_{2\dots j}^2 \right) \right]^{\frac{d-n_s-1-L+l}{2}}}{\left[ \mathrm{CM}(Q_{i\dots j}^2) \right]^{\frac{d-n_{s_l}-L+l}{2}}}, \qquad (40)$$

where the $Q$'s appearing in the Cayley-Menger determinants refer to the moduli of the momenta external to the $l$-th loop, *i.e.* some of them can be related to actual external momenta, $Q_{2\dots j} = P_{2\dots j}$, while others can depend on the edge weights associated with other loops (or, which is the same, on momenta running in the other loops). Thus, a $L$-loop integral $\mathcal{I}_{\mathcal{G}}$ associated to a graph $\mathcal{G}$ acquires the form

$$\mathcal{I}_{\mathcal{G}} = \prod_{l=1}^{L} \left[ \frac{\pi^{\frac{d-n_{s_l}-L+l}{2}}}{\Gamma\left( \frac{d-n_{s_l}-L+l}{2} \right)} \int_{\Gamma_l} \prod_{e_l \in \mathcal{E}_l^{(1)}} \left[ dy_{e_l}^2 \right] \frac{\left[ \mathrm{CM}\left( y_{e_l}^2, Q_{i\dots j}^2(y_{\ell_l}) \right) \right]^{\frac{d-n_{s_l}-1-L+l}{2}}}{\left[ \mathrm{CM}\left( Q_{i\dots j}^2(y_{\ell_l}) \right) \right]^{\frac{d-n_{s_l}-L+l}{2}}} \right] \frac{\mathfrak{n}_\delta(y_e)}{\prod_{\mathfrak{g} \subseteq \mathcal{G}} q_{\mathfrak{g}}(y_e)}, \qquad (41)$$

---

[13]Said differently, the measure $\mu_d(y_e, p; n_s)$ is just a function of the edge weights of $\mathcal{G}$.

where $y_{\not{e}_l}$ is associated to an edge which is not in $\mathcal{E}_l$, and the contour $\Gamma_l$ for the $l$-th loop is given by

$$\Gamma_l := \left\{ \frac{\mathrm{CM}\left(y_{e_l}^2, Q_{i\ldots j}^2(y_{\not{e}_l})\right)}{\mathrm{CM}\left(Q_{i\ldots j}^2(y_{\not{e}_l})\right)} \leq 0 \right\}. \tag{42}$$

Such an expression can be further manipulated to get

$$\mathcal{I}_{\mathcal{G}} = \mathfrak{c}_{d,n_e^{(L)},L} \int\limits_{\mathbb{R}_+^{n_s}\cup\Gamma} \left[ \frac{dz}{z} z^{\sigma-1} \right] \left[ \frac{\mathrm{Vol}^2\left(y_e, P_{i\ldots j}^2\right)}{\mathrm{Vol}^2\left(P_{i\ldots j}^2\right)} \right]^{\frac{d-n_s-L}{2}} \frac{\mathfrak{n}_\delta(z,\mathcal{X})}{\prod\limits_{\mathfrak{g}\subseteq\mathcal{G}} q_{\mathfrak{g}}(z,\mathcal{X})}, \tag{43}$$

where the integration region $\Gamma$ is determined by the intersection among all the contours $\Gamma_l$.

$$\Gamma = \bigcap_{l=1}^{L} \Gamma_l, \tag{44}$$

and the coefficient $\mathfrak{c}_{d,n_e,L}$ is given by

$$\mathfrak{c}_{d,n_e^{(L)},L} := \frac{\pi^{\frac{d-n_s+L(L-1)/2}{2}}}{\prod\limits_{l=1}^{L} \Gamma\left(\frac{d-n_{s_l}-L+l}{2}\right)} 2^{n_e^{(L)}} \left(n_e^{(L)}\right)^{d-n_e-L-2} \left(n_e^{(L)}!\right) \mathrm{Vol}\left\{\tilde{\Sigma}\left(P_{i\ldots j}\right)\right\}^{-1}. \tag{45}$$

As a final remark, the representation of the measure of integration in terms of the Cayley-Menger determinant allows for a geometrical interpretation in terms of volumes of a simplex in $\mathbb{P}^{n_e}$, with all the edge weights and the moduli $\{p_s, s\in\mathcal{V}\}$ of the momenta of the external states[14] at each site $s$ associated to the edges of the simplex, and the boundaries of the region of integration determined by projecting its vertices on a lower-dimensional hyperplane.

## 4 The asymptotic structure of cosmological integrals

It is useful to summarise the salient features of the general cosmological integral (28) associated to a given graph $\mathcal{G}$:

❏ its integrand $\Omega_{\mathcal{G}}(x,y)$ is a rational function whose denominator is a degree-$\tilde{v}$ factorisable polynomial, whose factors are $\tilde{v}$ linear polynomials which are in $1-1$ correspondence with subgraphs of $\mathcal{G}$ and individually identifies a facet of the relevant polytope;

❏ the numerator of $\Omega_{\mathcal{G}}(x,y)$ is a polynomial of degree $\tilde{v}-n_s-n_e$ that identifies the adjoint surface of the relevant polytope, and it's fixed by compatibility conditions among singularities [76, 79] and, in the case of a correlation function, by additional conditions on the residues [4];

❏ the integral (28) can be seen as a Mellin transform of $\Omega_{\mathcal{G}}(x,y)$ over the site weights, and over $\prod\limits_{l=1}^{L} \min\{n_e^{(l)}, d\}$-dimensional edge-weight space whose measure can be expressed in terms of squared volume of simplices;

---

[14]Importantly, if at a site $s$ more than one external state is attached, then $P_s$ parameterise the angle among them. If instead just a single external state is attached to it, then $P_s = X_s$.

❏ the integration measure in the edge-weight space is always positive in the interior of the integration region, and vanishes just on its boundary.

Because of all these features, the integral (28) can show divergences when the graph weights become large or small. Such a behaviour is controlled by the powers in the site and edge weights in the integrand $\Omega_{\mathcal{G}}(x, y)$ and in the weight integration measure. It is in turn codified in the combinatorics of the *Newton polytope* associated to it, which controls the convergence of the integral [85, 86], allow identifying the divergences and isolate them via sector decomposition [68–74].

**Newton polytopes and asymptotic behaviour**  In order to fix the ideas, let us consider the following toy integral

$$\mathcal{I}[\sigma] := \int_{\mathbb{R}_+^n} \left[ \frac{dz}{z} z^\sigma \right] \frac{\mathfrak{n}_\delta(z)}{[p_m(z)]^\tau} , \tag{46}$$

where $z := (z_1 \dots z_n) \in \mathbb{R}_+^n$, $\sigma := (\sigma_1, \dots, \sigma_n) \in \mathbb{C}^n$, while $p_m(z)$ and $\mathfrak{n}_\delta(z)$ are multivariate polynomials in $z$ of degrees $m \in \mathbb{Z}_+$ and $\delta \in \mathbb{Z}_+$ respectively. Such an integral falls into the class of integrals $\mathcal{I}_{\mathcal{G}}$ given in (28). Let us begin with the case for which the numerator is a degree-zero polynomial. The integral (46) can diverge in the regions identified by the zeroes of polynomial $p_m(z)$ in $\mathbb{R}_+^n$, when $z$ becomes large or small along some direction in $\mathbb{R}_+^n$. Hence, when the integration region does not intersect with the locus of the zeroes of $p_m(z)$, then the integral (46) converges and returns an analytic function of $\sigma \in \mathbb{C}^n$ in a tube domain, *i.e.* a region of $\mathbb{C}^n$ whose real part is bounded, and the imaginary part is unconstrained. As in this case the singularities can come just from the $z$ becoming large or small along some real direction, such a behaviour turns out to be captured by the powers of the polynomial $p_m(z)$, with the Mellin parameters just shifting such a behaviour. Hence, given the polynomial $p_m(z)$, it is possible to consider the space $\rho \in \mathbb{R}^n$ of powers of $z$: the powers in the monomials of $p_m(z)$ determine a collection of points in such a space whose convex hull defines the *Newton polytope* $\mathcal{N}[p_m(z)]$ of $p_m(z)$ with its facets encoding the possible divergent directions through the co-vectors which identify them. The tube domain where the integral (46) converges and defines an analytic function in $\sigma$ is therefore given by the requirement that $\mathrm{Re}\{\sigma\}$ lies in the interior of the Newton polytope $\mathcal{N}[p_m(z)]$ [85]. More formally, let us consider the multivariate polynomial $p_m(z)$ written as

$$p_m(z) := \sum_{\substack{\rho_1 \dots \rho_n = 0 \\ \rho_1 + \dots + \rho_n \le m}}^{m} a_{\rho_1 \cdots \rho_n} \prod_{j=1}^{n} z_j^{\rho_j} \equiv \sum_{\substack{\rho \in \mathbb{Z}^n \\ \rho_1 + \dots + \rho_n \le m}} a_\rho z^\rho , \tag{47}$$

where $z := (z_1, \dots, z_n)$, $\rho := (\rho_1, \dots, \rho_n)$, and $a_{\rho_1 \cdots \rho_n} \in \mathbb{C}^n$. It is possible to associate a collection of vertices $\left\{ \mathfrak{z} := (1, \mathrm{Re}\{\rho\}) \in \mathbb{P}^n \,\middle|\, \mathrm{Re}\{\rho\} \in \mathbb{Z}^n, \sum_{j=1}^{n} \rho_j \le m \right\}$ in $\mathbb{P}^n$. Then, the convex hull of the elements of this collection defines the Newton polytope $\mathcal{N}[p_m(z)]$ associated to $p_m(z)$. If the polynomial $p_m(z)$ is factorisable, *i.e.* $p_m(z) = p_{m_1}(z) \cdots p_{m_t}(z)$ with $m_1 + \dots + m_t = m$, then the Newton polytope $\mathcal{N}[p_m(z)]$ is given by the *Minkowski sum* of the Newton polytopes associated to each $\{p_{m_j}(z), j = 1, \dots, t, m_1 + \dots m_t = m\}$:

$$\mathcal{N}[p_m(z)] = \bigoplus_{j=1}^{t} \mathcal{N}\left[p_{m_j}(z)\right] , \tag{48}$$

with $\bigoplus$ indicating the Minkowski sum, which for polytopes is given by the convex hull of the Minkowski sum of their vertices, which in turn, is defined as the set of vectors resultant from

the sum of the vertices of all the distinct polytopes. Finally, if the polynomial factorises as $p_m(z) := \left[p_{m_1}(z)\right]^{\tau_1} \cdots \left[p_{m_t}(z)\right]^{\tau_{m_t}}$, with $\tau_1 m_1 + \ldots + \tau_{m_t} m_t = m$ and $\{\tau_j \in \mathbb{C} \mid j = 1, \ldots, t\}$, then its Newton polytope can be written as the *weighted Minkowski sum* of the individual polynomials $\left\{p_{m_j}(z), j = 1, \ldots, t\right\}$ with the weights given by $\{\mathrm{Re}\{\tau_j\} \geq 0, j = 1, \ldots t\}$:

$$\mathcal{N}[p_m(z)] = \bigoplus_{j=1}^{t} \mathrm{Re}\{\tau_j\} \mathcal{N}\left[p_{m_j}(z)\right]. \tag{49}$$

Interestingly, the combinatorial structure of (49) does not depend on the weights.

Note that the monomial $z^\sigma$ appearing in the measure of the Mellin transform in (46) can be included in the definition of Newton polytope:

$$\mathcal{N}[\mathcal{I}(\sigma)] := -\sigma \bigoplus_{j=1}^{t} \mathrm{Re}\{\tau_j\} \mathcal{N}\left[p_{m_j}(z)\right]. \tag{50}$$

As it just identifies a point in the space of powers, its only effect would be a shift of the polytope $\mathcal{N}[p_m(z)]$ that depends on $\mathrm{Re}\{\sigma\}$. Thus, the combinatorial structure of $\mathcal{N}[\mathcal{I}(\sigma)]$ is the same as $\mathcal{N}[p_m(z)]$.

What happens if the numerator is a polynomial with degree higher than zero?. The convergence of the integral (46) can be conveniently studied by considering the numerator $\mathfrak{n}_\delta(z)$ expanded in monomials and mapping (46) into a sum of integrals whose integrands share the very same denominator $[p_m(z)]^\tau$, have all numerator equal to one and differ just by their Mellin transform parameters which are shifted differently and with such a shift depending on the powers in the monomials in $\mathfrak{n}_\delta(x, y)$:

$$\mathcal{I}[\sigma] = \sum_{\substack{r=0 \\ r_1 + \ldots + r_n \leq \delta}} \mathfrak{a}_r^{(\mathfrak{n})} \mathcal{I}[\sigma + r] = \sum_{\substack{r=0 \\ r_1 + \ldots + r_n \leq \delta}} \mathfrak{a}_r^{(\mathfrak{n})} \int_0^{+\infty} \left[\frac{dz}{z} z^{\sigma+r}\right] \frac{1}{[p_m(z)]^\tau}. \tag{51}$$

The asymptotic structure of the full integral is still governed by the Newton polytope associated to the denominator, which is shared by all the integrals in the sum (51) but it is shifted differently for each integral. Each individual integral (51) then turns out to be convergent if $(1, \mathrm{Re}\{\sigma\} + r)$ lie in the interior of the relevant Newton polytope, and the full integral (46) converges in the overlap region of these Newton polytopes [85].

The condition on the individual integral can be equivalently checked by considering the facets of the associated Newton polytope and the co-vectors in the dual space that identify them. Note that such a collection $\{\mathfrak{W}_I^{(j)}\}$ of co-vectors in the dual space[15] is given by

$$\mathfrak{W}_I^{(j)} = (-1)^{j(n-1)} \epsilon_{IK_1 \ldots K_n} \mathfrak{Z}_{a_{j+1}}^{K_1} \cdots \mathfrak{Z}_{a_{j+n1}}^{K_n},$$

with $\epsilon_{IK_1 \ldots K_n}$ being the totally anti-symmetric $(n+1)$-dimensional Levi-Civita symbol. As the vectors identifying the vertices are of the form $\mathfrak{Z} = (1, \rho)$ ($\rho \in \mathbb{Z}^n$), the co-vectors $\mathfrak{W}$ turn out to have the structure $\mathfrak{W} = (\lambda, \omega)$, where $\lambda = \lambda(\mathrm{Re}\{\sigma\}, \mathrm{Re}\{\tau\})$ depends on the parameters $\mathrm{Re}\{\sigma\}$ and $\mathrm{Re}\{\tau\}$, while $\omega$ is a $n$-vector that does not have such a dependence. The function $\lambda$ can be explicitly written as

$$\lambda(\sigma, \tau; \omega) = \mathrm{Re}\{\sigma\} \cdot \omega - \mathrm{Re}\{\tau\} \sum_{\{\rho\}} \max\{\rho \cdot \omega\}, \tag{52}$$

---

[15]The dual space will be still indicated with $\mathbb{P}^n$.

and provides how the direction pointed by $\omega$ is approached.[16] Furthermore, the knowledge of the co-vectors allows to tile the region of integration into sectors $\Delta_{\mathfrak{W}_c}$ [68–74], each of which is identified by the collection of contiguous co-vectors $\mathfrak{W}_c$,

$$\mathbb{R}^n_+ = \bigcup_{\{\mathfrak{W}_c\}} \Delta_{\mathfrak{W}_c}, \tag{53}$$

and the integral (46), with a degree $\delta = 0$ numerator, can then be written as

$$I[\sigma] = \sum_{\{\mathfrak{W}_c\}} \int_{\Delta_{\mathfrak{W}_c}} \left[ \frac{dz}{z} z^\sigma \right] \frac{\mathfrak{n}_\delta(z)}{[p_m(z)]^\tau} \equiv \sum_{\{\mathfrak{W}_c\}} I_{\Delta_{\mathfrak{W}_c}}[\sigma]. \tag{54}$$

The knowledge of the co-vectors identifying a given sector $\Delta_{\mathfrak{W}_c}$ also provides a change of variables to suitably parametrise the integral $I_{\Delta_{\mathfrak{W}_c}}[\sigma]$

$$z_j \longrightarrow \prod_{\omega \in \omega_c} \zeta_\omega^{-\mathfrak{e}_j \cdot \omega}, \qquad \zeta_\omega \in [0, 1], \quad \forall \, \omega \in \omega_c, \tag{55}$$

$\forall \, j \in [1, n]$, where $\{\mathfrak{e}_j \in \mathbb{R}^n, j = 1, \ldots, n\}$ is the canonical basis in $\mathbb{R}^n$ and $\omega_c$ is the set of $n$-vectors in $\mathfrak{W}$ which belongs to the collection of compatible facets identified by $\mathfrak{W}_c$. With such a change of variables, the integral $I_{\Delta_{\mathfrak{W}_c}}[\sigma]$ acquires the form

$$I_{\Delta_{\mathfrak{W}_c}} = \int_0^1 \left[ \frac{d\zeta}{\zeta} \zeta^{-\lambda} \right] \frac{1}{[p_m(\zeta)]^\tau}, \tag{56}$$

where $\zeta := (\zeta_\omega)_{\omega \in \omega_c}$ and $\lambda := (\lambda_\omega)_{\omega \in \omega_c}$. In this way, the divergences can be isolated and Laurent expanding one of the $\lambda_\omega$ receives contribution from just a subset of integrals – see Appendix B for a simple example. Interestingly, it was shown in [74] that in case the full $\lambda$ is taken to zero, *e.g.* by rescaling it via a smallness parameter $\epsilon$ ($\lambda \longrightarrow \epsilon\lambda$), then as $\epsilon \longrightarrow 0$, then all sectors would contribute, and the leading divergence coefficient is the canonical function of the Newton polytope associated to $p_m(z)$,

With this general information at hand about how the asymptotic structure of the integrals are encoded into the combinatorics of the Newton polytopes, the following subsection will deal with the explicit analysis of the cosmological integrals. Despite the general idea is not affected by the topology of the graphs, we find convenient to discuss tree and loop graphs separately.

## 4.1 The asymptotic structure of tree-level graphs

Let us begin a detailed analysis starting with integrals associated to a tree-level graph $\mathcal{G}$: In this case the integrations are associated to the site weights of $\mathcal{G}$ only, from (28), it acquires the generic form

$$I_\mathcal{G}^{(0)}[\alpha, \tau, \mathcal{X}] = \int_0^{+\infty} \prod_{s \in \mathcal{V}} \left[ \frac{dx_s}{x_s} x_s^{\alpha_s} \right] \frac{\mathfrak{n}_\delta(x, \mathcal{X})}{\prod_{\mathfrak{g} \subseteq \mathcal{G}} [q_\mathfrak{g}(x, \mathcal{X})]^{\tau_\mathfrak{g}}} \equiv \int_{\mathbb{R}^n_+} \left[ \frac{dz}{z} z^\alpha \right] \frac{\mathfrak{n}_\delta(z, \mathcal{X})}{\prod_{\mathfrak{g} \subseteq \mathcal{G}} [q_\mathfrak{g}(z, \mathcal{X})]^{\tau_\mathfrak{g}}}, \tag{57}$$

where $x := (x_s)_{s \in \mathcal{V}}$, $\alpha := (\alpha_s)_{s \in \mathcal{V}}$, while $\mathfrak{n}_\delta(x)$ and $\{q_\mathfrak{g}(x), \forall \, \mathfrak{g} \subseteq \mathcal{G}\}$ are inhomogeneous polynomials in $x$ of degree $\delta = \tilde{v} - n_s - n_e$ and 1 respectively. It is useful to recall the explicit expression for the linear polynomial $q_\mathfrak{g}$

$$q_\mathfrak{g}(x) := \sum_{s \in \mathcal{V}_\mathfrak{g}} x_s + \mathcal{X}_\mathfrak{g}, \tag{58}$$

---

[16]In the context of Feynman integrals for which the tools of tropical geometry are used, the function $\lambda(\sigma, \tau_o; \omega)$ is referred to as *tropical function*, and it is indicated as $Trop$ – see [74, 87–91].

with $\mathcal{V}_{\mathfrak{g}}$ being the set of sites in $\mathfrak{g}$, and $\mathcal{X}_{\mathfrak{g}}$ parametrising the kinematic invariant associated to the $\mathfrak{g}$. As discussed above, the asymptotic behaviour of the integral (57) is encoded into the Newton polytope associated to the denominator factors $\{q_{\mathfrak{g}}(x), \mathfrak{g} \subseteq \mathcal{G}\}$. For a generic tree graph with $n_s$ sites, it lives in $\mathbb{P}^{n_s}$. As discussed earlier, the convergence of the integral (57) can be analysed by considering the weighted Minkowski sum of the Newton polytopes $\{\mathcal{N}[\mathfrak{g}], \mathfrak{g} \subseteq \mathcal{G}\}$ associated to the individual factors $\{q_{\mathfrak{g}}(x), \mathfrak{g} \subseteq \mathcal{G}\}$, shifted by the vector $\mathfrak{Z}_{\alpha,r} := (1, \operatorname{Re}\{\alpha\} + r)$ associated to the Mellin transform and to the monomials of the numerator $\mathfrak{n}_{\delta}(x)$ and taking the overlap among them – we will come back to this shift later:

$$\mathcal{N}_{\mathcal{G}}^{(\mathfrak{Z}_{\alpha,r})} = -\mathfrak{Z}_{\alpha,r} \oplus \mathcal{N}_{\mathcal{G}}, \qquad \mathcal{N}_{\mathcal{G}} := \bigoplus_{\mathfrak{g} \subseteq \mathcal{G}} \operatorname{Re}\{\tau_{\mathfrak{g}}\} \mathcal{N}[\mathfrak{g}], \tag{59}$$

where $\operatorname{Re}\{\tau_{\mathfrak{g}}\} > 0$. As emphasised earlier, nor the vector $\mathfrak{Z}_{\alpha,r}$ neither the positive weights $\operatorname{Re}\{\tau_{\mathfrak{g}}\}$ affect the combinatorial structure of $\mathcal{N}_{\mathcal{G}}^{(\mathfrak{Z}_{\alpha,r})}$, however they determine the asymptotic behaviour. Interestingly, we can think about the set of connected subgraphs $\{\mathfrak{g}, \mathfrak{g} \subseteq \mathcal{G}\}$ as the union of the sets $\mathfrak{G}(n_s^{(\mathfrak{g})})$, whose elements are all the connected subgraphs $\mathfrak{g} \subseteq \mathcal{G}$ with $n_s^{(\mathfrak{g})}$ sites

$$\{\mathfrak{g}, \mathfrak{g} \subseteq \mathcal{G}\} = \bigcup_{n_s^{(\mathfrak{g})}=1}^{n_s} \mathfrak{G}(n_s^{(\mathfrak{g})}), \tag{60}$$

with $\mathfrak{G}(n_s) = \{\mathcal{G}\}$, *i.e.* it contains a single element which is the whole graph $\mathcal{G}$, $\mathfrak{G}(2)$ containing $n_e = n_s - 1$[17] elements given by all the 2-site subgraphs constituting $\mathcal{G}$, and $\mathfrak{G}(1) = \{\mathfrak{g}_j, j = 1, \ldots n_s\}$ containing the $n_s$ graphs constituted by a single site. Furthermore, as $q_{\mathfrak{g}}(x)$ is a linear polynomial dependent on all the site weights $\{x_s, s \in \mathcal{V}_{\mathfrak{g}}\}$, the associated Newton polytope $\mathcal{N}[\mathfrak{g}]$ is a simplex $\Sigma[\mathfrak{g}]$ in $\mathbb{P}^{n_s^{(\mathfrak{g})}} \subseteq \mathbb{P}^{n_s}$. The Minkowski sum in (60) becomes a sum over simplices in $\mathbb{P}^{n_s^{(\mathfrak{g})}}$ ($n_s^{(\mathfrak{g})} = 1, \ldots, n_s$) embedded into $\mathbb{P}^{n_s}$:

$$\mathcal{N}_{\mathcal{G}} := \bigoplus_{\mathfrak{g} \subseteq \mathcal{G}} \operatorname{Re}\{\tau_{\mathfrak{g}}\} \Sigma[\mathfrak{g}]. \tag{61}$$

Because of (61), these polytopes constitute a special class of *nestohedrons*, with the latter being introduced in [67, 92]. The combinatorial structure of (61) does not depend on the choice of the weights $\{\operatorname{Re}\{\tau_{\mathfrak{g}}\}, \mathfrak{g} \subseteq \mathcal{G}\}$ [92]. These polytopes are constructed via (61) from a single top-dimensional simplex, $\mathcal{N}[\mathcal{G}]$, with the lower dimensional simplices being a subset of the lower dimensional faces of $\mathcal{N}[\mathcal{G}]$ and their number in codimension $n_s - n_s^{(\mathfrak{g})}$ being given by the dimension of the set $\mathfrak{G}(n_s^{(\mathfrak{g})})$.

Let us consider a simplex $\Sigma_{\mathfrak{g}} \in \mathbb{P}^{n_s^{(\mathfrak{g})}}$ associated to the subgraphs $\mathfrak{g}$, its facets can be associated to all the single tubings corresponding to its (not necessarily connected) subgraphs in a similar fashion as in [93, 94] – see Figure 2.

Let $\mathfrak{g} = \mathcal{G}$ and let us consider the associated simplex $\Sigma_{\mathcal{G}}$. Its facets are identified by the tubing corresponding to the graph itself, as well as all the other tubings which include $n_s - 1$ sites. Let $\not{s}$ be the site not included in a given tubing. This implies that *all* the vertices on this facet of $\Sigma_{\mathcal{G}}$ are zero along their $\not{s}$ component, and consequently they are on the codimension-1 hyperplane identified by the co-vector $\mathfrak{W}^{(\not{s})}$ with all zeroes but the $\not{s}$-component where it is $-1$.[18] The facet associated to the tubing that includes all the sites is then a vector whose components are all equal to 1. The other simplices appearing in the Minkowski sum (61) are all the codimension-$k$ faces of $\Sigma_{\mathcal{G}}$ $k \in [1, n_e - 1]$ identified by a connected tubing respect to the one associated to the faces of one codimension higher. The

---

[17]This relation between number of edges and sites of a graph holds at for tree graphs only.

[18]The minus sign is just a consequence of the convention that the co-vectors $\mathfrak{W}$ that identify the facets are directed outwards with respect to the polytope. It is possible to equivalently choose the opposite convention. In this case, the expression (52) would change as well, with the minimum replacing the maximum.

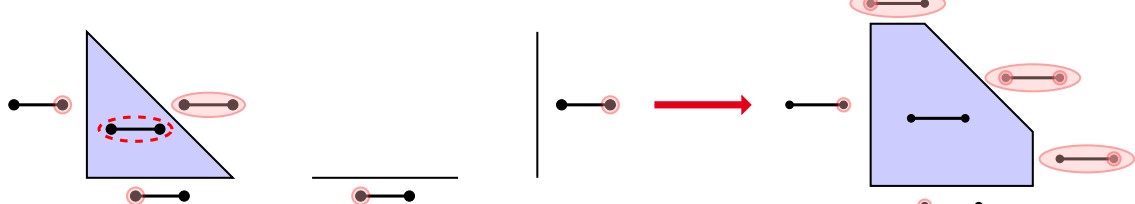

Figure 2: Newton polytope associated to a two-site graph $\mathcal{G}$. It is constructed as a Minkowski sum of the Newton polytopes associated to its subgraphs $\mathfrak{g}$, *i.e.* a triangle and two segments respectively for $\mathfrak{g} = \mathcal{G}$ and the two subgraphs containing a single site only– see the triple of pictures on the left. The final Newton polytope can be realised as a truncation of the triangle based on the underlying graph and the tubings corresponding to the single-site graphs, and hence its facets are associated to tubings of the graph.

Newton polytope associated to the full graph $\mathcal{G}$ can be then obtained iteratively truncating the top dimensional simplex $\Sigma_{\mathcal{G}}$ via the ones of lower codimension associated to connected subgraphs only, with the new facets associated to the overlap among the tubings related to the simplices contributing to it – see Figures 2 and 3. Note that this procedure keeps the original facets of $\Sigma_{\mathcal{G}}$, adding new ones. The formers are identified by the very same co-vectors $\left\{ \mathfrak{W}'^{(j)} := \left(0, -\mathfrak{e}_j\right)^{\mathrm{T}}, j = 1, \dots n_s \right\} \cup \left\{ \mathfrak{W}^{(1_1 \dots 1_{n_s})} := \left(\lambda^{(1_1 \dots 1_{n_s})}, \mathfrak{e}_{1 \dots n_s}\right)^{\mathrm{T}} \right\}$ – where $\{\mathfrak{e}_j \in \mathbb{R}^{n_s}, j = 1, \dots, n_s\}$ is the canonical basis for $\mathbb{R}^n$ and $\mathfrak{e}_{1 \dots n_s} \in \mathbb{R}^{n_s}$ is a vector with all entries equal to 1. As the new facets are obtained by truncating the top-dimensional simplex $\Sigma_{\mathcal{G}}$ via its facets which correspond to codimension-1 allowed tubings, they are identified by co-vectors of the form:

$$\mathfrak{W}^{(j_1 \dots j_{n_s^{(\mathfrak{g})}})} = \begin{pmatrix} \lambda^{(j_1 \dots j_{n_s^{(\mathfrak{g})}})} \\ \mathfrak{e}_{j_1 \dots j_{n_s^{(\mathfrak{g})}}} \end{pmatrix}, \tag{62}$$

where $\left\{ \mathfrak{e}_{j_1 \dots j_{n_s^{(\mathfrak{g})}}}, j_k \in [k, n_s - n_s^{(\mathfrak{g})} + k], k \in [1, n_s^{(\mathfrak{g})}] \right\}$ are such that the $j_k$-th entries, which correspond to the number of single site tubings, are 1 while all the others are zero. From the graph perspective, they are associated to the overlap between tubings and, for fixed $n_s^{(\mathfrak{g})}$ they are identified by those tubing overlaps containing $n_s^{(\mathfrak{g})}$ single-site tubings – see Figures 2 and 3. Furthermore, the function $\lambda^{(j_1 \dots j_{n_s^{(\mathfrak{g})}})}$ is given by (minus) the number of overlapping tubings corresponding to the facet identified by $\mathfrak{W}^{(j_1 \dots j_{n_s^{(\mathfrak{g})}})}$. Such a realisation makes explicit that the number of facets of $\mathcal{N}_{\mathcal{G}}$ can be counted as

$$\tilde{\nu}\left[\mathcal{N}_{\mathcal{G}}\right] = 2n_s + \sum_{n_s^{(\mathfrak{g})}=2}^{n_s} \binom{n_s}{n_s^{(\mathfrak{g})}} = 2^{n_s} + n_s - 1, \tag{63}$$

with the first term providing the dimension of the collection $\{\mathfrak{W}^{(j)}, \mathfrak{W}'^{(j)} \in \mathbb{P}^{n_s}, j = 1, \dots, n_s\}$, while each term in the sum counts the number of co-vectors $\{\mathfrak{W}^{(j_1 \dots j_{n_s^{(\mathfrak{g})}})}\}$ for fixed $n_s^{(\mathfrak{g})} \in [2, \dots, n_s]$. Interestingly, the number of facets $\tilde{\nu}\left[\mathcal{N}_{\mathcal{G}}\right]$ is independent on the topology of the graph $\mathcal{G}$ for a fixed number of sites $n_s$, Finally, two facets identified by any two of the co-vectors $\{\mathfrak{W}^{(j_1 \dots j_{n_s^{(\mathfrak{g})}})}, n_s^{(\mathfrak{g})} = 1, \dots, n_s\}$ of the Newton polytope $\mathcal{N}_{\mathcal{G}}$ turn out to be compatible if they correspond to tubings that can be mapped into each other by erasing or adding single or nested tubings.

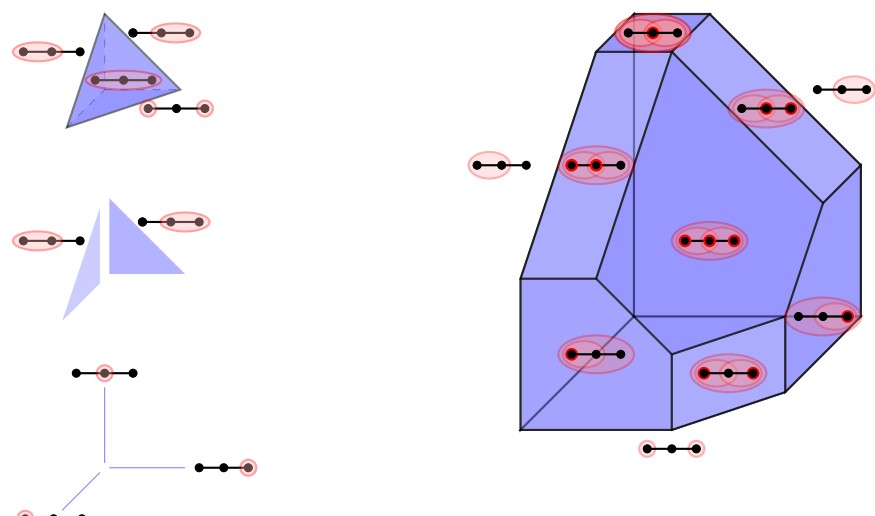

Figure 3: Newton polytope associated to the denominators of the three-site graph $\mathcal{G}$. It is constructed as a Minkowski sum of the simplices corresponding to all *connected subgraphs of $\mathcal{G}$* (*on the left*). It can be realised by truncating the top-dimensional simplex, based on all the allowed tubings, which corresponds to a subset of the facet of the simplex itself (*on the right*).

The knowledge of the compatible facets, allows dividing the domain of integration into sectors $\Delta_{\mathfrak{W}_c}$, $\mathfrak{W}_c$ being a given set of compatible facets:

$$\mathbb{R}_+^{n_s} := \bigcup_{\{\mathfrak{W}_c\}} \Delta_{\mathfrak{W}_c} \,. \tag{64}$$

Besides for the two-site line graph contributing to the wavefunction for conformally coupled scalars, the rational integrand in (57) has a numerator which is a polynomial of degree higher than zero. As discussed earlier, for such cases, it is useful to consider the integral $\mathcal{I}_{\mathcal{G}}^{(0)}$ as a sum of integrals that are a Mellin transform, whose parameter are shifted, of a rational function whose denominator is the same as the original integral, while its numerator is a constant. Alternatively, it is convenient to express the integrand via one of the triangulations of the underlying cosmological-type polytope in such a way that no spurious boundary is added [4, 65], *e.g.*

$$\mathcal{I}_{\mathcal{G}}^{(0)}[\alpha, \tau', \mathcal{X}] = \sum_{\{\mathfrak{G}_c\}} \int_{\mathbb{R}_+^{n_s}} \left[ \frac{dz}{z} z^{\alpha} \right] \prod_{\mathfrak{g}' \in \mathfrak{G}_c} \frac{1}{\left[ q_{\mathfrak{g}'}(z, \mathcal{X}_{\mathfrak{g}'}) \right]^{\tau'_{\mathfrak{g}'}}} \prod_{\mathfrak{g} \in \mathfrak{G}_{\circ}} \frac{1}{\left[ q_{\mathfrak{g}}(z, \mathcal{X}_{\mathfrak{g}}) \right]^{\tau'_{\mathfrak{g}}}} \,, \tag{65}$$

where $\mathfrak{G}_{\circ}$ is a set of $k$ subgraphs that identifies a subspace of the adjoint surface, while $\mathfrak{G}_c$ is a set of $(n_s + n_e - k)$ compatible subgraphs which are not in $\mathfrak{G}_{\circ}$. One of these representations can be expressed as a sum of all the possible ways of taking connected subgraphs [64, 65][19] and hence the Newton polytope associated to each term is still a nestohedron: it can be realised as a truncation of the top-dimensional simplices via lower-dimensional ones associated to these nested subgraphs – see Figure 4.

The representation (65) has the virtue of guaranteeing that no spurious possible divergent directions are added, and the intersection of the different Newton polytopes provides the convergence of the full integral.

---

[19]Such a recursion relation was found in the context of the Bunch-Davies wavefunction [64, 65], it is still applicable for the cosmological correlators: one of the triangulation of the weighted cosmological polytope provides a representation in terms of wavefunction graphs [4], to which the above-mentioned recursion relation can be applied.

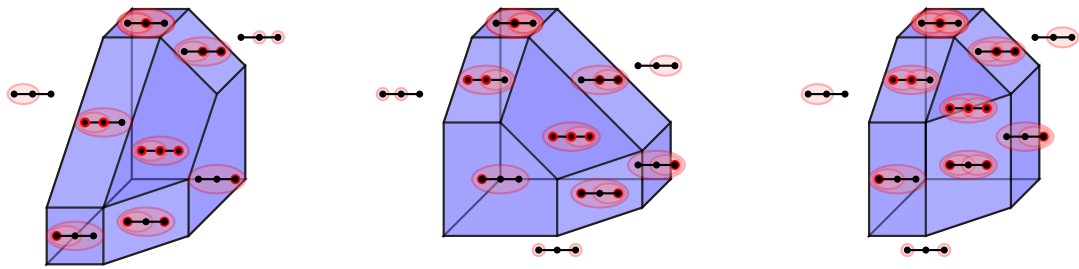

Figure 4: Newton polytope associated to the two integrals with numerator of degree 0 in which the three-site graph can represent, and their overlap. They both can be constructed via a truncation of the top-dimensional simplex via lower dimensional simplices associated to the underlying subgraphs in such a way that iteratively connected subgraphs are taken. Their overlap (*right*) represents the convergence of the full integral associated to the three-site tree graph, with its combinatorial structure identical to the nestohedron in Fig. 3.

As described above, once the compatible co-vectors are identified, the integral in each sector defined by them can be conveniently parametrised as

$$
\mathcal{I}_{\mathcal{G}}\big|_{\Delta_{\mathfrak{W}_c}} := \mathcal{I}_{\Delta_{\mathfrak{W}_c}} = \int_0^1 \prod_{\mathfrak{W} \in \mathfrak{W}_c} \left[ \frac{d\zeta_{\mathfrak{W}}}{\zeta_{\mathfrak{W}}} \zeta_{\mathfrak{W}}^{-\lambda_{\mathfrak{W}}} \right] \frac{\mathfrak{n}_\delta(\zeta, \mathcal{X})}{\prod\limits_{\mathfrak{g} \subseteq \mathcal{G}} \left[ q_{\mathfrak{g}}(\zeta, \mathcal{X}) \right]^{\tau_{\mathfrak{g}}}} \,,
\tag{66}
$$

with the variables $\zeta_{\mathfrak{W}}$ defined according to (55) as

$$
x_j \longrightarrow \prod_{\mathfrak{W} \in \mathfrak{W}_c} \zeta_{\mathfrak{W}}^{-\mathfrak{e}_j \cdot \omega_{\mathfrak{W}}} \,.
\tag{67}
$$

The infra-red behaviour is encoded in those sectors which are bounded by the co-vectors

$$
\left\{ \mathfrak{W}^{(j_1 \dots j_{n_s^{(\mathfrak{g})}})}, \, n_s^{(\mathfrak{g})} = 1, \dots, n_s \right\} \,.
$$

Which sectors actually contribute depend on which $\lambda_{\mathfrak{W}} \geq 0$. Note that as the actual value of lambda is related to the number of tubings that identify the associated facet of the Newton polytope, there is a hierarchy among the directions, with $\lambda^{(j_1 \dots j_{n_s^{(\mathfrak{g})}})} > \lambda^{(j_1 \dots j_{\tilde{n}_s^{(\mathfrak{g})}})}$ for $n_s^{(\mathfrak{g})} > \tilde{n}_s^{(\mathfrak{g})}$. Hence, $\lambda^{(1 \dots n_s)}$ is the highest possible value.

**Logarithmic divergences**  For $\lambda^{(1 \dots n_s)} \longrightarrow 0$, then the sectors contributing to the divergence are all those containing $\mathfrak{W}^{(1 \dots n_s)}$ – all the other sectors contains co-vectors such that the related $\lambda$ is negative. In this case, the divergence is logarithmic and the integral in one of the divergent sector can be written as

$$
\mathcal{I}_{\Delta_{\mathfrak{W}_c}}^{(0)} \sim \int_0^1 \frac{d\zeta_{\mathfrak{W}^{(1 \dots n_s)}}}{\zeta_{\mathfrak{W}^{(1 \dots n_s)}}} \zeta_{\mathfrak{W}^{1 \dots n_s}}^{-\lambda^{(1 \dots n_s)}} \times \int_0^1 \prod_{\mathfrak{W} \in \mathfrak{W}_c \setminus \{\mathfrak{W}^{(1 \dots n_s)}\}} \left[ \frac{d\zeta_{\mathfrak{W}}}{\zeta_{\mathfrak{W}}} \right] \frac{\mathfrak{n}_\delta(\zeta)}{\prod\limits_{\mathfrak{g} \subseteq \mathcal{G}} \left[ q_{\mathfrak{g}}(\zeta) \right]^{\tau_{\mathfrak{g}}}} + \dots
\tag{68}
$$

The first integration decouples and provides a pole in $\lambda^{(1 \dots n_s)}$, while the remaining integrations show an integrand which is nothing but the original integrand computed at $\zeta_{\mathfrak{W}^{(1 \dots n_s)}} = 0$. As this behaviour is common to all sectors sharing $\mathfrak{W}^{(1 \dots n_s)}$ and all the other sectors are well-behaved, the integral $\mathcal{I}_{\mathcal{G}}^{(0)}$ can be written in terms of the original site weights as

$$
\mathcal{I}_{\mathcal{G}}^{(0)} \sim \frac{1}{-\lambda^{(1 \dots n_s)}} \int_{\mathbb{R}_+^{n_s}} \prod_{s \in \mathcal{V}} \left[ \frac{dx_s}{x_s} x_s^{\alpha_s} \right] \frac{1}{\mathrm{Vol}\{\mathrm{GL}(1)\}} \frac{\mathfrak{n}_\delta(x, \mathcal{X} = 0)}{\prod\limits_{\mathfrak{g} \subseteq \mathcal{G}} \left[ q_{\mathfrak{g}}(x, \mathcal{X}_{\mathfrak{g}} = 0) \right]^{\tau_{\mathfrak{g}}}} + \dots
\tag{69}
$$

This integral is truly a $(n_s - 1)$-dimensional integral – as from (68), it decouples into a one-dimensional integral, that provides the divergences, and the remaining integrations. However, it is possible to write it in a $n_s$-dimensional fashion by introducing a GL(1)-redundancy, which is specified by Vol{GL(1)}.

It is important to note that the subgraphs $\{\mathfrak{g}_{s_j}, j = 1, \ldots, n_s\}$ containing a single site contributes via $q_{\mathfrak{g}_{s_j}}(x_{s_j}, \mathcal{X} = 0) := x_{s_j}$. Hence, the divergent part in (69) can be written as,

$$\mathcal{I}_{\mathcal{G}}^{(0)} \sim \frac{1}{-\lambda^{(1\ldots n_s)}} \int_{\mathbb{R}_+^{n_s}} \prod_{s \in \mathcal{V}} \left[ \frac{dx_s}{x_s} x_s^{\alpha_s - \tau_{\mathfrak{g}_s}} \right] \frac{1}{\text{Vol}\{\text{GL}(1)\}} \frac{\mathfrak{n}_\delta(x, \mathcal{X} = 0)}{\prod_{\mathfrak{g} \subseteq \mathcal{G}\backslash\{\mathfrak{g}_s\}} \left[ q_\mathfrak{g}(x, \mathcal{X} = 0) \right]^{\tau_\mathfrak{g}}} + \ldots, \quad (70)$$

where the integrand can be interpreted as coming from the original graph by iteratively contracting the pairs of sites onto each other and assigning the sum of their weights to the site obtained from such a contraction [62]. From a combinatorial-geometrical point of view, the integrand can be seen as being:

❏ for a (generalised) cosmological polytope $\mathcal{P}_{\mathcal{G}}$, the covariant restriction [66] of its canonical function $\Omega(\mathcal{Y}, \mathcal{P}_{\mathcal{G}})$ onto the hyperplane $\mathcal{H} := \bigcap_{e \in \mathcal{E}} \widetilde{\mathcal{W}}^{(\mathfrak{g}_e)}$ defined as the intersection of the hyperplanes $\{\widetilde{\mathcal{W}}^{(\mathfrak{g}_e)}, e \in \mathcal{E}\}$ containing the building blocks of the construction – *i.e.* the triangles associated to the 2-site tree graphs in which a general graph can be decomposed:

$$\Omega^{(n_s)}(\mathcal{Y}_{\mathcal{H}}, \mathcal{P}_{\mathcal{G}} \cap \mathcal{H}) = \oint_{\mathcal{H}} \frac{\langle \mathcal{Y} d^{n_s + n_e - 1} \mathcal{Y} \rangle}{(2\pi i)^{n_e}} \frac{\Omega(\mathcal{Y}, \mathcal{P}_{\mathcal{G}})}{\prod_{e \in \mathcal{E}} (\mathcal{Y} \cdot \widetilde{\mathcal{W}}^{(\mathfrak{g})})}, \quad (71)$$

❏ for a weighted cosmological polytope $\mathcal{P}_{\mathcal{G}}^{(w)}$, the hyperplane $\mathcal{H}$ defined above contains a codimension-$n_e$ face [4], and hence the integrand – that will still be indicated as $\Omega^{(n_s)}(\mathcal{Y}, \mathcal{P}_{\mathcal{G}}^{(w)})$ is encoded in the canonical function of such a face $\mathcal{P}_{\mathcal{G}}^{(w)} \cap \mathcal{H}$:

$$\Omega^{(n_s)}(\mathcal{Y}_{\mathcal{H}}, \mathcal{P}_{\mathcal{G}} \cap \mathcal{H}) = \frac{1}{\prod_{e \in \mathcal{E}} (\mathcal{Y} \cdot \widetilde{\mathcal{W}}^{(\mathfrak{g}_e)})} \Omega(\mathcal{Y}_{\mathcal{H}}, \mathcal{P}_{\mathcal{G}}^{(w)} \cap \mathcal{H}). \quad (72)$$

This provides a clear geometrical interpretation for the leading logarithmic infra-red divergences, which can know be directly extracted from the cosmological-like polytope description.

**Power-law divergences** If $\lambda^{(1\ldots n_s)} > 0$, then all the sectors defined via $\mathfrak{W}^{(j_1 \ldots j_{n_s^{(\mathfrak{g})}})}$ such that $\lambda^{(j_1 \ldots j_{n_s^{(\mathfrak{g})}})} \geq 0$ contribute to the divergences. Said differently, as $\lambda^{(j_1 \ldots n_s^{(\mathfrak{g})})} > \lambda^{(j_1 \ldots \tilde{n}_s^{(\mathfrak{g})})}$ for $n_s^{(\mathfrak{g})} > \tilde{n}_s^{(\mathfrak{g})}$, all the sectors defined via (some of) the co-vectors $\left\{ \mathfrak{W}^{(j_1 \ldots j_{n_s^{(\mathfrak{g})}})}, n_s^{(\mathfrak{g})} \in [\tilde{n}_s^{(\mathfrak{g})} + 1, n_s] \right\}$, with $\tilde{n}_s^{(\mathfrak{g})}$ being the highest value for which $\lambda^{(j_1 \ldots j_{\tilde{n}_s^{(\mathfrak{g})}})} < 0$. Let $\mathfrak{W}_{\text{div}}$ be the collection of co-vectors satisfying such a condition. From the structure of the integral (66) in a given sector $\Delta_{\mathfrak{W}_c}$, it is straightforward to see that the leading divergence is given by the sectors containing the highest number, namely $\tilde{\nu}_{\text{div}}$, of elements of $\mathfrak{W}_{\text{div}}$. Then, the integrals in these sectors develop a pole of multiplicity $\tilde{\nu}_{\text{div}}$ and its coefficient is a $(n_s - \tilde{\nu}_{\text{div}})$-fold integral. The subleading divergences, instead, will take contribution from a higher number of sectors: at order $\tilde{\nu}_{\text{div}} - k$ ($k \in [0, \tilde{\nu}_{\text{div}} - 1]$), the sectors contributing are all those with a number of elements of $\mathfrak{W}_{\text{div}}$ greater or equal to $\tilde{\nu}_{\text{div}} - k$.

For each sector, the leading and subleading divergences can be conveniently organised by expressing the integrand in terms of a (multiple) Mellin-Barnes representation, and mapping the integral in a (multiple) sum whose argument shows the poles in the $\lambda^{(j_1 \ldots j_{n_s^{(\mathfrak{g})}})}$'s – for an explicit computation, see the following example as well as Appendix B.

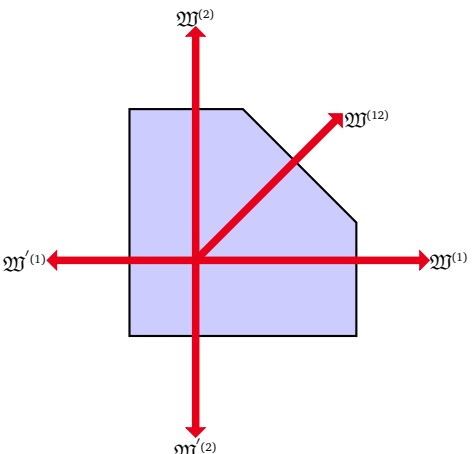
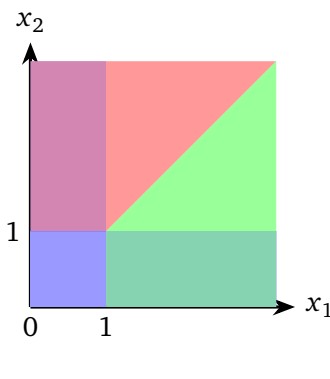

Figure 5: *On the left:* Newton polytope corresponding to the two-site tree graph. It has five possible divergent directions that divide the domain of integration in five sectors. *On the right:* Sectors in which the domain of integration is decomposed in the original site weights parametrisation.

**An illustrative example: The two-site tree graph**  It is instructive to fix the ideas with a concrete example. Let us consider the cosmological integral associated to the two-site line graph:

$$\mathcal{I}_2^{(0)}[\alpha] := \int_0^\infty \frac{dx_1}{x_1} x_1^\alpha \int_0^\infty \frac{dx_2}{x_2} x_2^\alpha \frac{1}{\left(x_1 + x_2 + \mathcal{X}_{\mathcal{G}}\right)\left(x_1 + \mathcal{X}_{\mathfrak{g}_1}\right)\left(x_2 + \mathcal{X}_{\mathfrak{g}_2}\right)}. \quad (73)$$

As discussed earlier, the asymptotic structure of $\mathcal{I}_2^{(0)}$ is captured by a nestohedron in $\mathbb{P}^2$ constructed from the triangle (the top-dimensional simplex in $\mathbb{P}^2$) and truncating via the segments corresponding to two of its side and corresponding to one single tubing each – see Figure 2. Such a nestohedron is a pentagon whose facets are identified by the co-vectors

$$\mathfrak{W}^{'(1)} = \begin{pmatrix} -\alpha_{\text{R}} \\ -1 \\ 0 \end{pmatrix}, \quad \mathfrak{W}^{'(2)} = \begin{pmatrix} -\alpha_{\text{R}} \\ 0 \\ -1 \end{pmatrix}, \quad \mathfrak{W}^{(1)} = \begin{pmatrix} \alpha_{\text{R}} - 2 \\ 1 \\ 0 \end{pmatrix}, \quad \mathfrak{W}^{(2)} = \begin{pmatrix} \alpha_{\text{R}} - 2 \\ 0 \\ 1 \end{pmatrix}, \quad \mathfrak{W}^{(12)} = \begin{pmatrix} 2\alpha_{\text{R}} - 3 \\ 1 \\ 1 \end{pmatrix}, \quad (74)$$

where $\alpha_{\text{R}} := \text{Re}\{\alpha\}$. The region of integration $\mathbb{R}_+^2$ can thus be divided into five sectors, each of which is determined by a pair of adjacent co-vectors – see Figure 5. The integral (73) is divergent along a given $\mathfrak{W}^{(j_1 \dots j_{n_s}(\mathfrak{g}))}$ if $\lambda^{(j_1 \dots j_{n_s}(\mathfrak{g}))} \geq 0$. Hence, depending on the divergences of interest, it is possible to focus on a subset of the sectors only. The integral (73) is divergent in the infra-red if $\lambda^{(12)} := 2\alpha_{\text{R}} - 3 \geq 0$ vel $\lambda^{(j)} : 2 - \alpha_{\text{R}} \geq 0$ – the equality signals a logarithmic divergence, while the strict positivity signals a power law divergence. If the divergence is logarithmic, it receives a contribution from two sectors, both of them sharing the co-vector $\mathfrak{W}^{(12)}$ and differing by $\mathfrak{W}^{(j)}$ ($j = 1, 2$). Let $\left\{\Delta_{\text{IR}}^{(j)}, j = 1, 2\right\}$ indicate these two sectors. In each of them, the integral can be parametrised as

$$\begin{aligned} \mathcal{I}_{\Delta_{\text{IR}}^{(j)}} &= \int_0^1 \frac{d\zeta_j}{\zeta_j} \left(\zeta_j\right)^{-\lambda^{(j)}} \\ &\times \int_0^1 \frac{d\zeta_{12}}{\zeta_{12}} \left(\zeta_{12}\right)^{-\lambda^{(12)}} \frac{1}{\left(1 + \zeta_j + \mathcal{X}_{\mathcal{G}}\zeta_{12}\zeta_j\right)\left(1 + \mathcal{X}_{\mathfrak{g}_{j+1}}\zeta_{12}\right)\left(1 + \mathcal{X}_{\mathfrak{g}_j}\zeta_{12}\zeta_j\right)}, \end{aligned} \quad (75)$$

with $\left\{x_j := \zeta_{12}^{-\mathfrak{e}_j \cdot \omega_{12}} \zeta_j^{-\mathfrak{e}_j \cdot \omega_j}, j = 1, 2\right\}$. If $\lambda^{12} \longrightarrow 0$, then the integral over $\zeta_j$ is finite and the divergence come just from the integration over $\zeta_{12}$. The divergent contribution is then given

by,

$$
\mathcal{I}_{\Delta_{\mathrm{IR}}^{(j)}} = \int_0^1 \frac{d\zeta_j}{\zeta_j} \frac{(\zeta_j)^{-\lambda^{(j)}}}{(1+\zeta_j)} \times \int_0^1 \frac{d\zeta_{12}}{\zeta_{12}} (\zeta_{12})^{-\lambda^{(12)}} + \ldots
$$
$$
= \frac{1}{2} \left[ \psi^{(0)}\left(\frac{-\lambda^{(j)}+1}{2}\right) - \psi^{(0)}\left(\frac{-\lambda^{(j)}}{2}\right) \right] \times \frac{1}{-\lambda^{(12)}} + \ldots , \tag{76}
$$

where $\psi^{(0)}(z)$ is the digamma function. As $\lambda^{(12)} := 2\alpha_{\mathrm{R}} - 3 \longrightarrow 0$, then $\lambda^{(j)} := \alpha_{\mathrm{R}} - 2 \longrightarrow -1/2$: the divergence is a simple pole and its coefficient becomes $\pi/2$. As discussed above, the integral can be recast as

$$
\mathcal{I}_{\Delta_{\mathrm{IR}}^{(j)}} = \frac{1}{-\lambda^{(12)}} \int_{\mathbb{R}_2^+} \prod_{s \in \mathcal{V}} \frac{1}{\mathrm{Vol}\{GL(1)\}} \left[ \frac{dx_s}{x_s} x_s^{\alpha-1} \right] \frac{1}{x_1 + x_2} , \tag{77}
$$

where $\alpha_{\mathrm{R}} = -\lambda^{(j)} + 1 = 3/2$. The integrand can be thought of a graph made of a single site with weight $x_1 + x_2$.

For power-law divergences, the divergences come from four out of the five sectors. Taking, for the sake of argument, $\lambda^{(12)} \longrightarrow 1$, then $\lambda^{(j)} \longrightarrow 0$. Two sectors defined by $(\mathfrak{W}^{(2)}, \mathfrak{W}^{(12)})$ and $(\mathfrak{W}^{(12)}, \mathfrak{W}^{(1)})$ shows both double and simple poles, while the other two just single poles: in order to correctly compute the subleading divergences, it is necessary to consider all these contributions. A way to organise them is via a Mellin-Barnes representation for the integrand $\Omega(\zeta, \Delta_{\mathrm{IR}}^{(j)})$ in (75):

$$
\Omega(\zeta, \Delta_{\mathrm{IR}}^{(j)}) = \int_{-i\infty}^{+i\infty} \prod_{r=1}^3 [d\xi_r \Gamma(-\xi_r)\Gamma(1+\xi_r)] \left(\mathcal{X}_\mathcal{G}\right)^{\xi_1} \left(\mathcal{X}_{\mathfrak{g}_{j+1}}\right)^{\xi_2} \left(\mathcal{X}_{\mathfrak{g}_j}\right)^{\xi_3} \frac{(\zeta_j)^{\xi_3+\xi_1} (\zeta_{12})^{\xi_1+\xi_2+\xi_3}}{(1+\zeta_j)^{\xi_1}} . \tag{78}
$$

The integral $\mathcal{I}_{\Delta_{\mathrm{IR}}^{(j)}}$ then becomes

$$
\mathcal{I}_{\Delta_{\mathrm{IR}}^{(j)}} = \frac{-\mathcal{X}_{\mathfrak{g}_{j+1}}}{(-\lambda^{(12)}+1)(-\lambda^{(j)})} + \frac{1}{-\lambda^{(12)}+1} \left[ -\mathcal{X}_\mathcal{G} \int_0^1 \frac{d\zeta_j}{\zeta_j} \frac{\zeta_j^{-\lambda^{(j)}+1}}{1+\zeta_j} + \frac{-\mathcal{X}_{\mathfrak{g}_j}}{-\lambda^{(j)}+1} \right]
$$
$$
+ \frac{1}{-\lambda^{(j)}} \left[ \frac{1}{-\lambda^{(12)}} + \sum_{m \geq 2} \frac{\left(-\mathcal{X}_{\mathfrak{g}_j}\right)^m}{\Gamma(1+m)(-\lambda^{(12)}+m)} \right] + \ldots \tag{79}
$$

As a final comment, this approach is completely general and allows tackling leading and subleading divergences in arbitrary power-law FRW cosmologies. While this example as well as all the discussion in this section was devoted to tree-level, the combinatorics of the nestohedra turns out to encode the asymptotic behaviour of loop integral as well, as we will discuss in the next section. Despite it is possible to treat all divergences, what is still missing in this story is a full-fledge combinatorial understanding of the subleading divergences: while the leading one is understood as the restriction of the polytope associated to a given graph onto a special codimension-$n_e$ hyperplane, an understanding for the subleading ones along similar lines is still not available, and we leave it for future work.

## 4.2 The asymptotic structure of loop graphs

The above analysis showed how the asymptotic behaviour of tree-level contributions to cosmological observables is controlled by the combinatorial structure of a nestohedron, whose realisation as sequential truncations of a top-dimensional simplex by a collection of lower-dimensional simplices given by a subset of its faces, allow determining straightforwardly all

the directions along which they can diverge as well as their degree of divergences. The compatibility condition on the facets, expressed in terms of tubings on the underlying graph, allow determining the sectors in which such directions decompose the domain of integration. This in turn allows extracting all the divergences in any of the above directions.

Let us turn now to the loop contributions to cosmological observables. Given an arbitrary graph $\mathcal{G}$ with $n_s$ sites, $n_e$ edges – of which $n_e^{(L)}$ are loop edges – and $L$ loops, it has associated an integral $\mathcal{I}_{\mathcal{G}}^{(L)}$ of the form

$$
\mathcal{I}_{\mathcal{G}}^{(L)}[\alpha,\beta] := \int_{\mathbb{R}_+^{n_s}} \prod_{s\in\mathcal{V}}\left[\frac{dx_s}{x_s} x_s^{\alpha_s}\right] \int_{\Gamma} \prod_{e\in\mathcal{E}^{(L)}}\left[\frac{dy_e}{y_e} y_e^{\beta_e}\right] \mu_d(y_e,\mathcal{X};n_s) \frac{\mathfrak{n}_\delta(x,y;\mathcal{X})}{\prod_{\mathfrak{g}\subseteq\mathcal{G}}\left[q_{\mathfrak{g}}(x,y;\mathcal{X}_{\mathfrak{g}})\right]^{\tau_{\mathfrak{g}}}}\,, \quad (80)
$$

with $\mathcal{E}^{(L)}$ being the set of loop edges, $\mu_d$ and $\Gamma$ being respectively the measure and the contour of integration, and $\mathcal{X}_{\mathfrak{g}}$ parametrising the external kinematics associated to the subgraph $\mathfrak{g}\subseteq\mathcal{G}$. Such a class of integral is *almost* of the Mellin type: while the integration over the site weights is effectively a Mellin transform, the integration over the edge weights is along a contour given by the non-negative condition on the volume of a top-dimensional simplex in $\mathbb{R}^{n_e}$ and all its faces – the vanishing of volume determines the boundary of the integration region. The rational integrand is again determined by the combinatorics of the cosmological polytopes, with the linear polynomials $\{q_{\mathfrak{g}}, \mathfrak{g}\subset\mathcal{G}\}$ associated to the subgraphs of $\mathcal{G}$ and the facets of the polytopes, while the degree-$\delta$ polynomial $\mathfrak{n}_\delta$ providing the adjoint surface. Finally, together with the contour of integration, the integration measure represents a feature that was not present for tree graphs: it is related to the volume of simplices in the edge weight space, and can be written as a polynomial. Such polynomial turns out to have a power which depends on both space dimension and number of edges of the graph, and can be integer or half-integer – see Section 3.1. If for an integer power, it would be possible, at least in principle, to expand the integral over the monomials of the measure, in practise this becomes quickly cumbersome (except for the case in which the edge weight space is two-dimensional, this polynomial is not factorisable, and becomes quickly of higher order as the number of edges of the underlying graph is increased) and does not apply to half-integer powers. However, when the edge weight are considered small, the non-negative condition that defines the contour of integration forces to take just one of the edge weights per loop to be arbitrarily small, while the others acquire a definite and finite value: this is the only point in the domain of integration which is allowed, and it is located on its boundary. Hence, the problem of determining the sectors that contribute to this type of divergence drastically simplifies and, for a given loop, the measure can be expanded as a single edge weight becomes small. This type of divergent directions determines the infra-red behaviour of the loop integral. As instead the edge weights are taken to be large, the non-negativity condition of the integration contour force all those associated to the same loop to be taken large in the same way. This feature simplifies the analysis and, as for the previous case, allows for a large edge weight expansion of the measure. This divergent direction codifies the ultraviolet behaviour of the integral.

Finally, the polynomial appearing in the measure, coming from a determinant, shows also negative coefficients, which can make the Newton polytope analysis more subtle for those regions where the polynomial vanishes within the integration domain. However, the integration is over the region where this polynomial is positive and vanishes just at its boundaries. Thus, the asymptotic analysis can be carried out with no modifications, as there are no regions inside the domain of integration where this polynomial changes sign.

With this information at hand, it is therefore possible to perform the analysis of the asymptotic behaviour of the integral similarly as for tree graphs, bearing in mind the non-negative condition from the contour of integration just outlined.

**Divergent directions: The site weight integration**   Let us begin with considering the integration over the site weights only. The Newton polytope associated to a loop graph $\mathcal{G}$ then lives in $\mathbb{P}^{n_s}$ and it still has the structure of a nestohedron: it can be obtained as the Minkowski sum of simplices, as in the tree case. The main differences with the latter are constituted by: *i*) the weight for the top-dimensional simplex as, for a loop graph, there are $n_e + 1$ subgraphs which include all the sites of $\mathcal{G}$, *i.e.* $\mathcal{G}$ itself and the $n_e$ subgraphs obtaining by erasing one edges;[20] *ii*) the collection of simplices in codimension-$k$ ($k \in [1, n_s - 1]$) on which the Minkowski sum runs is larger than for the tree case. For the 2-site $L$-loop graphs ($L \geq 1$), just *i*) holds, while *ii*) is as for the tree graphs – this implies that the set of co-vectors $\{\mathfrak{W}^{'(j_1)}, \mathfrak{W}^{(j_1 \ldots j_{n_s^\mathfrak{g}})} \in \mathbb{P}^{n_s}, j_{n_s^{(\mathfrak{g})}} = 1, \ldots, n_s, n_s^\mathfrak{g} = 1, \ldots, n_s\}$ is the same, up to their component $\lambda^{(j_1 \ldots j_{n_s^{(\mathfrak{g})}})}$ which is affected by the weights. Said differently, the divergent directions for all 2-site $L$-loop graphs are the same, and what changes is the way in which they are approached.

**Divergent directions: The edge weight integration**   Let us now consider the edge integration only – this is relevant when either there is no site weight integration at all (*e.g.* a conformally-coupled scalar with conformal interactions), or when it can be replaced by a derivative operator (*e.g.* when the real part of the Mellin parameter is negative – it is related to metric warp factor of the type $a(\eta) \sim \eta^\gamma$, with $\gamma > 0$). For the sake of simplicity, let us consider graphs with no tree substructure, in such a way that the number of loop edges is the number of total edges. At one-loop, the poles of the canonical function of the relevant polytope can depend on either one or two edge weight. Consequently, the Newton polytope associated to a one-loop graph is given by a weighted Minkowski sum of triangles and segments, irrespectively of the dimension. This implies that just the two-site one-loop graph involves the top-dimensional simplex. Nevertheless, as for the tree graphs, it is possible to realise this nestohedron starting with the top-dimensional simplex, and truncating based on tubings on the underlying subgraphs – see Figure 6. The facets are then identified by the co-vectors

$$\mathfrak{W}^{'(j)} = \begin{pmatrix} 0 \\ -\mathfrak{e}_j \end{pmatrix}, \qquad \mathfrak{W}^{(j_1 \ldots j_{n_e^{(\mathfrak{g})}})} = \begin{pmatrix} \lambda^{(j_1 \ldots j_{n_e^{(\mathfrak{g})}})} \\ \mathfrak{e}_{j_1 \ldots j_{n_e^{(\mathfrak{g})}}} \end{pmatrix}, \tag{81}$$

where $n_e^{(\mathfrak{g})} = 1, \ldots, n_e$, with $\lambda^{(j_1 \ldots j_{n_e^{(\mathfrak{g})}})}$ being given by the number of tubings associated to a given facet, as in the tree case.

The loop integration can also be considered as the Mellin transform of an integrand (80), and hence the Newton polytope get shifted by the vector $(1, \beta) - \beta := (\beta_e)_{e \in \mathcal{E}}$ – made out by the Mellin parameters, which reflects into the co-vectors identifying the facets by the shift

$$\lambda^{'(j)} \longrightarrow \tilde{\lambda}^{'(j)} = -\beta_{e_j}, $$
$$\lambda^{(j_1 \ldots j_{n_e^{(\mathfrak{g})}})} \longrightarrow \tilde{\lambda}^{(j_1 \ldots j_{n_e^{(\mathfrak{g})}})} = \lambda^{(j_1 \ldots j_{n_e^{(\mathfrak{g})}})} + \sum_{e \in \mathcal{E}_\mathfrak{g}^{\text{ext}}} \beta_e, \tag{82}$$

where, as usual, $\mathcal{E}_\mathfrak{g}^{\text{ext}}$ is the set of edges departing from the subgraph $\mathfrak{g}$ – if $\beta_e = \beta$, $\forall e \in \mathcal{E}$, then the shift becomes simply $\lambda^{(j_1 \ldots j_{n_e^{(\mathfrak{g})}})} \longrightarrow \tilde{\lambda}^{(j_1 \ldots j_{n_e^{(\mathfrak{g})}})} = \lambda^{(j_1 \ldots j_{n_e^{(\mathfrak{g})}})} + n_e^{(\mathfrak{g})} \beta$. For the sake of simplicity, let us focus on one-loop graphs.

The infra-red behaviour of the integrals associated to them, according to the nestohedron analysis, is encoded into the sector identified by $\{\mathfrak{W}^{'(j)}, j = 1, \ldots, n_e\}$. Note that the change of variables dictated by this sector map the edge variable into themselves. Hence, the integral in this sector is the very same original integral but with the domain of integration which is

---

[20]To be precise, the statement as formulated is valid for graphs such that all the edges are in a loop. If there are also tree edges, then the number of such subgraphs is $n_e^{(L)}$.

bounded by an arbitrary cut-off according to the contour of integration $\Gamma$. As we argued earlier, the non-negativity condition imposed by the contour of integration, make the integral divergent just when one of the edge variables approaches zero. In order to determine the degree of divergence along the directions $\{\mathfrak{W}'^{(j)}, j = 1, \dots n_e\}$ it is necessary to understand how the measure contributes. As for the time being we are restricting ourselves to one-loop and to graphs with no tree substructure, the graphs are all polygons – thus they have the same number of sites and edges. It is convenient to label the edge connecting the site $s_j$ to the site $s_{j+1}$ with $y_{j,j+1}$. Firstly, as we argued earlier, the non-negative condition selects which divergent direction can be taken simultaneously. This implies that even in a given sector identified by a set of compatible co-vectors of the Newton polytope, the singularity that would be reached along two (or more) directions cannot be accessed. It is therefore convenient to separate them by considering any given sector where this phenomenon occurs as a union of sectors, each of which contains just the directions which are allowed by the non-negative condition from the integration contour. For example, let us consider the sectors which contains the singularity at $y_{12} \longrightarrow 0$. The further split highlighted above can be obtained via the change of variables

$$y_{12} = P_{23}\frac{\omega}{1+\omega}, \qquad y_{j,j+1} = P_{2\dots j} + 2\omega_{j,j+1}\frac{\omega_{j,j+1}}{1+\omega_{j,j+1}}. \tag{83}$$

Under this change of variable, the edge-weight integration measure acquires the form[21]

$$\mu_d\left(y^2, P^2\right) = \left[\frac{\omega}{(1+\omega)^2}\right]^{d-n_e-1} \tilde{\mu}_d\left(\omega, \omega_{ij}\right). \tag{84}$$

In this case, the singularity for $y_{12} \longrightarrow 0$ is separated from the others that are pushed to infinity:

$$\mathcal{I}_\Delta^{(1)} = \int_0^{+\infty} \frac{d\omega}{\omega} \omega^{\beta+d-3} \int_{\Gamma'} \prod_{e\in\mathcal{E}} d\omega_e \left[P_{2\dots j_e} + P_{23}\omega_e\frac{\omega}{1+\omega}\right]^{\beta_e-1} \tilde{\mu}_d(\omega, \omega_e)\Omega(\omega, \omega_e; \mathcal{X}), \tag{85}$$

where factors of $(1+\omega)$ have been absorbed into $\tilde{\mu}_d(\omega, \omega_e)$. Having the directions which are not compatible with $y_{12} \longrightarrow 0$ be pushed to infinity, the sector $\Delta$ can be further decomposed in two sectors, one containing the divergence at $y_{12} \longrightarrow 0$ only:

$$\mathcal{I}_\Delta^{(1)} = \int_0^1 \frac{d\omega}{\omega} \omega^{\beta+d-3} \int_{\Gamma'} \prod_{e\in\mathcal{E}} d\omega_e \left[P_{2\dots j_e} + P_{23}\omega_e\frac{\omega}{1+\omega}\right]^{\beta_e-1} \tilde{\mu}_d(\omega, \omega_e)\Omega(\omega, \omega_e; \mathcal{X}). \tag{86}$$

In this form, the extraction of the leading and subleading divergences proceeds as discussed in the previous sections.

---

[21]This is straightforward to see considering that the Cayley-Menger determinant which characterises it can be written as:

$$\begin{aligned}
\text{CM}(y_{i,i+1}^2, P_{2\dots j+1}^2) &= \omega^2\text{CM}(P_{2\dots j+1}^2) + \frac{1}{4}\sum_{i=1}^{n-1}\left\{\left[(-1)^i\left(\omega^2 + P_{i\,i+1}^2 - (P_{i\,i+1} - \omega\,\omega_{i\,i+1})^2\right)\right]\right.\\
&\qquad\qquad\qquad \left. \times \sum_{j=1}^{n-1}(-1)^{j+1}\left(\omega^2 + P_{j\,j+1}^2 - \left(P_{j\,j+1} + \omega\,\omega_{j\,j+1}\right)^2\right)G_{\sigma_j}\right\}\\
&= \omega^2\text{CM}(P_{2\dots j+1}^2) + \frac{\omega^2}{4}\sum_{i,j=1}^{n-1}(-1)^{i+j+1}\left\{\left(\omega\left(1 - \omega_{i\,i+1}^2\right) - 2\,a_{i\,i+1}P_{i\,i+1}\right)\right.\\
&\qquad\qquad\qquad\qquad\qquad \left. \times \left(\omega\left(1 - \omega_{j\,j+1}^2\right) - 2\,\omega_{j\,j+1}P_{j\,j+1}\right)\text{CM}_{\sigma_j}\right\},
\end{aligned}$$

where $\text{CM}_{\sigma_j}$ is the determinant of a minor of the total Cayley-Menger matrix, which contains only entries depending on $P_e$.

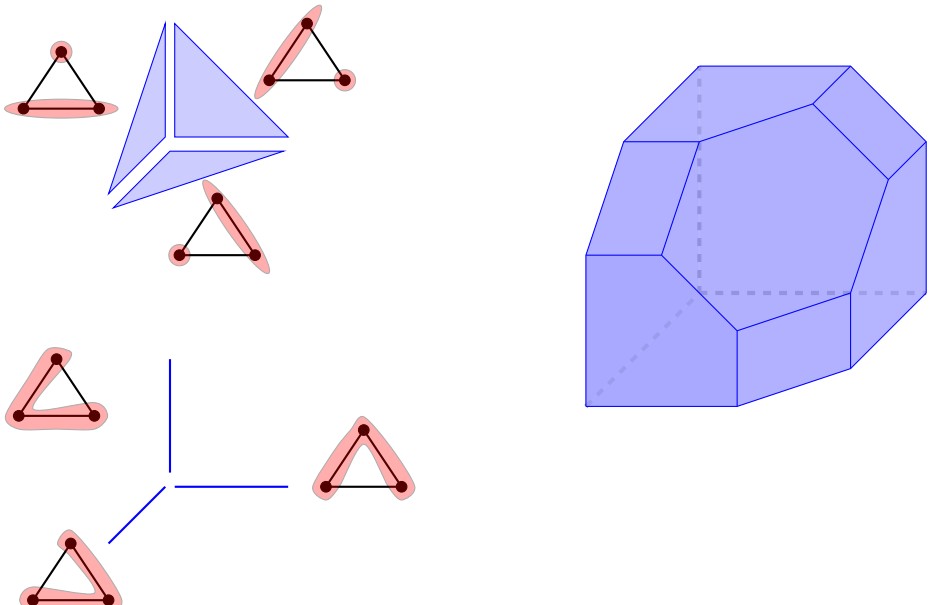

Figure 6: Newton polytope associated to the edge weight integration only for a three-site one-loop graph. *On the left:* The simplices building blocks and their tubings. *On the right:* The Newton polytope is a nestohedron which can be realised by truncating the top-dimensional simplex (a tetrahedron) via the tubings associated to the underlying graph.

The second subsector contains divergences that occur when other edges weights approach zero. However, polygons are symmetric objects, so the results for these divergences can be obtained from the one for $y_{12}$ via a simple relabelling.

For the ultraviolet divergences, the domain of integration forces all the edge weight to approach infinity in the same way, which means that only the direction $\mathfrak{W}^{(1\cdots n_e)}$ becomes divergent. Similarly, one can focus on the sectors which are bounded by the co-vector $\mathfrak{W}^{(1\cdots n_e)}$, and separate the related singularities from the others via the following change of variables

$$y_{12} = \omega_{12}, \qquad y_{j,j+1} = \omega_{12} + P_{j,j+1}\omega_{j,j+1}. \tag{87}$$

For higher loops, a similar strategy applies: it is possible to construct the Newton polytope in a similar fashion, and then restrict the divergent directions allowed by the non-negative condition from the contour of integration. In order to impose such a condition, it is convenient to proceed in a loop by loop fashion – which is also a way in which both measure and contour of integration are constructed. This also allows working recursively, applying the lower-loop treatment to the higher loop graphs.

**Divergent directions: The full integration**   Let us now consider both site and weight integration at the same time. This allows to understand the interplay between divergences coming from the loop modes and the one coming from the expansion of the universe.

In this case, it is possible to determine the possible divergent directions in an unconstrained fashion from the analysis of the full Newton polytope, and, as in the previous discussion, constrain them with the non-negative conditions from the edge-weight integration contour.

The Newton polytope in this case is still a nestohedron living in $\mathbb{P}^{n_s+n_e}$: despite the top-dimensional simplex does not correspond to any of the linear polynomials in the denominators, it is still possible to realise it as a sequential truncation based on the underlying graph of the top-dimensional simplex. The top-dimensional simplex, to which no tubing is associated, is

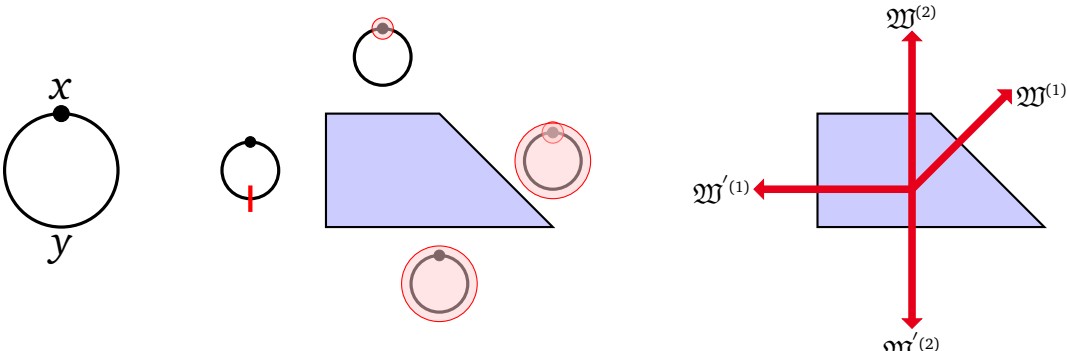

Figure 7: *On the left:* The one-site one loop graph. *Center:* the realisation of the Newton polytope for the integral (88) associated to the one-site one-loop graph in terms of tubings. *On the right:* Possible divergent directions for the integral (88).

sequentially truncated via simplices of $\mathbb{P}^{n_s^{(\mathfrak{g})}+n_e^{\text{ext}}}$, where $n_s^{(\mathfrak{g})}$ and $n_e^{\text{ext}}$ are respectively the number of sites in the subgraph $\mathfrak{g} \subseteq \mathcal{G}$ and the number of edges departing from it. All the facets correspond to tubings in the same way as outlined in the previous sections. As the top-dimensional simplex does not correspond to any denominator, there is no tubing associated to it and hence it does not contribute to the asymptotic behaviour and serves just for the construction itself. The rest of the analysis follows the discussion for the edge weight integration.

## 4.3 Examples

It is instructive to illustrate the procedure discussed above in some concrete, simple, but yet non-trivial and illustrative examples.

**The one-site one-loop graph**    Let us begin with the simplest graph, with the integrand provided by a generalised cosmological polytope [65]

$$\mathcal{I}_{\mathcal{G}}^{(1)} = \int_0^\infty \frac{dx}{x} x^\alpha \int_0^\infty \frac{dy}{y} y^\beta \frac{\partial^2}{\partial x^2} \frac{1}{(x+X_{\mathcal{G}})(x+X_{\mathcal{G}}+2y)}, \tag{88}$$

where the parameters $\alpha$ and $\beta$ are analytically continued to regularise the integral, which has now the form of a usual Mellin transform of a rational function. The Newton polytope is the weighted Minkowski sum of a triangle and a segment in $\mathbb{P}^2$ and can be realised as sequential truncation of the triangle – see Figure 7. It has thus four possible divergent directions, given by the co-vectors associated to the tubings:

$$\mathfrak{W}^{'(1)} = \begin{pmatrix} -\alpha_{\text{R}} \\ -1 \\ 0 \end{pmatrix}, \quad \mathfrak{W}^{'(2)} = \begin{pmatrix} -\beta_{\text{R}} \\ 0 \\ -1 \end{pmatrix}, \quad \mathfrak{W}^{(2)} = \begin{pmatrix} \beta_{\text{R}} - 3 \times 1 \\ 0 \\ 1 \end{pmatrix}, \quad \mathfrak{W}^{(12)} = \begin{pmatrix} \alpha_{\text{R}} + \beta_{\text{R}} - 3 \times 2 \\ 1 \\ 1 \end{pmatrix}, \tag{89}$$

where in the first entries of $\mathfrak{W}^{(1)}$ and $\mathfrak{W}^{(2)}$ has been emphasised the contribution of the weight $\tau_{\text{R}}$ and the number of tubings $n^{(\mathfrak{g})}$ as $\tau \times n^{(\mathfrak{g})}$. Note also that for the time being, the numerator of (88) has not been taking into account. It is a second order polynomial with 6 monomials. So one strategy would be to split the integral into a sum of integrals according to the monomial expansion of the numerator: for each term, the Newton polytope is the same as in Figure 7 but shifted by the powers of the relevant monomial. Then, the integral is convergent in the overlap among these Newton polytopes. The co-vectors have then the same form as (89), but with

$$(\alpha_{\text{R}}, \beta_{\text{R}}) \longrightarrow (\alpha_{\text{R}}, \beta_{\text{R}}) + (\rho_x, \rho_y), \tag{90}$$

where

$$(\rho_x, \rho_y) = \{(0,0), (1,0), (2,0), (0,1), (1,1), (1,2)\} . \tag{91}$$

Alternatively, it is possible to decompose the integrand according to one of the triangulations of the generalised cosmological polytope with physical poles only

$$\mathcal{I}_{\mathcal{G}}^{(1)} = \int_0^{+\infty} \frac{dx}{x} x^\alpha \int_0^{+\infty} \frac{dy}{y} y^\beta \left[ \frac{2}{(x+X_{\mathcal{G}})^3(x+X_{\mathcal{G}}+2y)} \right.$$
$$\left. + \frac{2}{(x+X_{\mathcal{G}})^2(x+X_{\mathcal{G}}+2y)^2} + \frac{2}{(x+X_{\mathcal{G}})(x+X_{\mathcal{G}}+2y)^3} \right]. \tag{92}$$

Each term shows again the very same Newton polytope arising, but with different weights while the Mellin parameters are unchanged.

The integration domain is therefore divided into four sectors $\Delta_{2',1'}$, $\Delta_{1',2}$, $\Delta_{2,12}$ and $\Delta_{12,2'}$, which are respectively bounded by $(\mathfrak{W}^{'(2)}, \mathfrak{W}^{'(1)})$, $(\mathfrak{W}^{'(1)}, \mathfrak{W}^{(2)})$, $(\mathfrak{W}^{(2)}, \mathfrak{W}^{(12)})$, and $(\mathfrak{W}^{(12)}, \mathfrak{W}^{'(2)})$.

The leading infra-red divergence is captured by the sector $\Delta_{12,2'}$, with the two directions $(\mathfrak{W}^{(12)}, \mathfrak{W}^{(2)})$ becoming simultaneously divergent for $\lambda^{(12)} := \alpha_{\mathrm{R}} + \beta_{\mathrm{R}} - (\tau_{\mathfrak{g}} + \tau_{\mathcal{G}}) \times 2 - \rho_x - \rho_y \geq 0$ and $\lambda^{'(2)} := -\beta_{\mathrm{R}-\rho_y} \geq 0$, and developing a double pole when these conditions are satisfied simultaneously. More precisely, this sector captures the simultaneous divergence from the infinite volume (*i.e.* $x \longrightarrow +\infty$) and from the low energy mode in the loop (*i.e.* $y \longrightarrow 0$).

Subleading divergences are taken into account by supplementing this sector with $\Delta_{1',2}$ and $\Delta_{12,2'}$, where the two directions $(\mathfrak{W}^{'(2)}, \mathfrak{W}^{'(1)})$ diverge individually.

Let us explicitly consider an integral in the sector $\Delta_{2',1'}$

$$\mathcal{I}_{\Delta_{12,2'}} = 2^{1-\beta} \left[ X_{\mathcal{G}} \right]^{\alpha-4} \int_0^1 \frac{d\zeta_{12}}{\zeta_{12}} (\zeta_{12})^{-\lambda^{(12)}} \int_0^1 \frac{d\zeta_2'}{\zeta_2'} (\zeta_2')^{-\lambda^{'(2)}} \frac{1}{(1+\zeta_{12})^{\tau_{\mathcal{G}}} \left(1+\zeta_{12}+\zeta_2'\right)^{\tau_{\mathfrak{g}}}}, \tag{93}$$

where $x$ and $y$ have been rescaled by $X_{\mathcal{G}}$ and $X_{\mathcal{G}}/2$ respectively, and $(x, y) = (\zeta_{12}^{-1}, \zeta_2' \zeta_{12}^{-1})$. If $(\lambda^{(12)}, \lambda^{'(2)}) \longrightarrow (0,0)$, the divergence is logarithmic and can be readily extracted to be

$$\mathcal{I}_{\Delta_{12,2'}} = 2^{1-\beta_\star} \left[ X_{\mathcal{G}} \right]^{\alpha_\star-4} \int_0^1 \frac{d\zeta_{12}}{\zeta_{12}} (\zeta_{12})^{-\lambda^{(12)}} \int_0^1 \frac{d\zeta_2'}{\zeta_2'} (\zeta_2')^{-\lambda^{'(2)}} + \dots$$
$$= 2^{1-\beta_\star} \left[ X_{\mathcal{G}} \right]^{\alpha_\star-4} \frac{1}{(-\lambda^{(12)})(\lambda^{'(2)})} + \dots, \tag{94}$$

where $(\alpha_\star, \beta_\star)$ are the values of the Mellin parameters computed at $(\lambda^{(12)}, \lambda^{'(2)}) = (0,0)$.

For power-law divergences as well as to extract the subleading contribution from this sector, as we discussed earlier, it is convenient to express the integrand via a Mellin-Barnes representation, allowing to perform the integration and mapping $\mathcal{I}_{\Delta_{12,2'}}$ into a double sum

$$\mathcal{I}_{\Delta_{12,2'}} = \frac{1}{\Gamma(\tau_{\mathfrak{g}})} \sum_{k\geq 0} \frac{(-1)^k}{\Gamma(k+1)} \sum_{m\geq 0} \frac{(-1)^m}{\Gamma(m+1)} \frac{\Gamma(\tau_{\mathfrak{g}}+m)}{\Gamma(\tau_{\mathcal{G}}+\tau_{\mathfrak{g}}+m)} \frac{\Gamma(\tau_{\mathcal{G}}+\tau_{\mathfrak{g}}+m+k)}{(k-\lambda^{(12)})(m-\lambda^{'(2)})}. \tag{95}$$

One can readily see that the integral develops poles when $(\lambda^{(12)}, \lambda^{'(2)})$ are non-negative and makes it straightforward to extract the information for the specific values of interest, obtaining both the information about double and simple poles.

The leading ultraviolet divergence is along $\mathfrak{W}^{(2)}$ and therefore is captured by the two sectors $\Delta_{2,12}$ and $\Delta_{1',2'}$, and can be extracted in a similar fashion.

From equation (93), we see the generality of this procedure. At no point we have introduced any information about the specific theory we have. In the subsequent equations there

is the underlying assumption that the theory must have logarithmic divergences in (94), or power law divergences (95), but that is it. We can use the above results to compute the leading terms for a theory with quartic couplings ($\varphi^4$), in de Sitter, and massless scalars. We will apply this to the correlator, in particular to the disconnected component, for which the total energy singularity is replaced by a factor of $1/y$, thus $(\tau_{\mathcal{G}}, \tau_{\mathfrak{g}}) = (0, 3)$, and $(\alpha_\star, \beta_\star) = (3, 0)$, in this case the numerator is still one, so $(\rho_x, \rho_y) = (0, 0)$. This means that we are in the case where $(\lambda^{(12)}, \lambda^{'(2)}) = (0, 0)$, and the result (94) applies directly. From this we learn that we have two logarithmic divergences. One of the divergences coming from the $x$-integration, which in the related literature is referred to as secular divergence. The other logarithmic divergence comes from the massless state running in the loop, and this is a standard infrared divergence. The coefficient of the leading divergence is also proportional to $p^{-3}$, where $p$ is the external momentum. All of this agrees with the literature on the subject, check [42]. In the above discussion, we considered only the disconnected component contributing to the correlator. For this process, there are only two contributions to the correlator, the disconnected part and the wavefunction. To compute the contribution from the wavefunction, we have the total energy singularity but one less factor of $y$ in the denominator. Then, $(\lambda^{(12)}, \lambda^{'(2)}) = (-1, -1)$ which means we have no divergences anymore. This is a verification of the fact that the disconnected components contributing to the correlator are the most divergent, and thus contribute with the leading divergences. This has been noted in the literature, and in particular one can check [51] for a detailed discussion on this. We hope that this simple example has motivated the generality of our method, we can use it to compute leading and sub-leading divergences, for the different terms in the correlator, as well as for different theories and cosmologies. It becomes particularly simple to compare different theories, as the Newton polytope is the same for a given process, and we just need to change the parameters on which the $\lambda$-function depends on.

**The two-site one-loop graph**  Let us move on to a more interesting example, where the non-trivial form of the edge weight integration measure and the related integration domain appear. The simplest of these cases is the two-site one-loop graph: The

$$\mathcal{I}_{\mathcal{G}}^{(1)} = \prod_{s \in \mathcal{V}} \left[ \frac{dx_s}{x_s} x^\alpha \right] \int_\Gamma \prod_{e \in \mathcal{E}} \left[ \frac{dy_e}{y_e} y_e^\beta \right] \frac{[-\mathrm{CM}(y^2, P^2)]^{\frac{d-3}{2}}}{P^{d-2}} \, \Omega(x, y; \mathcal{X}) \,, \tag{96}$$

where $\Omega(x, y; \mathcal{X})$ is the canonical function of the (generalised/weighted) cosmological polytopes associated to this graph, and the contour of integration is

$$\Gamma := \left\{ x \in \mathbb{R}_+^2, \, y \in \mathbb{R}_+^2 \, | -\mathrm{CM}(y^2, P^2) \geq 0 \right\}$$

– see Appendix A for more details. Note that for $d = 3$ the measure of integration simplifies, with the only novelty with respect to the previous case given by the contour of integration $\Gamma$. As discussed previously, the contour of integration selects a subset of possible divergent directions that would come from the analysis of the integrand alone. In particular, the edge weights can get to zero just once at a time – as $y_a \longrightarrow 0$ then $y_b \longrightarrow P$, and vice versa, when $y_b \longrightarrow 0$, $y_a \longrightarrow P$ –, while they have to approach infinity simultaneously.

Let us look at the asymptotic structure by first considering the Newton polytope associated to the integrand $\Omega(x, y; \mathcal{X})$. In order to give a general account, let us consider the following form

$$\Omega(x, y; \mathcal{X}) \tag{97}$$

$$= \frac{\mathfrak{n}_\delta(x, y; \mathcal{X})}{[q_{\mathcal{G}}(x, y; \mathcal{X})]^{\tau_{\mathcal{G}}} [q_{\mathfrak{g}_a}(x, y; \mathcal{X})]^{\tau_{\mathfrak{g}_a}} [q_{\mathfrak{g}_b}(x, y; \mathcal{X})]^{\tau_{\mathfrak{g}_b}} [q_{\mathfrak{g}_1}(x, y; \mathcal{X})]^{\tau_{\mathfrak{g}_1}} [q_{\mathfrak{g}_2}(x, y; \mathcal{X})]^{\tau_{\mathfrak{g}_1}}} \,,$$



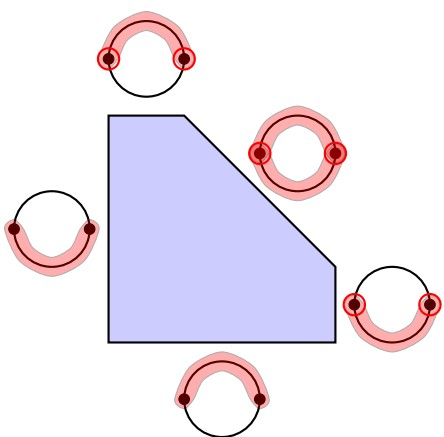

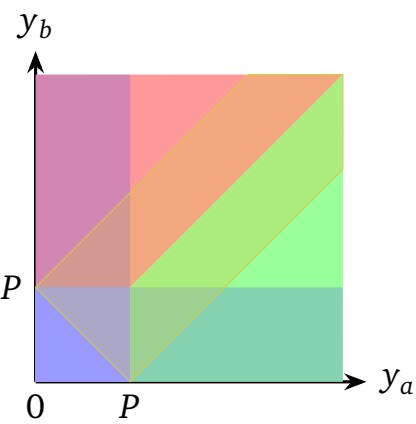

Figure 8: *On the left:* The Newton polytope associated to the edge weight integration for the two-site one-loop graph is a nestohedron whose facets are identified by certain tubings. They correspond to directions in which the integral might diverge. The number of tubings instead correspond to how these directions are approached. A subset of these directions are selected by the integration contour. *On the right:* The knowledge of compatible facets of the Newton polytope allows to decompose $\mathbb{R}^2_+$ in sectors, which can be restricted onto the domain of integration via the non-negativity condition which defines it.

where

$$q_{\mathcal{G}} := x_1 + x_2 + \mathcal{X}_{\mathcal{G}}, \qquad q_{\mathfrak{g}_a} := x_1 + x_2 + 2y_a + \mathcal{X}_{\mathcal{G}}, \quad q_{\mathfrak{g}_b} := x_1 + x_2 + 2y_b + \mathcal{X}_{\mathcal{G}},$$
$$q_{\mathfrak{g}_1} := x_1 + y_a + y_b + \mathcal{X}_{\mathfrak{g}_1}, \quad q_{\mathfrak{g}_2} := x_2 + y_a + y_b + \mathcal{X}_{\mathfrak{g}_2}. \tag{98}$$

Let us begin with consider solely the loop integration – as mentioned earlier this is sensible for those cases in which the integration over the site weight is absent. The Newton polytope then lives in $\mathbb{P}^2$ and it is a nestohedron built via sequential truncation of a triangle via two segments corresponding to two of its sides – combinatorially it is the same polytope obtained in the two-site tree graph:

$$\mathfrak{W}^{'(1)} = \begin{pmatrix} -\beta_{\mathrm{R}} \\ -1 \\ 0 \end{pmatrix}, \quad \mathfrak{W}^{'(2)} = \begin{pmatrix} -\beta_{\mathrm{R}} \\ 0 \\ -1 \end{pmatrix}, \quad \mathfrak{W}^{(1)} = \begin{pmatrix} \lambda^{(1)} \\ 1 \\ 0 \end{pmatrix}, \quad \mathfrak{W}^{(2)} = \begin{pmatrix} \lambda^{(2)} \\ 0 \\ 1 \end{pmatrix}, \quad \mathfrak{W}^{(12)} = \begin{pmatrix} \lambda^{(12)} \\ 1 \\ 1 \end{pmatrix}, \tag{99}$$

where $\lambda^{(1)} := \beta - \tau_{\mathfrak{g}_a} - \tau_{\mathfrak{g}_1} - \tau_{\mathfrak{g}_2}$, $\lambda^{(2)} := \beta - \tau_{\mathfrak{g}_b} - \tau_{\mathfrak{g}_1} - \tau_{\mathfrak{g}_2}$ and $\lambda^{(12)} := 2\beta - \tau_{\mathfrak{g}_a} - \tau_{\mathfrak{g}_b} - \tau_{\mathfrak{g}_1} - \tau_{\mathfrak{g}_2}$.

The restriction onto the contour of integration allows three out of these five directions: $\mathfrak{W}^{'(1)}$, $\mathfrak{W}^{'(2)}$ and $\mathfrak{W}^{(12)}$. So the sectors contributing to the infra-red divergences are $\Delta_{2'1'}$, $\Delta_{1'2}$ and $\Delta_{12'}$, which are respectively identified by $(\mathfrak{W}^{(2')}, \mathfrak{W}^{(1')})$, $(\mathfrak{W}^{'(1)}, \mathfrak{W}^{'(2)})$ and $(\mathfrak{W}^{(1)}, \mathfrak{W}^{'(2)})$; while the sectors $\Delta_{2,12}$ an $\Delta_{12,1}$ – respectively bounded by $(\mathfrak{W}^{(2)}, \mathfrak{W}^{(12)})$ and $(\mathfrak{W}^{(12)}, \mathfrak{W}^{(1)})$ – codify the ultraviolet divergences. In the sector $\Delta_{2'1'}$, the integral acquires the form[22]

$$\mathcal{I}_{\Delta_{2'1'}} = \int_0^P \frac{d\zeta_1'}{\zeta_1'} (\zeta_1')^\beta \int_{P-\zeta_1'}^P \frac{d\zeta_2'}{\zeta_2'} (\zeta_2')^\beta \frac{\left[\mathrm{CM}(\zeta'^2, P^2)\right]^{\frac{d-3}{2}}}{P^{d-2}} \Omega(x, \zeta'; \mathcal{X}), \tag{101}$$

---

[22] A peculiarity of this case is that it is the only one in which the Cayley-Menger determinant factorises in a product of linear polynomials

$$-\mathrm{CM}(y^2, P^2) = (y_a + y_b + P)(y_a - yb + P)(y_a + y_b - P)(-y_a + y_b + P), \tag{100}$$

with the zeroes reached at the boundary of the domain of integration. Consequently, it is possible to have an explicit form for the contour of integration, with $y_a \in \mathbb{R}_+$ and $y_b \in [|y_a - P|, y_a + P]$ or, which is the same $\Delta_{\mathrm{IR}} \cup \Delta_{\mathrm{UV}} \equiv \{y_a \in [0, P], y_b \in [P - y_a, y_a + P]\} \cup \{y_a \in [P, +\infty[, y_b \in [y_a - P, y_a + P]]\}$.

where $\Omega(x, \zeta'; \mathcal{X})$ can be thought to be expresses via one of its triangulation with physical poles only.

This sector contains two singularities which cannot be taken simultaneously. It is useful to make the contour of integration independent of any integration variable

$$\mathcal{I}_{\Delta_{2'1'}} = \int_0^{+\infty} \frac{d\omega_1}{\omega_1} \frac{\omega_1^{\beta+d-1}}{(1+\omega_1)^{2(\beta+(d-3))+1}}$$
$$\times \int_{-1}^0 d\omega_2 \, (1+\omega_1(1+\omega_2))^{\beta-1} \, \widetilde{\mu}(\omega_1, \omega_2) \, \Omega(\omega_1, \omega_2, \mathcal{X}) \,, \quad (102)$$

where $\widetilde{\mu}(\omega_1, \omega_2) := [(1-\omega_2)(1+\omega_2)(2+(1+\omega_2)\omega_1)(2+(3+\omega_2)\omega_1)]^{(d-3)/2}$ – it is achieved via $\zeta_1' = P\omega_1(1+\omega_1)^{-1}$, $\zeta_2' = P(1+\omega_2\omega_1(1+\omega_1)^{-1})$. In this form the two divergences, originally approached as $y_a \longrightarrow 0$ and $y_b \longrightarrow 0$ separately, are clearly separated: the former is reached as $\omega_1 \longrightarrow 0$ while the latter as $\omega_1 \longrightarrow +\infty$.

The integral (102) can be further decomposed into two subsectors $\Delta_{2'1'} := \Delta_{2'} \cup \Delta_{1'}$, each of which containing just one of the two divergences. For example,

$$\mathcal{I}_{\Delta_{2'}} = \int_0^1 \frac{d\omega_1}{\omega_1} \frac{\omega_1^{\beta+d-3}}{(1+\omega_1)^{2(\beta+(d-3))+1}}$$
$$\times \int_{-1}^0 d\omega_2 \, (1+\omega_1(1+\omega_2))^{\beta-1} \, \widetilde{\mu}(\omega_1, \omega_2) \, \Omega(\omega_1, \omega_2, \mathcal{X}) \,, \quad (103)$$

and the divergent terms can be extracted expanding around $\omega_1 \longrightarrow 0$

$$\mathcal{I}_{\Delta_{2'}} = \frac{\mathfrak{a}}{\beta+d-3} \Omega(0, \mathcal{X}) + \ldots \quad (104)$$

– at leading order, the canonical function becomes independent on $\omega_2$, and $\mathfrak{a}$ represents the integral over $\omega_2$ which is just a number. As we observed at tree-level, also at loops the coefficient of the leading divergence can be obtained by restricting the canonical function onto a special hyperplane.

The other divergence along the other infrared divergence can be deduced from this one, as the original integral is completely symmetric under the exchange of the two edge weights.

Finally, let us comment on the analysis of both the site- and edge-weight integration simultaneously. In this case, the Newton polytope lives in $\mathbb{P}^4$. Constructing, as usual, the nestohedron as a truncation based on the underlying graph, and considering a generic point in a system of local coordinates in $\mathcal{P}^4$ of the form $(t_1, t_a, t_b, t_2)$ labelling the powers of $(x_1, y_a, y_b, x_2)$ respectively, its facets are given by

$$\mathfrak{W}'^{(j)} = \begin{pmatrix} \lambda'^{(j)} \\ -\mathfrak{e}_j \end{pmatrix}, \quad \mathfrak{W}^{(j)} = \begin{pmatrix} \lambda^{(j)} \\ \mathfrak{e}_j \end{pmatrix}, \quad \mathfrak{W}^{(23)} = \begin{pmatrix} \lambda^{(23)} \\ \mathfrak{e}_{23} \end{pmatrix}, \quad \mathfrak{W}^{(1\ldots4)} = \begin{pmatrix} \lambda^{(1\ldots4)} \\ \mathfrak{e}_{1\ldots4} \end{pmatrix}, \quad (105)$$

where

$$\begin{aligned}
\lambda'^{(1)} &= -\alpha_{\mathrm{R}} = \lambda'^{(4)} \,, & \lambda'^{(2)} &= -\beta_{\mathrm{R}} = \lambda'^{(3)} \,, \\
\lambda^{(1)} &= \alpha_{\mathrm{R}} - \tau_{\mathcal{G}} - \tau_{\mathfrak{g}_a} - \tau_{\mathfrak{g}_b} - \tau_{\mathfrak{g}_1} \,, & \lambda^{(2)} &= \beta_{\mathrm{R}} - \tau_{\mathfrak{g}_a} - \tau_{\mathfrak{g}_1} - \tau_{\mathfrak{g}_2} \,, \\
\lambda^{(3)} &= \beta_{\mathrm{R}} - \tau_{\mathfrak{g}_b} - \tau_{\mathfrak{g}_1} - \tau_{\mathfrak{g}_2} \,, & \lambda^{(4)} &= \alpha_{\mathrm{R}} - \tau_{\mathcal{G}} - \tau_{\mathfrak{g}_a} - \tau_{\mathfrak{g}_b} - \tau_{\mathfrak{g}_2} \,, \\
\lambda^{(23)} &= 2\beta_{\mathrm{R}} - \tau_{\mathfrak{g}_a} - \tau_{\mathfrak{g}_b} - \tau_{\mathfrak{g}_1} - \tau_{\mathfrak{g}_2} \,, & \lambda^{(1\ldots4)} &= 2(\alpha_{\mathrm{R}} + \beta_{\mathrm{R}}) - \tau_{\mathcal{G}} - \tau_{\mathfrak{g}_a} - \tau_{\mathfrak{g}_b} - \tau_{\mathfrak{g}_1} - \tau_{\mathfrak{g}_2} \,.
\end{aligned} \quad (106)$$

Some comments are now in order. First, note that the collection of co-vectors $\{\mathfrak{W}'^{(j)}, \mathfrak{W}^{(j)}, \mathfrak{W}^{(23)}\}$ constitute the five possible divergent directions emerging from the analysis of the sole edge weight integration. The non-negativity condition imposed by the contour of integration, prevents the integral to become divergent along more than one direction

$\{\mathfrak{W}'^{(j)}, j = 2,3\}$ and $\{\mathfrak{W}^{(j)}, j = 2,3\}$. Hence, those sectors involving more than one of these directions can be further split, as we saw earlier. Secondly, the infra-red behaviour is encoded into the $\{\mathfrak{W}'^{(j)}, j = 2,3\}$ for internal low energy modes, while $\{\mathfrak{W}^{(j)}, j = 4,1\}$ codify the ones due to the expansion of the universe. Finally, $\lambda^{(1\dots4)}$ encodes the effect of internal high energy modes as the universe expands.

Finally, let us consider a sector $\Delta_{41}$ containing both $\mathfrak{W}^{(1)}$ and $\mathfrak{W}^{(4)}$ and take either of the two to be divergent, . *i.e.* $\lambda^{(1)} \geq 0$ or $\lambda^{(4)} \geq 0$. Then, following what discussed above for the site-weight integration, and taking $\lambda^{(j)} \longrightarrow 0$ the integral factorises into an integral over the remaining site-weight integration and an integral over the edge weight only

$$\mathcal{I}_{\Delta_{41}} = \frac{1}{-\lambda^{(j)}} \int_0^{+\infty} \prod_{s \in \mathcal{V}} [dx_s] \frac{1}{\text{Vol}\{GL(1)\}} \Omega(x, y = 0; \mathcal{X} = 0) \times \mathcal{I}_{\Delta_{41}}^{(\mathcal{E})}, \qquad (107)$$

where $\mathcal{I}_{\Delta_{41}}^{(\mathcal{E})}$ can be cast in the terms of the usual loop momentum as

$$\mathcal{I}_{\Delta_{41}}^{(\mathcal{E})} := \int d^d l \, \frac{1}{l^\beta \left( \vec{l} + \vec{P} \right)^\beta} \, . \qquad (108)$$

Interestingly enough, the contribution from the site-weight integration is related to the restriction of the relevant cosmological polytopes along a special hyperplane $\bigcap_{e \in \mathcal{E}} \widetilde{\mathcal{W}}^{(\mathfrak{g}_e)}$, while the edge weight integration can be recast in a flat-space loop integral associated to the same graph.

**The two-site two-loop graph** Let us now close this example section with the simplest two-loop example, given by the two-site two-loop graph – see Figure 9. As for the one-loop case discussed above, we consider the following general form for the integral associated to this graph

$$\mathcal{I}_{\mathcal{G}}^{(2)} = \int_{\mathbb{R}_+^2} \prod_{s \in \mathcal{V}} \left[ \frac{dx_s}{x_s} x_s^\alpha \right] \int_{\Gamma^{(2)}} \prod_{e \in \mathcal{E}} \frac{dy_e}{y_e} y_e^{\beta-1} \Omega_{\mathcal{G}}(x, y; \mathcal{X}), \qquad (109)$$

where

$$\Omega_{\mathcal{G}}(x, y; \mathcal{X}) = \frac{\mathfrak{n}_\delta(x, y; \mathcal{X})}{[q_{\mathcal{G}}]^{\tau_{\mathcal{G}}} \left[ \prod_{e=a,b,c} [q_{\mathfrak{g}_e}]^{\tau_{\mathfrak{g}_e}} \right] \left[ \prod_{(e_1,e_2)} [q_{\mathfrak{g}_{(e_1,e_2)}}]^{\tau_{\mathfrak{g}_{(e_1,e_2)}}} \right] \left[ \prod_{j=1,2} [q_{\mathfrak{g}_j}]^{\tau_{\mathfrak{g}_j}} \right]}, \qquad (110)$$

where $\mathfrak{g}_{a/b/c}$ is the subgraph with the edge labelled by $a/b/c$ departing from it, $\mathfrak{g}_{(e_1,e_2)}$ is the subgraph with the pair $(e_1, e_2)$ of edges departing from it (where

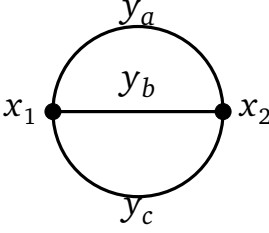

Figure 9: Two-site two-loop graph. The associated Newton polytope is lives in $\mathbb{P}^5$ and characterised by 16 facets. They signal the possible divergent direction, which are further limited by the non-negativity condition from the integration contour.

$(e_1, e_2) = \{(a, b), (b, c), (c, a)\})$, and as usual $\mathfrak{g}_j$ is the subgraph made out of the site $s_j$ only. The linear polynomials $\{q_{\mathfrak{g}}, \mathfrak{g} \subseteq \mathcal{G}\}$ are then explicitly given by

$$q_{\mathfrak{g}} := \sum_{s \in \mathfrak{V}_{\mathfrak{g}}} x_s + \sum_{e \in \mathcal{E}_{\mathfrak{g}}^{\text{ext}}} y_e + \mathcal{X}_{\mathfrak{g}}. \tag{111}$$

First, note that were we to consider the site integration only, the associated nestohedron would be the very same as for the other two-site graphs examined, with the $\mathfrak{W}$'s differing only by their $\lambda$-component: the possible divergences are the same, what differs is the way that such divergences are approached.

The edge weight integration is more interesting. As discussed in Section 3.1 and showed more explicit for the present case in Appendix A, it can be recast in the form of an iterated integral with two-site one-loop graphs. Thus, we can easily use the lesson from the analysis in the previous example: for each loop integration, the non-negative condition imposed by the contour of integration – see Appendix A for the explicit form of such contour – restrict the number of possible divergent directions. Hence, we can perform a full Newton polytope analysis, decompose the integral according to these sectors, and each sector involving directions that cannot become simultaneously divergent because of the contour non-negativity condition, can be split in subsectors each of which contain only the directions which can become divergent simultaneously.

The Newton polytope for the full integral lives in $\mathbb{P}^5$ with a generic point given by, $(\rho_1, \rho_a, \rho_b, \rho_c, \rho_2)$ which are respectively the powers of $(y_1, y_a, y_b, y_c, x_2)$. Its facets are given by

$$\mathfrak{W}^{'(j)} = \begin{pmatrix} \lambda^{'(j)} \\ -\mathfrak{e}_j \end{pmatrix}, \qquad \mathfrak{W}^{(j)} = \begin{pmatrix} \lambda^{(j)} \\ \mathfrak{e}_j \end{pmatrix}, \qquad \mathfrak{W}^{(j_1, j_2)} = \begin{pmatrix} \lambda^{(j_1, j_2)} \\ \mathfrak{e}_{j_1, j_2} \end{pmatrix},$$
$$\mathfrak{W}^{(234)} = \begin{pmatrix} \lambda^{(234)} \\ \mathfrak{e}_{234} \end{pmatrix}, \qquad \mathfrak{W}^{(1\dots5)} = \begin{pmatrix} \lambda^{(1\dots5)} \\ \mathfrak{e}_{1\dots5} \end{pmatrix}, \tag{112}$$

where $(j_1, j_2) = \{((2, 3), (3, 4), (4, 2))\}$. While in the one-loop case the integration contour was forbidding to take simultaneously two divergences both identified by one of the $\mathfrak{W}$'s covectors, in this case some of them are allowed, *e.g.* $\mathfrak{W}^{'(2)}$ and $\mathfrak{W}^{'(4)}$ – they belong to different loop subgraphs.

# 5 Conclusion and outlook

Understanding the infra-red behaviour of cosmological processes is of fundamental importance both for the reliability of our predictions for the initial condition of the subsequent classical evolution, and for getting full-fledged theoretical insights on the early universe physics. Ideally, a well-defined cosmological observable should be free of infra-red divergences. However, they turn out to be plagued by them, and hence they need to be re-summed or cancelled. The analysis under different approaches for de Sitter space showed that the leading divergences should re-sum to give the probability distribution predicted by stochastic inflation. Despite has been argued that this Markovian behaviour should survive at subleading order, it is not clear how this should occur [43, 61].

Reasonably, any well-defined probability distribution should be subject to the exact renormalisation group equations, which *are* diffusion-type equations – see for example [95]. An idea along these lines was explored for de Sitter in [51], confirming the expectation for the leading divergences.

Hence, the question about the fate of the subleading divergences and the underlying mechanism of a possible re-summation is still open. More generally, one can broaden the problem by

asking which consistency conditions a cosmological observable and their infra-red behaviour ought to satisfy, in an arbitrary FRW background, in order to re-sum. In the case of QCD, for example, the exponentiation of double and single poles for hard processes is tied to factorisation properties of scattering amplitude [96]. So, it is worth to ask which properties of cosmological observables in perturbation signal that the infra-red divergences are bound to resum.

The exact renormalisation group, supported by a perturbative analysis, seems promising for unravelling this issue. The recent combinatorial formulation for large classes of scalar theories in terms of (generalised/weighted) cosmological polytopes allowed to gain a deeper understanding of perturbation theory for cosmological observables.

In this paper, we begin to explore how these combinatorial ideas can encode the renormalisation group structure for cosmological observables in power-law FRW cosmologies, and the condition for re-summability. In particular, we show how the asymptotic behaviour is governed by a special class of nestohedra, which are directly tied to the cosmological polytope structure. It allowed us to predict all possible divergent directions, both in the infra-red and in the ultraviolet, and their degree of divergences from graphical rules, and simplify the extraction of the divergent behaviour via sector decomposition. Our work fixes a set-up in which the questions about sub-leading divergences, their re-summation and the consistency condition for re-summation on the infra-red divergent structure can be sharpened.

**The structure of the infra-red coefficients**    The combinatorial picture we introduced in this paper, allows to systematically study the asymptotic structure of an arbitrary graph in perturbation theory and to extract the divergences. For leading corrections, it was possible to establish a relation between the coefficient of the leading divergence and the restriction of the canonical form of the underlying (generalised/weighted) cosmological polytope onto a special hyperplane. While it is indeed possible to explicitly compute subleading divergences, no clear combinatorial-geometrical interpretation of the coefficients became manifest. In order to address the questions we posed above, it is important to understand: *i)* how the structure of the coefficients of the leading divergences is tied to an exact renormalisation group equation and hence to re-summation; *ii)* the combinatorial structure of the subleading coefficients and their properties; *iii)* the implication of changing of the divergence degree.

**Combinatorics and the exact renormalisation group**    The nestohedra we described encode both the infra-red and the ultraviolet behaviour of a given graph. This allows to have a handle of a number of effects which are described by the subleading corrections, such as decoherence. It would be interesting to understand on more general grounds the relation between such a combinatorial structure and the exact renormalisation group picture.

**The mathematical side: Newton polytopes and Minkowski differences**    The Newton polytope analysis for the asymptotic behaviour is beautifully simple when the integrands have either a numerator which is a polynomial $\mathfrak{n}_0$ of order $0$ or it is just a monomial. When it is a more general polynomial $\mathfrak{n}_\delta$ it cannot be naively be considered on the same footing as the denominators by considering its associated Newton polytope $\mathcal{N}[\mathfrak{n}_\delta]$ as having negative weight in the Minkowski sum with the Newton polytope associated to the denominators: this definition of Minkowski difference for a polytope *is not* the inverse operation of the Minkowski sum. In order for considering both numerator and denominator of an integrand on the same footing, it is necessary to have a definition of a Minkowski difference for polytopes which satisfied such condition. Such a definition for two polytopes $\mathcal{P}_1, \mathcal{P}_2 \in \mathbb{R}^n$ was proposed as $\mathcal{P}_1 - \mathcal{P}_2 := \{x \in \mathbb{R}^n \,|\, x + \mathcal{P}_2 \subseteq \mathcal{P}_1\}$, and it is such that $(\mathcal{P}_1 + \mathcal{P}_2) - \mathcal{P}_2 = \mathcal{P}_1$ [67]. It would be

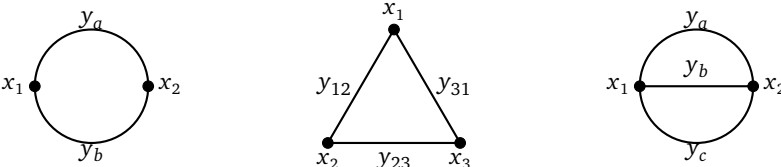

Figure 10: Examples of loop graphs. The associated loop integrals are $n_e$-fold if $n_e < d$, and the integration measure is proportional to the squared volume of a simplex in $\mathbb{P}^{n_s+n_e-2}$.

interesting to explore whether such a definition allows treating numerator and denominator on equal footing in the Newton polytope analysis.

## Acknowledgments

It is a pleasure to thank Giulio Salvatori for valuable discussions. P.B. would like to thank Subodh Patil and the Leiden Institue of Physics; Guillermo Silva and the Physics Institute in La Plata; Fiorenzo Bastianelli and the University of Bologna; Pierpaolo Mastrolia and the University of Padova, for the possibility of presenting the results reported in this paper. P.B. would also like to thank the developers of SageMath [97], Maxima [98], Polymake [99–102], TOPCOM [103], and Tikz [104]. P.B. would like to thank Dieter Lüst and Gia Dvali for making finishing it possible, as well as the Instituto Galego de Física de Altas Enerxías of the Universidade de Santiago de Compostela for hospitality during the very last stages of this work.

**Funding information** P.B. and F.V. have been partially supported during the first part of this work by the European Research Council under the European Union's Horizon 2020 research and innovation programme (No 725110).

## A The loop measure: Examples

In Section 3.1 we have discussed how, in the space of loop edge weights, the loop measure is associated to the squared volume of a certain simplex, and the integration region is given by the requirement of such volume to be non-negative, vanishing just at the boundary of that region. For the sake of clarity, we provide here some explicit example both at one- and two-loops which can be taken as a reference – concretely, the two- and three-site one loop graphs and the two-site two-loop graph.

**Two-site one-loop graph** Let us begin with the simplest non-trivial one-loop example, the two-site one-loop graph – see Figure 10. Let $\mathfrak{P} := \{\vec{p}_j,\, j = 1,\dots,n\}$ be the set of external spatial momenta. The external kinematics can be parametrised via

$$X_1 := \sum_{\vec{p}\in\mathfrak{P}_1} |\vec{p}|, \qquad X_2 := \sum_{\vec{p}\in\mathfrak{P}_2} |\vec{p}|, \qquad P := \left|\sum_{\vec{p}\in\mathfrak{P}_1} \vec{p}\right| = \left|\sum_{\vec{p}\in\mathfrak{P}_2} \vec{p}\right|, \tag{A.1}$$

where $\mathfrak{P}_1, \mathfrak{P}_2 \subset \mathfrak{P}$ such that $\mathfrak{P}_1 \cup \mathfrak{P}_2 = \mathfrak{P}$ and $\mathfrak{P}_1 \cap \mathfrak{P}_2 = \varnothing$ – *i.e.* $\mathfrak{P}_1$ and $\mathfrak{P}_2$ are the sets of external momenta at the two vertices of the graph. The loop space can instead be parametrised as

$$y_a := |\vec{l}|, \qquad y_b := |\vec{l}+\vec{P}|, \qquad \vec{P} := \sum_{\vec{p}\in\mathfrak{P}_1} \vec{p}. \tag{A.2}$$

For $d \geq 2$, *i.e.* the number of spatial dimension greater than the number of edges of the graph, then the loop integration is a two-fold integral in $y_a$ and $y_b$. From (37), the measure can be written in terms of the squared volume of a triangle in $\mathbb{P}^2$, whose boundaries' volumes are given by the triple $(y_a, y_b, P)$ and that is proportional to (minus) the Cayley-Menger determinant $\mathrm{CM}(y^2, P^2)$:

$$
\begin{aligned}
\mathrm{Vol}^2\{\Sigma_2(y, P)\} \sim -\mathrm{CM}(y^2, P^2) &= -\begin{vmatrix} 0 & 1 & 1 & 1 \\ 1 & 0 & y_a^2 & y_b^2 \\ 1 & y_a^2 & 0 & P^2 \\ 1 & y_b^2 & P^2 & 0 \end{vmatrix} \\
&= \left[(y_a + P)^2 - y_b^2\right]\left[y_b^2 - (y_a - P)^2\right].
\end{aligned} \tag{A.3}
$$

The proportionality factor in the measure depends on the $(2, 2)$-minor of (A.3), which returns the volume of the codimension-1 boundary of the triangle that purely depends on the external kinematics, *i.e* $\mathrm{CM}^{(2,2)}(y^2, P^2) = 2P^2$. The case (A.3) is the only one in which the Cayley-Menger determinant is factorisable. The non-negativity of (A.3) as well as of the individual integration variables $y_a$ and $y_b$ as well as of the external kinematic parameter $P$ defines the contour of integration $\Gamma$ to be,

$$
\Gamma_2 := \left\{\left[(y_a + P)^2 - y_b^2\right]\left[y_b^2 - (y_a - P)^2\right] \geq 0, \quad y_a \geq 0, \quad y_b \geq 0\right\}. \tag{A.4}
$$

This geometrical picture also allows to straightforwardly understand the behaviour of the measure as certain limits are taken. For example, as any of the elements of the triple $(y_a, y_b, P)$ are taken to zero, the triangle associated to the measure gets mapped into a segment by collapsing two of its vertices onto each other:

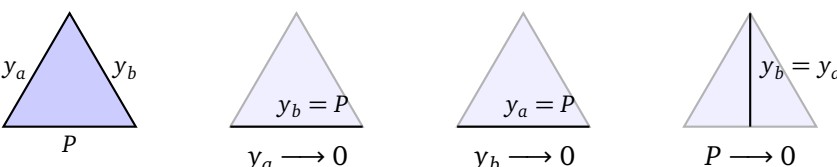

Figure 11: Example of the simplex whose volume provides the measure for the two-site one-loop graph. Taking the limit where one side shrinks, the other two sides collapse into the same length.

If the graph is characterised by just an external state for each site, then spatial momentum conservation implies that $X := X_1 = X_2$ and $P = X$. In this case, as $y_a \sim \rho X$,[23] the triangle volume also vanishes.

The full integral corresponding to the 2-site 1-loop graph then acquires the form

$$
\begin{aligned}
I_2^{(1)} = \frac{\pi^{\frac{d-2}{2}}}{\Gamma\left(\frac{d-2}{2}\right)} \int_0^{+\infty} \frac{dx_1}{x_1} x_1^\alpha \int_0^{+\infty} \frac{dx_2}{x_2} x_2^\alpha \\
\times \int_{\Gamma_2} dy_a^2 \, dy_b^2 \frac{\left[((y_a + P)^2 - y_b^2)(y_b^2 - (y_a - P)^2)\right]^{\frac{d-3}{2}}}{P^{d-2}} \Omega(x, y),
\end{aligned} \tag{A.5}
$$

where $\Omega(x.y)$ is the universal integrand provided by the combinatorial picture of standard / generalised / weighted cosmological polytopes. Note that for $d = 2$ – when the number of

---

[23]This is nothing but the collinear limit $\vec{l} \longrightarrow \rho\vec{P}$.

spatial dimension is the same as the number of edges of the graph, the squared volume of the triangle appears in the denominator and its vanishing – that typically occurs at the boundary of the integration region – might imply the appearance of a singularity from the measure which does not appear for $d > n_e = 2$. However, this is not the case, as the factor $y_a y_b$ cancels such potential divergence.

For $d < n_e = 2$, *i.e.* $d = 1$, then the edge weights are not independent, and the loop space is parameterised by one of them only. In the simple case under discussion, such a space is 1-dimensional and can be parameterised by any of the two edge weights. Without loss of generality, it is possible to take $y_a$. Then $y_b = y_a + P$ and $dl = dy_a$, with the integration region being just the positive real axis $\mathbb{R}_+$. The integral associated to two-site one-loop in $d = 1$ can then be written as

$$I_2^{(1)}\big|_{d=1} = \delta(X_2 - X_1) \int\limits_0^{+\infty} \frac{dx_1}{x_1} x_1^\alpha \int\limits_0^{+\infty} \frac{dx_2}{x_2} x_2^\alpha \int\limits_0^{+\infty} dy_a \int\limits_0^{+\infty} dy_b\, \delta(y_b - y_a - P)\, \Omega(x, y), \quad \text{(A.6)}$$

where the overall delta function simply enforces the two external states to have the same energy, and we kept the integration over $y_b$ but constrained by the delta function in such a way that the integrand can be still written in terms of the canonical function of the relevant polytope – then integrating it out is geometrically equivalent to a covariant restriction[24] of the relevant geometry on to the hyperplane $y_b - y_a - P = 0$.

**Three site, one loop graph**   Let us turn now to the next-to-simplest case, the three-site one-loop graph – see Figure 10. The external kinematics is parametrised as

$$X_j := \sum_{\vec{p} \in \mathfrak{P}_j} |\vec{p}|, \qquad P_j := \left| \sum_{\vec{p} \in \mathfrak{P}_j} \vec{p} \right|, \qquad j = 1, 2, 3, \quad \text{(A.7)}$$

where $\{\mathfrak{P}_j, j = 1, 2, 3\}$ are such that $\mathfrak{P}_1 \cup \mathfrak{P}_2 \cup \mathfrak{P}_3 = \mathfrak{P}$ and $\{\mathfrak{P}_i \cap \mathfrak{P}_j = \varnothing, \forall i \neq j \; i, j = 1, 2, 3\}$ – they are the sets of momenta at the vertices $1, 2, 3$. Notice that in case the graph has one momentum for each vertex, then $\{X_j = P_j, \forall j = 1, 2, 3\}$. The loop momentum can be parametrised in terms of

$$y_{12} := |\vec{l}|, \qquad y_{23} := |\vec{l} + \vec{P}_2|, \qquad y_{31} := |\vec{l} - \vec{P}_1|, \quad \text{(A.8)}$$

where

$$\vec{P}_j := \sum_{\vec{p} \in \mathfrak{P}_j} \vec{p}, \qquad j = 1, 2, 3. \quad \text{(A.9)}$$

For $d \geq 3$, the loop momentum is a three-fold integral over the variables (A.8). The loop integration measure is now given in terms of the squared volume of a tetrahedron in $\mathbb{P}^3$:

$$\text{Vol}^2\left\{\Sigma_3\left(y^2, P^2\right)\right\} = +\text{CM}\left(y^2, P^2\right) = \begin{vmatrix} 0 & 1 & 1 & 1 & 1 \\ 1 & 0 & y_{12}^2 & y_{23}^2 & y_{31}^2 \\ 1 & y_{12}^2 & 0 & P_2^2 & P_1^2 \\ 1 & y_{23}^2 & P_2^2 & 0 & P_3^2 \\ 1 & y_{31}^2 & P_1^2 & P_3^2 & 0 \end{vmatrix}. \quad \text{(A.10)}$$

As for the previous case, the proportionality factor depends on the volume of the simplex in one dimension less give by the $(2, 2)$-minor of (A.10), *i.e.* a triangle whose sides' volumes are given by the triple $(P_1, P_2, P_3)$

---

[24]For a general definition and discussion of the covariant restriction, see [66].

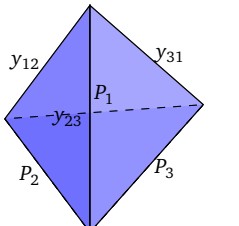 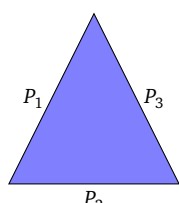

Figure 12: Example of the simplex whose volume provides the measure for the three-site one-loop graph.

The contour of integration is then given by

$$\Gamma_3 := \left\{ (-1)^{k+1} \mathrm{CM}^{(\mathcal{I}_k, \mathcal{J}_k)} \geq 0, \forall\, (\mathcal{I}_k, \mathcal{J}_k)\ k = 1, \ldots 3 \right\}, \tag{A.11}$$

where $\mathcal{I}_k$ and $\mathcal{J}_k$ are sets of $3 - k$ rows and $3 - k$ columns respectively. In words, all the minors of $CM(y_{j,j+1}^2, P_j^2)$, including the full Cayley-Menger determinant, with the appropriate (-1) factors have to be non-negative, with the equality for $(-1)^4 CM(y_{j,j+1}^2, P_j^2) = 0$ establishing the boundary of the region of the integration. The Cayley-Menger determinant $(-1)^4 CM(y_{j,j+1}^2, P_j^2)$ proportional to the squared volume of a tetrahedron whose sides have lengths $\{y_{12}, y_{23}, y_{31}, P_1, P_2, P_3\}$. The boundary of the contour of integration implies that the four vertices of the tetrahedron become co-planar. As in the previous case, this geometrical picture makes manifest the behaviour of the measure as several limits are taken. In particular, as any of the $\{y_{i,i+1}, i = 1, 2, 3\}$ is taken to zero, the tetrahedron is mapped into a triangle.

The full integral corresponding to the 3-site 1-loop graph then acquires the form

$$I_3^{(1)} = \frac{2\pi^{\frac{d-3}{2}}}{\Gamma\left(\frac{d-3}{2}\right)} \prod_{j=1}^{3} \left[ \int_0^{+\infty} \frac{dx_j}{x_j} x_j^\alpha \right] \int_{\Gamma_3} \prod_{e \in \mathcal{E}} dy_e^2 \frac{\left[(-1)^4 \mathrm{CM}\left(y^2, P^2\right)\right]^{\frac{d-4}{2}}}{\left[(-1)^3 \mathrm{CM}^{(2,2)}\left(P^2\right)\right]^{\frac{d-3}{2}}} \Omega_{\mathcal{G}}(x, y). \tag{A.12}$$

For $d < n_e = 3$, not all the edge weights are independent and, hence, the integral is $d$-fold integral.

**Two-site, two-loop graph**  Let us conclude with a two-loop example, the two-site two-loop graph – see Figure 10. Its kinematics can be parametrised as

$$X_1 := \sum_{\vec{p} \in \mathfrak{P}_1} |\vec{p}|, \qquad X_2 := \sum_{\vec{p} \in \mathfrak{P}_2} |\vec{p}|, \qquad P := \left| \sum_{\vec{p} \in \mathfrak{P}_1} \vec{p} \right| = \left| \sum_{\vec{p} \in \mathfrak{P}_2} \vec{p} \right|, \tag{A.13}$$

where $\mathfrak{P}_1, \mathfrak{P}_2 \subset \mathfrak{P}$ such that $\mathfrak{P}_1 \cup \mathfrak{P}_2 = \mathfrak{P}$ and $\mathfrak{P}_1 \cap \mathfrak{P}_2 = \varnothing$. The edge weights of the graph instead parametrise the loop space via

$$y_a := |\vec{l}_1|, \qquad y_b := |\vec{l}_1 + \vec{l}_2 + \vec{P}|, \qquad y_x := |\vec{l}_2|. \tag{A.14}$$

Let us proceed one loop at a time, as described in the main text, focusing on the loop subgraph with edge weights $y_a$ and $y_b$. It can be taken to have external kinematics to be given by $y_d = |\vec{l}_2 + \vec{P}|$ – from a graph perspective this is equivalent to open up one of the sites into two:

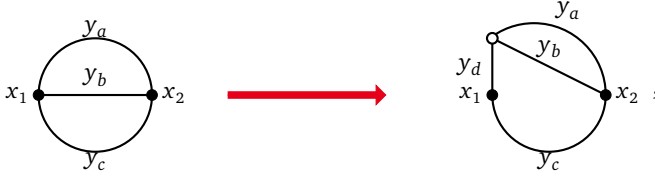



with white site not carrying any weight. Then, a measure $\mu_d^{(1)}(y_a, y_b, y_d)$ gets associated to the 2-site 1-loop subgraph being constitutes by the white site and the black one with weight $x_2$. A second measure is associated to the graph obtained by replacing the previous 1-loop subgraph by an edge with weight $y_d$ (*i.e.* the modulus of the momentum flowing through the deleted subgraph):

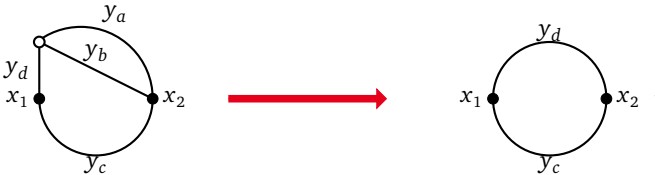

The measure associated to the 2-site 2-loop graph can be written as

$$\mu_d^{(2)}(y, P) := \int dy_d^2 \, \mu_d^{(1)}(y_a, y_b, y_d) \, \mu_d^{(1)}(y_d, y_c, P) \,, \tag{A.15}$$

with $\mu_d^{(1)}$ being the measure for the 2-site 1-loop graph computed earlier. The contour of integration is then given by $\Gamma = \Gamma_1 \bigcap \Gamma_2$ where

$$
\begin{aligned}
\Gamma_1 &= \left\{ (-1)^{k+1} \mathrm{CM}^{(\mathcal{I}_k, \mathcal{J}_k)}(y_a, y_b, y_d) \geq 0, \quad \forall\, (\mathcal{I}_k, \mathcal{J}_k),\ k = 1, 2 \right\}, \\
\Gamma_2 &= \left\{ (-1)^{k+1} \mathrm{CM}^{(\mathcal{I}_k, \mathcal{J}_k)}(y_d, y_c, P) \geq 0, \quad \forall\, (\mathcal{I}_k, \mathcal{J}_k),\ k = 1, 2 \right\}.
\end{aligned} \tag{A.16}
$$

The integral associated to 2-site 2-loop graph acquires the form

$$I_2^{(2)} \sim \int\limits_0^{+\infty} \frac{dx_1}{x_1} x_1^\alpha \int\limits_0^{+\infty} \frac{dx_2}{x_2} x_2^\alpha \int_\Gamma \prod_{e \in \mathcal{E} \cup \{e_d\}} dy_e^2 \, \mu_d^{(1)}(y_a, y_b, y_d) \mu_d^{(1)}(y_d, y_c, P) \, \Omega_{\mathcal{G}}(x, y) \,, \tag{A.17}$$

where $\sim$ indicates the omission of factors which are irrelevant to the present discussion, and $e_d$ is the edge with weight $y_d$. Note that the integrand $\Omega_{\mathcal{G}}(x.y)$ does not depend on the additional variable $y_d$ which, consequently, can be integrated out returning a measure that depends only on the edge weights of the original graph as well as its external kinematics.

## B Sector decomposition and divergences

Despite the presence of a number of examples in the main body of the paper, we consider useful to discuss in detail here a simple example that can be of help to fix the main ideas. Let us consider the simplest possible, but yet non-trivial, example of an integral of the type (46)

$$I[\sigma] := \int_0^{+\infty} \frac{dx_1}{x_1} x_1^{s_1} \int_0^{\infty} \frac{dx_2}{x_2} x_2^{s_2} \frac{1}{(X_{\mathcal{G}} + x_1 + x_2)^\tau} \equiv \int_0^{+\infty} \left[ \frac{dz}{z} z^\sigma \right] \frac{1}{(X_{\mathcal{G}} + \mathfrak{e}_{12} \cdot z)^\tau} \,, \tag{B.1}$$

where, as in the main text, $z := (x_1, x_2)$, $\sigma := (s_1, s_2) \in \mathbb{C}^2$ and $\mathfrak{e}_{12} := (1, 1) \in \mathbb{R}^2$. It can be thought to be associated to a single site graph obtained from a two-site tree graph by collapsing its two sites onto each other. The possible asymptotic divergent direction and the way that they are taken are encoded into the Newton polytope associated to the polynomial $1 + \mathfrak{e}_{12} \cdot z$ and shifted by $-\mathrm{Re}\{\sigma\}$, which is a simple triangle, whose facets are identified by the co-vectors

$$\mathfrak{W}'^{(1)} = \begin{pmatrix} -\mathrm{Re}\{s_1\} \\ -\mathfrak{e}_1 \end{pmatrix}, \quad \mathfrak{W}'^{(2)} = \begin{pmatrix} -\mathrm{Re}\{s_2\} \\ -\mathfrak{e}_2 \end{pmatrix}, \quad \mathfrak{W}^{(12)} = \begin{pmatrix} \mathrm{Re}\{s_1\} + \mathrm{Re}\{s_2\} - \mathrm{Re}\{\tau\} \\ \mathfrak{e}_{12} \end{pmatrix}, \tag{B.2}$$

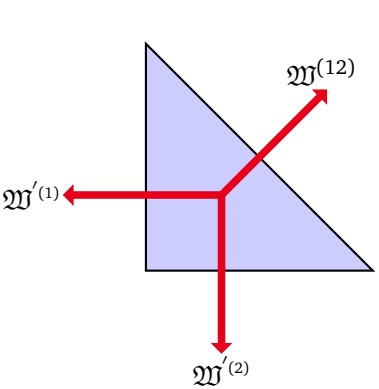

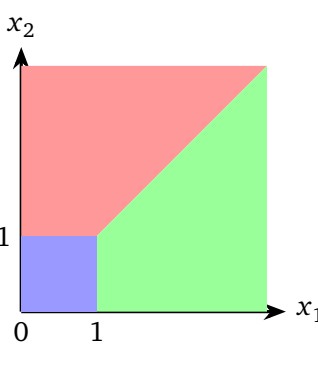

Figure 13: *On the left:* Newton polytope associated to (B.1). Its facets are identified by $\mathfrak{W}'^{(1)} := (-\text{Re}\{s_1\}, -1, 0)^{\text{T}}$, $\mathfrak{W}^{(12)} := (\text{Re}\{s_1 + s_2 - \tau\}, 1, 1)$, $\mathfrak{W}'^{(2)} := (-\text{Re}\{s_2\}, 0, -1)$ and divide the region of integration in three sectors, each of which bounded by a pair of co-vectors associated to the facets. *On the right:* Decomposition into sectors of the domain of the integration in the original integration variables. The blue square is the sector identified by the pair $\left(\mathfrak{W}'^{(2)}, \mathfrak{W}'^{(1)}\right)$ – it is the only sector containing the possible infra-red divergences. The red and the green areas instead single out the sector identified by $\left(\mathfrak{W}'^{(1)}, \mathfrak{W}^{(12)}\right)$ and $\left(\mathfrak{W}^{(12)}, \mathfrak{W}'^{(2)}\right)$.

where, as before, $\{\mathfrak{e}_j \in \mathbb{R}^2, j = 1, 2\}$ is the canonical basis for $\mathbb{R}^2$. Being a triangle, all its facets are compatible with each other, and thus, the domain of integration can be divided into three sectors, identified by all the three pairs of co-vectors (B.2).

The behaviour in each sector is made manifest via the change of variables

$$\left(\mathfrak{W}'^{(2)}, \mathfrak{W}'^{(1)}\right): \quad x_1 = \left(\zeta_2'\right)^{-\mathfrak{e}_1 \cdot \omega_2'} \left(\zeta_1'\right)^{-\mathfrak{e}_1 \cdot \omega_1'} = \zeta_1', \qquad x_2 = \left(\zeta_2'\right)^{-\mathfrak{e}_2 \cdot \omega_2'} \left(\zeta_1'\right)^{-\mathfrak{e}_2 \cdot \omega_1'} = \zeta_2',$$

$$\left(\mathfrak{W}'^{(1)}, \mathfrak{W}^{(12)}\right): \quad x_1 = \left(\zeta_1'\right)^{-\mathfrak{e}_1 \cdot \omega_1'} \left(\zeta_{12}\right)^{-\mathfrak{e}_1 \cdot \omega_{12}} = \frac{\zeta_1'}{\zeta_{12}}, \qquad x_2 = \left(\zeta_1'\right)^{-\mathfrak{e}_2 \cdot \omega_1'} \left(\zeta_{12}\right)^{-\mathfrak{e}_2 \cdot \omega_{12}} = \frac{1}{\zeta_{12}},$$

$$\left(\mathfrak{W}^{(12)}, \mathfrak{W}'^{(2)}\right): \quad x_1 = \left(\zeta_{12}\right)^{-\mathfrak{e}_1 \cdot \omega_{12}} \left(\zeta_2'\right)^{-\mathfrak{e}_1 \cdot \omega_2'} = \frac{1}{\zeta_{12}}, \qquad x_2 = \left(\zeta_{12}\right)^{-\mathfrak{e}_2 \cdot \omega_{12}} \left(\zeta_2'\right)^{-\mathfrak{e}_2 \cdot \omega_2'} = \frac{\zeta_2'}{\zeta_{12}},$$

$$\tag{B.3}$$

and the integral (B.1) can be written as

$$\mathcal{I}[\sigma] = \left[\mathcal{X}_{\mathcal{G}}\right]^{s_1 + s_2 - \tau} \int_0^1 \frac{d\zeta_1'}{\zeta_1'} \left(\zeta_1'\right)^{s_1} \int_0^1 \frac{d\zeta_2'}{\zeta_2'} \left(\zeta_2'\right)^{s_2} \frac{1}{\left(1 + \zeta_1' + \zeta_2'\right)^{\tau}}$$

$$+ \left[\mathcal{X}_{\mathcal{G}}\right]^{s_1 + s_2 - \tau} \int_0^1 \frac{d\zeta_1'}{\zeta_1'} \left(\zeta_1'\right)^{s_1} \int_0^1 \frac{d\zeta_{12}}{\zeta_{12}} \left(\zeta_{12}\right)^{\tau - s_1 - s_2} \frac{1}{\left(1 + \zeta_1' + \zeta_{12}\right)^{\tau}} \tag{B.4}$$

$$+ \left[\mathcal{X}_{\mathcal{G}}\right]^{s_1 + s_2 - \tau} \int_0^1 \frac{d\zeta_{12}}{\zeta_{12}} \left(\zeta_{12}\right)^{\tau - s_1 - s_2} \int_0^1 \frac{d\zeta_2'}{\zeta_2'} \left(\zeta_2'\right)^{s_2} \frac{1}{\left(1 + \zeta_2' + \zeta_{12}\right)^{\tau}},$$

where the three integrals correspond to three sectors as in (B.3) – it is straightforward to check that, upon the change of variables in (B.3), the region of integration $\mathbb{R}^2$ is split into the domains $\Delta' := \{x_1 \in [1, +\infty[, x_2 \in [1, +\infty]\}$, $\Delta_{1'2} := \{x_1 \in [0, x_2], x_2 \in [1, +\infty]\}$ and $\Delta_{12'} := \{x_1 \in [1, +\infty[, x_2 \in [0, x_1]\}$ as shown in Figure 2, where they respectively correspond the blue, red and green regions. The three integrals in (B.4) are convergent for different values of the parameters $(s_1, s_2, \tau)$, concretely: $(\text{Re}\{s_1\} > 0, \text{Re}\{s_2\} > 0)$, $(\text{Re}\{s_1\} > 0, \text{Re}\{\tau - s_1 - s_2\} > 0)$ and $\text{Re}\{\tau - s_1 - s_2\} > 0, (\text{Re}\{s_1\} > 0)$.

Let us consider the limit for which both $s_1$ and $s_1$ are taken to zero. This is equivalent to taking both the directions $\mathfrak{W}^{(1)}$ and $\mathfrak{W}^{(2)}$ to be divergent. Then the leading contribution is given by the first integral in (B.4) only:

$$\mathcal{I}_{\Delta'}[\sigma] \sim \int_0^1 \frac{d\zeta_1'}{\zeta_1'} (\zeta_1')^{s_1} \int_0^1 \frac{d\zeta_2'}{\zeta_2'} (\zeta_2')^{s_2} + \ldots = \frac{1}{s_1 s_1} + \ldots, \tag{B.5}$$

where $\mathcal{I}_{\Delta'}[\sigma]$ is the integral in the first line of (B.4), without the pre-factor $\left[\mathcal{X}_{\mathcal{G}}\right]^{s_1+s_2-\tau}$. In order to extract all subleading divergences, one can rewrite $(1+\zeta_1'+\zeta_2')^{-\tau}$ as a double Mellin-Barnes integral

$$\frac{1}{\left(1+\zeta_1'+\zeta_2'\right)^\tau} = \frac{1}{\Gamma(\tau)} \int_{-i\infty}^{+i\infty} d\xi_1 (\zeta_1')^{\xi_1} \int_{-i\infty}^{+i\infty} d\xi_2 (\zeta_2')^{\xi_2} \Gamma(-\xi_1)\Gamma(-\xi_2)\Gamma(\tau+\xi_1+\xi_2), \tag{B.6}$$

so that the integrations over $\{\zeta_j', j=1,2\}$ are of the same form as (B.5), but with the powers shifted, $(s_1, s_2) \longrightarrow (s_1 + \xi_1, s_2 + \xi_2)$, and can be performed giving the simple factor $[(s_1+\xi_1)(s_2+\xi_2)]^{-1}$. The contour integral can be then performed by closing both contour of integration in the positive half plane, providing a series representation for $\mathcal{I}_{\Delta'}$ which can be now safely expanded for $(s_1, s_2) \longrightarrow (0,0)$ to give:

$$\mathcal{I}_{\Delta'}(\sigma) \sim \frac{1}{s_1 s_2} - \frac{\tau\left(2\tau^2 - 3\tau + 31\right)}{36}\left(\frac{1}{s_1} + \frac{1}{s_2}\right) + \ldots \tag{B.7}$$

Indeed, contributions to the subleading divergences are given by the other two sectors as well

$$\mathcal{I}_{\Delta_{1'2}}[\sigma] \sim \frac{1}{s_1} \times \int_0^1 \frac{d\zeta_{12}}{\zeta_{12}} (\zeta_{12})^\tau \frac{1}{(1+\zeta_{12})^\tau},$$
$$\mathcal{I}_{\Delta_{12'}}[\sigma] \sim \frac{1}{s_2} \times \int_0^1 \frac{d\zeta_{12}}{\zeta_{12}} (\zeta_{12})^\tau \frac{1}{(1+\zeta_{12})^\tau}, \tag{B.8}$$

with the leftover integral which evaluate to a Gaussian hypergeometric function.

A similar treatment can be carried out in the other limits. In particular, the infra-red behaviour of $\mathcal{I}[\sigma]$ is encoded into the direction $\mathfrak{W}^{(12)}$. It becomes divergent for $\tau - s_1 - s_2 \longrightarrow 0$ and receives contribution from both $\mathcal{I}_{\Delta_{1'2}}[\sigma]$ and $\mathcal{I}_{\Delta_{12'}}[\sigma]$. Indicating both of them as $\mathcal{I}_{\Delta_{1'2'}}[\sigma]$, their leading behaviour can be written as,

$$\mathcal{I}_{\Delta_{1'2'}}[\sigma] \sim \int_0^1 \frac{d\zeta_{12}}{\zeta_{12}} (\zeta_{12})^{\tau-s_1-s_2} \times \int_0^1 \frac{d\zeta_j'}{\zeta_j'} (\zeta_j')^{s_j} \frac{1}{\left(1+\zeta_j'\right)^\tau}$$
$$= \frac{1}{\tau - s_1 - s_2} \times \int_0^1 \frac{d\zeta_j'}{\zeta_j'} (\zeta_j')^{s_j} \frac{1}{\left(1+\zeta_j'\right)^\tau}, \tag{B.9}$$

with the integral evaluating to a Gaussian hypergeometric function. In the infra-red limit, the integral factorises into two integrals, one containing the divergence, which manifests itself as the pole in $\tau - s_1 - s_2$ – signalling a logarithmic divergence – and the other one which is finite.

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
