# Peer review of "The Asymptotic Structure of Cosmological Integrals"

_SciPost Physics, doi:SciPost Phys. 19, 029 (2025)_

## Round 2 · Referee Report · Anonymous (Referee 1) · 2024-11-15

Strengths

  1. This work systematically investigates the UV and IR divergences in the perturbative computation of the Bunch-Davies wavefunction, with the combinatoric perspective of cosmological polytope. The underlying question is about the resummability of subleading IR divergences, which is well-defined and deserves careful consideration.
  2. In particular, the authors focused on the cosmological integrals from a power-law FRW background, and examined the presence (and the degree) of divergences for general tree-level and loop-level processes.
  3. Certainly this paper made a step forward to tackle the chanllege of loop correlators in cosmology.

Weaknesses

The authors aim for solving universal questions on cosmological correlators and take a combinatiorial perspective toward this goal, but in the end of the analysis the connection with these questions seems a bit weak and less concrete.

Report

The paper contains interesting results and detailed analysis. I think it is well-fitted for publication in SciPost Physics, though some revisions may be needed for further improvement and clarification.

Requested changes

  1. One major motivation of the current paper is to understand how the resummation of divergences leads to IR-safe observables in cosmology. This question has been extensively discussed in the context of interacting light scalars in de Sitter spacetime. While the authors here look into a more general setup of power-law FRW universe, it will be very helpful to make the connection with the well-studied dS examples. For instance, can the new method here reproduce the standard results of $\lambda \phi^4$ in dS? There have been several recent works using the Bunch-Davies wavefunction to study these IR effects. Would the combinatorial analysis agree with them at both tree- and loop- levels? If so, what are the new ingredients here? With this part, I think the paper will be more complete and also become beneficial to readers who are less familiar with the polytope language.

  2. One confusing point is about the power-law divergences. Unlike the logarithmic divergences, these can be really dangerous instabilities which are not resummable. I understand that they may arise in certain forms of the cosmological integrals, but are they mathematical constructions only, or can be physical contributions to the real parts of wavefunction coefficients? It will be surprising to see any power-law form divergences in realistic scenarios.

  3. The authors discussed about the renormalization group structure in the combinatiorial approach, but it is less clear how the current analysis, which is still within perturbation theory, can shed light on the non-perturbative RG. Are there certain advantages of the polytope formalism which cannot be gained in other approaches? This connection (if exist) may need to be further clarified.

Recommendation

Publish (meets expectations and criteria for this Journal)

  • validity: top
  • significance: high
  • originality: top
  • clarity: high
  • formatting: perfect
  • grammar: perfect

Author:  Francisco Vazao  on 2025-02-10  [id 5203]

(in reply to Report 1 on 2024-11-15)

We thank the referee for recommending our paper for publication. In the following answer, we address the referee's comments/concerns.

We will start with the weakness found by the referee. It is not true that in this paper we aim to solve any universal questions on cosmological correlators. Those questions are indeed the motivation for us to embark on the work that originated this paper. However, the paper in and of itself is the first step towards such a solution, and it concerns the development of a novel technology based on combinatorics, and recognizing that the asymptotic structure of the integrals that contribute to physical processes is controlled by a combinatorial object that can be predicted for an arbitrary graph, and with it the degree of divergence and the divergence coefficients. We found such a proof important enough to propose a paper to be published, but we have never claimed to have solved the full problem. We do think that these findings will help us to solve the problem in a very transparent way, but this will be the subject of future publications. 

We now move on to comment on the requested changes.

  1. We find that this request is intimately tied to what we explained in the paragraph above: our paper does not aim to solve the full IR problem, which is indeed one of the motivations for writing it, but to develop a technology that can allow having a deeper understanding of it, and it identifies that the asymptotic behaviour can be generally predicted (and the coefficients of the divergences more generally computed) because it is governed by a special combinatorial structure for all graphs. Looking at problems like $\lambda\phi^4$ in $dS_4$ -- the case more extensively studied in the literature -- is beyond the scope of this paper, and it will be the subject of future publications. We would also like to point out that our findings already required a very long paper. Requiring to face the IR problem in it, will make the paper inevitably and unnecessarily longer, as the method+structure and the physical problem can be separated. And this was our declared intention from the start.

  2. The cosmological integrals depend on certain parameters -- which we indicate with $\alpha$ -- that encode a substantial amount of information, among which is how fast the universe expands. If the universe expands sufficiently fast, these power-law divergences can appear. Is this a physical scenario? From our point of view, one of our general goals (not of this paper but of our approach) is to understand whether there is a pattern (or more generally some structure) in the IR divergences in perturbation theory that tells us that they are bound to resum, cancel or make the theory pathological without having to do perform any other computation. For this reason, we need to consider any possible behaviour, irrespectively of the scenario. Said differently, we aim to find consistency conditions on the IR behaviour that tells us whether the theory is healthy or not.

  3. We only comment on the renormalization group in the outlook, as an interesting link to understand. Even in flat space, an understanding of the RG that is more suited to modern techniques (e.g. the on-shell approach which is the inspiration for the cosmological bootstrap) is still not available. We only argued that our combinatorial approach might lead to a different understanding of the RG in the long run. In the specific case of dS, the leading IR divergences are found to resum so that the probability distribution satisfies a Fokker-Planck equation (compatibly with stochastic inflation). However, a Fokker-Planck/diffusion equation is an RG equation (and any RG equation can be put in a diffusion equation form). In this sense, if our approach can shed light on the IR problem, it also has the potential to provide a novel understanding of the RG. Now, it is true that the RG concerns UV divergences, whose physical meaning is completely different from the IR. However, our combinatorial approach unifies them (putting them on the same mathematical footing), thus allowing a complete understanding of the RG. Furthermore, having a combinatorial/polytope description gives us novel mathematical rules whose exploitation might lead us beyond the realm in which such a description has been formulated. Again, all this is based on our intuition, and for this reason, this point has been just mentioned in the outlook as a future research direction to explore.

---

## Round 2 · Referee Report · Anonymous (Referee 2) · 2024-12-4

Strengths

1 - this paper develops the mathematical technology needed for a systematic understanding of loop corrections for cosmological correlators in FRW spacetimes.

2 - the approach mirrors similar techniques in flat space and thus avoids many of the pitfalls of cosmological loop calculations (namely that they break symmetries, etc).

Weaknesses

1 - this paper is not written for an audience that works on cosmological correlators, except those familiar with these techniques already. There is a high bar of mathematical sophistication that is unique to their approach that is needed to understand the results.

2 - there is very little physics in this paper. The examples are phrases in terms of graphs, not physical processes. This will make it difficult for someone working on loops from a more cosmological perspective to match their intuition with this approach.

Report

This is a very technical paper, but it seems correct and likely to be very useful to a group of researchers working on these kinds of FRW correlators. Benincasa has a long history of writing papers of this kind that are ultimately of high impact but may take others to translate the tools to more physical situations. It is also the case that most progress in direct calculations of cosmological loop corrections have been build on similar mathematical developments and therefore this will surely find use in the community. I recommend publication in the current form.

Recommendation

Publish (surpasses expectations and criteria for this Journal; among top 10%)

  • validity: high
  • significance: good
  • originality: high
  • clarity: ok
  • formatting: good
  • grammar: good

Author:  Francisco Vazao  on 2025-02-10  [id 5204]

(in reply to Report 2 on 2024-12-04)

We thank the referee for recommending the publication of our paper in its current form. However, we would like to address some of the comments in the report.

We find that both the alleged weaknesses contradict the strengths. Precisely because the paper develops a mathematical technology (strength1) typically unfamiliar to cosmologists, it is unavoidable that it is an unusual paper for the 'cosmological correlator community'. However, we reviewed just the basic material needed to follow the paper. Indeed, as the review part has this aim (and thus is not a full-fledged review, which, in our opinion, would have been unnecessary and would have made the paper unnecessarily longer than what it already is) is also unavoidable that understanding more deeply some of the statements in it requires to go back to the literature. The style we chose for writing the paper aims to make the paper understandable, providing that certain statements already proved in the literature are accepted by the reader (and referring to the literature for a deeper comprehension). Secondly, as the language is novel in cosmology, it unavoidably requires an extra effort on the reader's side: the reader cannot expect to read the paper with the same fluidity as it was written in a more standard field theoretical language.

Secondly, precisely because our approach mirrors similar techniques in flat space (avoiding many of the pitfalls of cosmological loop calculations -- strength 2), the analysis concerns individual graphs. Indeed, we agree with the referee that a single graph is not a full-fledged physical process. However, in the case of scalars (which is the case we treat here), no particular simplification occurs over the sum of graphs (which is true even in flat space), and, no particular redundancy is associated with them (except field redefinition in our case, which, in any case, does not alter the singularities at finite location of the graphs and of the physical process). Hence, from the analysis of individual graphs, it is still possible to extract general conclusions for the full process. Furthermore, it is likely that, as it happens in flat space, processes involving spinning states can be decomposed in terms of scalar graphs. Hence, our findings have the potential to be extended beyond the realm of scalar theories. For this reason, we find the comment "there is very little physics in the paper" too bold (and a bit unfair). Also, for the asymptotic behaviour -- which is the subject of our paper -- it is clear that graphs contribute to singularities in different ways. This can also be seen field theoretically: analysing the asymptotic behaviour of a certain physical process is often phrased in how (and which) individual graph contributes to the divergences. For this reason, we find that weakness two does not hold.

In general, we have the impression that there is some uneasiness from the referee's side to deal with a novel, and hence, unusual language. However, the purpose of our paper is exactly to develop such a language that we think is going to help us understand both the IR and UV behaviour of actual physical processes.

---

## Round 3 · Referee Report · Anonymous (Referee 1) · 2025-6-28

Report

The authors have addressed my previous comments and added the relevant discussions in the revised version. It is still a quite technical paper with unusual mathematical tools, but the new version made a bit more connection with well-know examples, and could be of interest for a broader range of readers. I am happy to recommend publication in the current form.

Recommendation

Publish (meets expectations and criteria for this Journal)

---

## Round 3 · Author Response

We thank the referees for recommending our paper for publication. In the following answer, we address the referee's comments/concerns.

We added a discussion after the one loop two point example, relating it to the existing literature on $\lambda \phi^4$ in $dS_4$, which hopefully addresses some of the concerns of referee 1.

Referee 2

We find that both the alleged weaknesses contradict the strengths. Precisely because the paper develops a mathematical technology (strength1) typically unfamiliar to cosmologists, it is unavoidable that it is an unusual paper for the 'cosmological correlator community'. However, we reviewed just the basic material needed to follow the paper. Indeed, as the review part has this aim (and thus is not a full-fledged review, which, in our opinion, would have been unnecessary and would have made the paper unnecessarily longer than what it already is) is also unavoidable that understanding more deeply some of the statements in it requires to go back to the literature. The style we chose for writing the paper aims to make the paper understandable, providing that certain statements already proved in the literature are accepted by the reader (and referring to the literature for a deeper comprehension). Secondly, as the language is novel in cosmology, it unavoidably requires an extra effort on the reader's side: the reader cannot expect to read the paper with the same fluidity as it was written in a more standard field theoretical language.

Secondly, precisely because our approach mirrors similar techniques in flat space (avoiding many of the pitfalls of cosmological loop calculations -- strength 2), the analysis concerns individual graphs. Indeed, we agree with the referee that a single graph is not a full-fledged physical process. However, in the case of scalars (which is the case we treat here), no particular simplification occurs over the sum of graphs (which is true even in flat space), and, no particular redundancy is associated with them (except field redefinition in our case, which, in any case, does not alter the singularities at finite location of the graphs and of the physical process). Hence, from the analysis of individual graphs, it is still possible to extract general conclusions for the full process. Furthermore, it is likely that, as it happens in flat space, processes involving spinning states can be decomposed in terms of scalar graphs. Hence, our findings have the potential to be extended beyond the realm of scalar theories. For this reason, we find the comment "there is very little physics in the paper" too bold (and a bit unfair). Also, for the asymptotic behaviour -- which is the subject of our paper -- it is clear that graphs contribute to singularities in different ways. This can also be seen field theoretically: analysing the asymptotic behaviour of a certain physical process is often phrased in how (and which) individual graph contributes to the divergences. For this reason, we find that weakness two does not hold.

In general, we have the impression that there is some uneasiness from the referee's side to deal with a novel, and hence, unusual language. However, the purpose of our paper is exactly to develop such a language that we think is going to help us understand both the IR and UV behaviour of actual physical processes.

Referee 1

We will start with the weakness found by the referee. It is not true that in this paper we aim to solve any universal questions on cosmological correlators. Those questions are indeed the motivation for us to embark on the work that originated this paper. However, the paper in and of itself is the first step towards such a solution, and it concerns the development of a novel technology based on combinatorics, and recognizing that the asymptotic structure of the integrals that contribute to physical processes is controlled by a combinatorial object that can be predicted for an arbitrary graph, and with it the degree of divergence and the divergence coefficients. We found such a proof important enough to propose a paper to be published, but we have never claimed to have solved the full problem. We do think that these findings will help us to solve the problem in a very transparent way, but this will be the subject of future publications.  We now move on to comment on the requested changes. 1. We find that this request is intimately tied to what we explained in the paragraph above: our paper does not aim to solve the full IR problem, which is indeed one of the motivations for writing it, but to develop a technology that can allow having a deeper understanding of it, and it identifies that the asymptotic behaviour can be generally predicted (and the coefficients of the divergences more generally computed) because it is governed by a special combinatorial structure for all graphs. Looking at problems like $\lambda \phi^4$ in $dS_4$ -- the case more extensively studied in the literature -- is beyond the scope of this paper, and it will be the subject of future publications. We would also like to point out that our findings already required a very long paper. Requiring to face the IR problem in it, will make the paper inevitably and unnecessarily longer, as the method+structure and the physical problem can be separated. And this was our declared intention from the start. The cosmological integrals depend on certain parameters -- which we indicate with α -- that encode a substantial amount of information, among which is how fast the universe expands. If the universe expands sufficiently fast, these power-law divergences can appear. Is this a physical scenario? From our point of view, one of our general goals (not of this paper but of our approach) is to understand whether there is a pattern (or more generally some structure) in the IR divergences in perturbation theory that tells us that they are bound to resum, cancel or make the theory pathological without having to do perform any other computation. For this reason, we need to consider any possible behaviour, irrespectively of the scenario. Said differently, we aim to find consistency conditions on the IR behaviour that tells us whether the theory is healthy or not. We only comment on the renormalization group in the outlook, as an interesting link to understand. Even in flat space, an understanding of the RG that is more suited to modern techniques (e.g. the on-shell approach which is the inspiration for the cosmological bootstrap) is still not available. We only argued that our combinatorial approach might lead to a different understanding of the RG in the long run. In the specific case of dS, the leading IR divergences are found to resum so that the probability distribution satisfies a Fokker-Planck equation (compatibly with stochastic inflation). However, a Fokker-Planck/diffusion equation is an RG equation (and any RG equation can be put in a diffusion equation form). In this sense, if our approach can shed light on the IR problem, it also has the potential to provide a novel understanding of the RG. Now, it is true that the RG concerns UV divergences, whose physical meaning is completely different from the IR. However, our combinatorial approach unifies them (putting them on the same mathematical footing), thus allowing a complete understanding of the RG. Furthermore, having a combinatorial/polytope description gives us novel mathematical rules whose exploitation might lead us beyond the realm in which such a description has been formulated. Again, all this is based on our intuition, and for this reason, this point has been just mentioned in the outlook as a future research direction to explore.

---

## Editorial Decision

published